# RCEMIP-II: Mock-Walker Simulations as Phase II of the Radiative-Convective Equilibrium Model Intercomparison Project

Allison A. Wing[1], Levi G. Silvers[2,3], and Kevin A. Reed[2]

[1]Department of Earth, Ocean, and Atmospheric Science, Florida State University
[2]School of Marine and Atmospheric Sciences, Stony Brook University
[3]Current affiliation: Department of Atmospheric Science, Colorado State University

**Correspondence:** Allison A. Wing (awing@fsu.edu)

**Abstract.** The Radiative-Convective Equilibrium (RCE) Model Intercomparison Project (RCEMIP) leveraged the simplicity of RCE to focus attention on moist convective processes and their interactions with radiation and circulation across a wide range of model types including cloud-resolving models (CRMs), general circulation models (GCMs), single column models, global cloud-resolving models, and large eddy simulations. While several robust results emerged across the spectrum of models that participated in the first phase of RCEMIP (RCEMIP-I), two points that stand out are (1) the strikingly large diversity in simulated climate states and (2) the strong imprint of convective self-aggregation on the climate state. However, the lack of consensus in the structure of self-aggregation and its response to warming is a barrier to understanding. Gaining a deeper understanding of convective aggregation and tropical climate will require reducing the degrees of freedom with which convection can vary. Therefore, we propose a Phase II of RCEMIP (RCEMIP-II) that utilizes a prescribed sinusoidal sea surface temperature (SST) pattern to provide a constraint on the structure of convection and move one critical step up the model hierarchy. This so-called "mock-Walker" configuration generates features that resemble observed tropical circulations. The specification of the mock-Walker protocol for RCEMIP-II is described, along with example results from one CRM and one GCM. RCEMIP-II will consist of five required simulations: three simulations with the same three mean SSTs as in RCEMIP-I but with an SST gradient, as well as two additional simulations at one of the mean SSTs with different values of the SST gradients. We also test the sensitivity to the imposed SST gradient and the domain size. Under weak SST gradients, unforced self-aggregation emerges across the entire domain, similar to what was found in RCEMIP. As the SST gradient increases, the convective region narrows and is more confined to the warmest SSTs. At warmer mean SSTs and stronger SST gradients, low-frequency variability of the convective aggregation emerges, suggesting that simulations of at least 200 days may be needed to achieve robust equilibrium statistics in this configuration. Simulations with different domain sizes generally have similar mean statistics and convective structures, depending on the value of the SST gradient. The prescribed SST boundary condition is the only difference in the setup between RCEMIP-II and RCEMIP-I, which enables comparison between the two; however, we also welcome participation in RCEMIP-II from models that did not participate in RCEMIP-I.

# 1 Introduction

On Earth, the tropics play an important role in climate through coupled interactions of clouds, circulation, and radiative fluxes. As such, tropical regions influence the global energy balance and tropical variability has far-reaching global effects on weather patterns and extremes. Thus, understanding the underlying mechanisms that connect climate, clouds and circulation remains paramount in climate science, particularly in the context of global warming (Bony et al., 2015). Global models, such as general circulation models, have evolved over recent decades to advance our understanding and simulation of the climate system, but biases and uncertainty in climate projections remain. The links between tropical circulation and clouds on a wide range of space and time scales are complicated by a host of scale interactions that are challenging to effectively represent in comprehensive Earth system models.

Given the complexities of these interactions, theory and idealized models of the tropical atmosphere are important tools for advancing understanding. Idealization allows for experiments that target specific features of interest, while retaining the fundamental properties of the tropical climate. Furthermore, the simplicity and flexibility of idealized models provides an opportunity for developing new understanding. One line of idealized modeling has focused on radiative-convective equilibrium (RCE), which approximates the tropical atmosphere as a statistical balance between radiative cooling and convective heating. RCE represents the simplest approximation of not only the tropics, but global climate more broadly. There has been a long history exploring RCE in idealized models for advancing understanding of the tropics (e.g., Manabe and Strickler, 1964; Held et al., 1993; Tompkins and Craig, 1998; Satoh et al., 2016; Held et al., 2007; Popke et al., 2013).

The RCE Model Intercomparison Project (RCEMIP; Wing et al., 2018) was a coordinated international effort to standardize the boundary conditions and forcing assumptions of RCE simulations to allow for the intercomparison across various atmospheric model types. RCEMIP included cloud-resolving models (CRMs), large eddy simulation models (LES), global cloud-resolving models (GCRMs) and general circulation models (GCMs). The initial phase of RCEMIP (hereafter RCEMIP-I) focused on changes in clouds and convective activity with surface warming, as well as associated cloud feedbacks, the implications for climate sensitivity, and the role of aggregation in tropical climate. Results of RCEMIP-I have revealed robust behaviors across the model hierarchy. For example, in response to SST warming, deep convective anvil clouds rise, slightly warm, thin, and decrease in extent (Wing et al., 2020a; Stauffer and Wing, 2022, 2023) and the large-scale circulation weakens (Silvers et al., 2023). Self-aggregation occurs across nearly all large domain RCEMIP-I simulations and it consistently warms and dries the mean state with a reduction in the extent of high clouds (Wing et al., 2020a; Stauffer and Wing, 2022). Cloud-radiative feedbacks are consistently found to be the most impoortant mechanism driving self-aggregation (Pope et al., 2023) and variability in the degree of self-aggregation modulates cloud feedbacks and climate sensitivity (Becker and Wing, 2020; Stauffer, 2023). The fact that these common behaviors emerged across the wide variety of model numerics and physics present in RCEMIP indicates that they are a result of fundamental physical mechanisms. However, the intercomparison also revealed substantial disagreement in the representation of mean profiles of temperature, humidity, and cloudiness, a wide range of static stability and climate sensitivities, a large variation in the degree of convective aggregation, and no consensus in the response of aggregation to warming (Wing et al., 2020a; Becker and Wing, 2020; Wing and Singh, 2024; Silvers et al., 2023).

RCEMIP-I prescribed homogeneous thermal forcings at the boundaries, which consisted of uniform sea surface temperature (SST) and insolation. The simplification of the boundary conditions demonstrates how sensitive the simulation of convection and its complex interactions with radiation and circulation are to model design. The divergent behavior in RCEMIP-I reveals dependencies on representations of convection, microphysics, turbulence, and dynamical cores that may have been masked in other intercomparisons by dynamical constraints. To build on this work we propose a second phase of RCEMIP (hereafter RCEMIP-II) in which standardized heterogeneities in the SST are prescribed to explore robust behaviors in the tropical system when strict RCE is relaxed to include more realistic, but still idealized, circulations.

A "mock-Walker" circulation (Raymond, 1994) confines deep convection to regions of low-level convergence and large-scale ascent, determined by the SST pattern (Lindzen and Nigam, 1987; Bretherton and Sobel, 2002; Back and Bretherton, 2009). The general features resemble observed tropical circulations (Grabowski et al., 2000; Larson and Hartmann, 2003), reflecting an interplay of convection, radiation, humidity, and large-scale circulation that is fundamental to tropical deep convective regions (Bretherton et al., 2006). While the SST pattern fixes the location of the circulation and constrains its strength, interactions between convection, surface gustiness, water vapor, and radiation modify the large-scale flow and the strength and spatial extent of convection (Tompkins, 2001; Liu and Moncrieff, 2008; Wofsy and Kuang, 2012; Silvers and Robinson, 2021). Cloud radiative feedbacks narrow the convective region in a manner reminiscent of self-aggregation (Grabowski et al., 2000; Bretherton and Sobel, 2002; Liu and Moncrieff, 2008), similar to the zonal contraction of convection in spherical simulations with zonally uniform, meridionally varying SST (Müller and Hohenegger, 2020).

The prescribed SST gradient in the RCEMIP-II mock-Walker simulations will drive a large-scale circulation that provides a partial, dynamical constraint on the structure of convection, compared to the strict RCE set-up in RCEMIP-I in which there were no external constraints on the location or spatial pattern of convection. The "forced" aggregation in RCEMIP-II provides a common null expectation for convective structure while still allowing for "un-forced" self-aggregation intrinsic to the model to emerge via radiative-convective feedbacks (e.g., Grabowski et al., 2000; Bretherton and Sobel, 2002; Liu and Moncrieff, 2008; Müller and Hohenegger, 2020). By varying the strength of the SST gradient with warming, the impact of aggregation on climate sensitivity will be able to be attributed to the forced vs. unforced aggregation. An additional motivation for the simulations described below is that mock-Walker simulations are a valuable component of the model hierarchy for tropical dynamics (Jeevanjee et al., 2017) and can be performed by both CRMs and GCM-like models in limited-area planar domains and GCMs on the global sphere. By changing just one parameter compared to the RCEMIP-I simulations (the analytic SST boundary condition) the proposed RCEMIP-II simulations represent a single, clean move up the model hierarchy of model complexity that is closer to the observed tropics than strict RCE.

This paper is organized as follows. Section 2 discusses the motivation for the experimental design of RCEMIP-II in more detail and provides an overview of the scientific objectives. The experimental design, which builds on RCEMIP-I to add an SST pattern to enable an intercomparison of idealized tropical circulations across the model hierarchy, is described in detail in Section 3. Section 4 presents sample results for RCEMIP-II to demonstrate the setup and Section 5 provides some next steps and brief discussion.

## 2 Motivation and Science Objectives

RCEMIP was designed to explore questions about tropical clouds, convective aggregation, and climate, with a particular focus on three scientific themes:

1. The robustness of the simulated mean state across the spectrum of models,

2. The response of convective clouds to warming and climate sensitivity,

3. The dependence of convective self-aggregation on temperature.

While RCEMIP-I dictated common domain configurations, grid spacing, trace gas concentrations, insolation, and SST boundary conditions, it purposefully minimized the code changes (such as modifications to physical schemes) needed to run the simulations, both to ensure broad participation by the community and to reveal the true spread in RCE states using a model's

"out of the box" suite of physical schemes (Wing et al., 2018). As summarized in the prior section, several common behaviors emerged from RCEMIP-I despite the great diversity in model physics and numerics, providing strong evidence for fundamental physical mechanisms that are not dependent on the details of physical parameterizations. The model diversity also provided an opportunity to use simple theory to explain the intermodel spread (Wing and Singh, 2024). However, two other points that stand out in considering the RCEMIP-I results are (1) how strikingly large the spread in simulated RCE states is; and (2) how

strong of an imprint convective self-aggregation has on the climate state (Wing et al., 2020a). In particular, the wide range in the degree of self-aggregation and the lack of consensus in its temperature dependence is a barrier to understanding.

Our vision for RCEMIP was always that the initial simulations would serve as a starting point, but deep understanding would require performing additional simulations to address issues such as the robustness of the results to experimental design, the sensitivity to model physics and dynamics, and the impact of other factors such as ocean-atmosphere interactions or rotation.

In considering possibilities for RCEMIP-II we sought a protocol that follows the following four principles, in the spirit of the design of RCEMIP-I:

1. The ability to directly compare limited-area models with explicit convection and global climate models with parameterized convection,

2. Ease of implementation, to encourage the broadest possible participation,

3. Continued investigation of the above three themes of RCEMIP, while moving a step up the model hierarchy of complexity,

4. Providing an external constraint on convection.

The proposed mock-Walker configuration follows the above philosophy. As described in detail below, it maintains an identical set-up to RCEMIP-I with the exception of a simple, prescribed SST pattern. This is easy to implement and allows for direct

comparison with RCEMIP-I. Care was taken to maintain consistency between the CRM and GCM domains, as much as possible. These characteristics satisfy principles 1 and 2.

From its inception, RCEMIP has been motivated in part by a desire to better understand how the balance between convection and radiation interacts with large-scale circulations (Wing et al., 2018). However, the only large-scale circulations present in RCEMIP-I are those generated by self-aggregation (Silvers et al., 2023). One of our motivations for selecting mock-Walker simulations for RCEMIP-II is a desire to broaden the range of dynamical regimes and cloud types that can be simulated by moving one step up the model hierarchy from RCE (principle 3). In order to do so, we need to relax the idealization of the RCE boundary conditions. Instead of uniform SST, a prescribed sinusoidal SST pattern provides a more realistic (i.e., heterogeneous) boundary condition with a clearer tie to observations. Interactions between convection and a large-scale circulation that is forced by SST anomalies have direct analogues on Earth to the ITCZ, the Walker Circulation, and the Hadley Circulation (in a non-rotating context). The presence of subsiding circulations consistently occurring over regions of cooler SST also allows for the possibility of simulations that include stratocumulus clouds. The SST gradient, combined with an overturning circulation, also allows for the possibility of modeling the transition between shallow and deep convective clouds. The mock-Walker configuration thus allows investigation on the three themes of RCEMIP in a framework focused on cloud-circulation coupling that is one step up the model hierarchy from RCE, satisfying principle 3. In particular, varying the SST gradient with mean warming will permit an investigation of the "pattern effect", supporting the second theme of RCEMIP in a new way compared to RCEMIP-I. The pattern of SST warming, that is, whether warming is enhanced in warm regions where deep convection occurs or in cold regions where there is subsidence and low clouds, is known to influence climate feedbacks (Andrews et al., 2022; Fueglistaler and Silvers, 2021). Prior work using GCMs has shown that if warming is focused on subsidence regions with cold SST, this has little remote effect and results in a negative lapse rate feedback and large positive low cloud feedback. However, if the warming is instead enhanced in regions of deep convection, the warming is communicated through the free troposphere resulting in increased stability that leads to a stronger negative lapse rate feedback and a weaker (less positive/more negative) low cloud feedback through increases in low-level cloud cover in the cold SST region (Andrews and Webb, 2018; Ceppi and Gregory, 2017; Dong et al., 2019; Zhou et al., 2016). An RCEMIP-II with mock-Walker simulations will allow us to investigate these cloud feedbacks on climate in a setting with both CRMs and GCMs, satisfying principles 1 and 3.

The prescribed SST gradient in the mock-Walker simulations provides forcing for low-level convergence towards the warmest SSTs (Lindzen and Nigam, 1987; Bretherton and Sobel, 2002; Back and Bretherton, 2009) and will drive a large-scale circulation that provides a dynamical constraint on the location and spatial pattern of convection relative to uniform SSTs, satisfying principle 4. The ability of SST gradients to at least partially dynamically constrain and organize convection is clear in the test simulations described below as well as numerous previous studies (Grabowski et al., 2000; Tompkins, 2001; Bretherton et al., 2006; Lutsko and Cronin, 2018; Silvers and Robinson, 2021). The extent to which the prescribed SST gradient constrains convection and circulation in an environment of complex interactions between moist convective processes, radiation, and microphysics will be a subject of investigation across the RCEMIP-II ensemble.

Other possibilities were considered for RCEMIP-II, such as rotation, interactive SST with a slab mixed-layer ocean, and simplified physics. However, none of these options satisfy the above four principles as well as the mock-Walker configuration. It is not possible to satisfy principle 1 while adding rotation, since f-plane RCE simulations in a limited area CRM domain are quite different than realistic rotation on the GCM sphere with uniform thermal forcing (Merlis and Held, 2019; Sobel

et al., 2021). There is no obvious CRM analog to the latter other than a global CRM or perhaps a large beta-plane, but either would be more computationally expensive, in opposition to principle 2. While there are many important questions that could be investigated in simulations with rotation, they are likely to be tropical cyclone-focused, rather than supporting the current themes of RCEMIP, opposing principle 3. Silvers et al. (2024) provide an example of rotating RCE experiments using a GCM in the context of RCEMIP-I as the control configuration. Simulations with interactive SST involve jumping much further up the model hierarchy and add significant complexity, which complicates interpretation (Coppin and Bony, 2017, 2018; Drotos et al., 2020; Hartmann and Dygert, 2022; Dygert and Hartmann, 2023). Slab mixed-layer oceans of even relatively shallow depth take many hundreds of days to equilibrate (Cronin and Emanuel, 2013), greatly increasing the computational expense and opposing principle 2. Simplified physics (i.e., microphysics and radiation) is more challenging to implement in most models, opposing principle 2 and would move down the hierarchy towards more idealization and could remove some phenomena of interest, opposing principle 3.

By utilizing a mock-Walker configuration, RCEMIP-II will satisfy all four of the above principles and will facilitate deeper understanding of convective aggregation and its role in climate - including the hydrological cycle, cloud feedbacks, and climate sensitivity.

## 3  Experimental Design

The experimental design of the RCEMIP-II simulations follows the philosophy set out by RCEMIP-I (Wing et al., 2018) and in line with the principles describe above; that is, a small set of experiments that are designed to maximize the utility of the simulations in answering the above questions, while minimizing the effort required of modeling groups. To facilitate comparison with the RCEMIP-I simulations, an identical configuration is used to that described by Wing et al. (2018) *except* for the analytic SST boundary condition. Participation in RCEMIP-I, while beneficial for comparison, is **not** required to participate in RCEMIP-II. The experimental design for the RCEMIP-II mock-Walker simulations is motivated by similar model configurations in recent studies of tropical convection and circulation (e.g., Silvers and Robinson, 2021; Lutsko and Cronin, 2018; Müller and Hohenegger, 2020; Lutsko and Cronin, 2023).

### 3.1  Basic model setup

A non-rotating aquaplanet model configuration is to be used (i.e., the Coriolis parameter, $f$, or Earth's angular velocity, $\Omega$, are set to zero), with no sea ice and no land. Recommended geophysical constants and parameters are provided in Table 1 of Wing et al. (2018), following the convention of the Aqua-Planet Experiment (APE; http://climate.ncas.ac.uk/ape/design.html).

Models which participated in RCEMIP-I should ideally use the same model version and configuration as RCEMIP-I for their RCEMIP-II mock-Walker simulations, to ensure that the SST boundary condition is the only thing that is different. If this is not possible due to model development in the intervening years, the `RCE_large300` simulation from RCEMIP-I should be repeated if possible with the new version of the model, to represent a reference point for comparison.

The RCEMIP-II simulations are to be initialized by the same sounding that was used to initialize a model's RCEMIP-I `RCE_large` simulation at the corresponding mean SST. That is, they should be initialized by a horizontally averaged equilibrium sounding from the corresponding `RCE_small` simulation.

We welcome participation in RCEMIP-II from models that did not participate in RCEMIP-I. Such models should at minimum complete the `RCE_small` simulations described in Wing et al. (2018) to derive a sounding from which to initialize their RCEMIP-II simulations. We encourage but do not require that they complete the RCEMIP-I `RCE_large` simulations, to serve as a reference.

## 3.2 Domain configuration

The domain configuration follows the `RCE_large` set-up described in Wing et al. (2018), for CRMs, GCMs, or GCRMs, which is reviewed here. SCM models are not eligible to participate in RCEMIP-II. While it is likely prohibitively computationally expensive for models at LES resolutions (200 m grid spacing was used in RCEMIP-I) to perform simulations on the RCEMIP-II domain, they are welcome to do so if the necessary computational resources are available.

### 3.2.1 CRMs

CRMs (models with explicit convection run on a limited-area planar domain) are to employ a three-dimensional domain with doubly periodic lateral boundary conditions. A horizontal grid spacing of 3 km is to be used with an elongated channel geometry of ∼6000 km in the long ($x$) direction and ∼400 km in the short ($y$) direction. The domain should be as close to 6000 km long as possible given the numerical limitations of a given model. The vertical grid is to be at least 74 vertical levels, a model top no lower than 33 km, and a sponge layer in the top model layers to damp gravity waves. Table 3 in Wing et al. (2018) provides the recommended vertical grid. The simulations are to be performed for at least 200 days, or longer if needed to reach equilibrium.

### 3.2.2 GCMs

GCMs (models with parameterized convection run on the global sphere) should employ whichever dynamical core and grid are standard for each model and the horizontal resolution, vertical coordinate, and grid of their CMIP6 configuration. We have chosen not to constrain the GCMs to a common horizontal and vertical grid because the physical parameterizations are sensitive to particular configurations. The simulations are to be performed for at least 1000 days, but should be run longer if needed to reach equilibrium.

### 3.2.3 GCRMs

GCRMs (models with explicit convection run on a sphere) should ideally be run with the same grid spacing as CRMs (3 km) and the same domain size as GCMs (the real Earth radius, $R_E$). However, a reduced Earth radius may be used to reduce the computational expense. A radius of $R_E/3.336$ will yield the correct wavelength of the SST pattern (see below). The simulations are to be performed for at least 200 days, or longer if needed to reach equilibrium.

### 3.3 Surface boundary condition

Surface enthalpy fluxes are to be calculated interactively from the resolved surface wind speed and air-sea enthalpy disequilibrium. If allowed by a model's surface layer formulation, a minimum wind speed of 1 ms$^{-1}$ should be enforced. The lower boundary represents the thermodynamic state of a sea surface that is fixed in time but varies spatially according to a prescribed sinusoidal temperature pattern. We desire to keep the mean SST and the SST gradient (change in SST per unit distance) as consistent as possible between the Cartesian CRM and spherical GCM configurations.

#### 3.3.1 CRMs

For CRMs on the Cartesian channel domain described above,

$$SST(x) = \langle SST \rangle - \frac{\Delta SST}{2} cos\left(\frac{2\pi x}{L_x}\right),\tag{1}$$

where $\langle SST \rangle$ is the mean SST, $\Delta SST$ is the difference between the maximum and minimum SST, $x$ is the horizontal position along the long axis, and $L_x$ is the domain length. This sets the wavelength of the SST pattern, $\lambda$, equal to $L_x$ and places the maximum SST at $L_x/2$, to maintain periodicity at the lateral boundaries.

There are several relevant SST-related quantities that may matter for the climate of mock-Walker simulations. From the perspective of the weak temperature gradient approximation, the absolute SST contrast ($\Delta SST$) is what ought to matter to the dynamics, and the maximum SST (where the deep convection occurs) will set the mean temperature in the free troposphere (Sobel and Bretherton, 2000; Bretherton and Sobel, 2002). Climate changes in response to SST warming are typically referenced to the domain mean SST $\langle SST \rangle$. The SST gradient (dSST/dx) might plausibly set horizontal flow speeds, based on Lindzen and Nigam (1987), and if that is the case, the SST Laplacian ought to matter for vertical motion and precipitation. Due to code numerics optimization and limitations, the CRMs participating in RCEMIP use slightly different domain lengths, $L_x$. This means that it is not possible to keep all of these parameters fixed across the simulations. We have elected to fix $\Delta SST$ across models but to allow $\lambda = L_x$ for each particular model, after testing the alternative options. With slightly different $L_x$, the SST gradient will vary slightly across models. Over the range of $L_x$ in the RCEMIP-I simulations, the resulting SST gradients differ from that with $L_x = 6000$ km by less than 3% (except for one model that used $L_x = 6480$ km, which has an SST gradient that is 7.4% smaller than that with $L_x = 6000$ km). While not ideal, this situation is more desirable than the alternative, which is to set $\lambda = 6000$ km regardless of the domain length. A fixed $\lambda$ would lead to slightly different $\langle SST \rangle$ across the models, a discontinuous SST distribution at the boundaries (since $\lambda \neq L_x$), and, most critically, the projection of the prescribed SST forcing onto all scales. The first two are minor issues, but while the vast majority of the power would be in the desired domain-scale wave, $\lambda \neq L_x$ introduces substantial noise at higher wavenumbers (contributing to the majority of the variance of dSST$^2$/dx$^2$; not shown). A third option would be to adjust the value of $\Delta SST$ to maintain the same SST gradient, but this would cause problematic differences in $\Delta SST$ as well as the maximum SST. Therefore, we elect to use Equation 1 to define the SST in the CRMs. Keeping $\Delta SST$, maximum SST, and $\langle SST \rangle$ consistent is the most elegant and simplest option to implement. It maintains the quantities that plausibly matter most for dynamics, precipitation, and clouds based on weak

temperature gradient arguments and avoids non-physical artifacts. We acknowledge that the SST gradients may differ slightly across models if their $L_x \neq 6000$ km, which could make it difficult to disentangle the relative contributions of $L_x$ difference from other aspects of model physics and numerics to intermodel differences. While we cannot rule out the differences in $L_x$ as a source of intermodel spread, we anticipate that it will be a small effect based on the test simulations we performed.

### 3.3.2 GCMs

For GCMs on a sphere using the observed radius of Earth,

$$SST(\phi) = \langle SST \rangle + \frac{\Delta SST}{2} cos \left( \frac{360°\phi}{\lambda} \right), \tag{2}$$

where $\langle SST \rangle$ is the mean SST, $\Delta SST$ is the difference between the maximum and minimum SST, $\phi$ is latitude in degrees, and $\lambda = 54°$ yields a wavelength of 6004.53 km (when centered on the equator), to approximately match the CRM configuration. Note that since these simulations are non-rotating there is no dynamical difference between a wave in latitude and a wave in 260 longitude. The wave in latitude proposed here maintains a consistent distance between each peak that is comparable to that in the CRM domain.

Since the simulations are non-rotating and the Cartesian CRM domain is doubly periodic, the $x-$ and $-y$ dimensions are interchangeable and the CRM domain should be conceptualized as being infinitely repeated in both dimensions (Figure 1c). This means that, other than the sphericity, it is analogous to a GCM set-up with a meridional circulation forced by a zonally 265 uniform meridionally varying SST (Figure 1d), which has also previously been used to study convective aggregation (e.g., Müller and Hohenegger, 2020). Setting the GCM SST pattern with zonal bands of warm and cool SSTs, following Equation 2, ensures the closest possible match to the CRM domain and avoids any irregularities that could arise from an SST pattern that decreases in width as the poles are approached, as would be the case for a zonally varying SST. Since one of the core principles of RCEMIP is to be able to compare limited area CRMs and GCMs, we chose this configuration, in which the warm SSTs are 270 confined in only one direction in both model types.

### 3.3.3 GCRMs

For GCRMs on a sphere with reduced Earth radius of $R_E/n$, where $R_E$ is the observed radius of Earth,

$$SST(\phi) = \langle SST \rangle + \frac{\Delta SST}{2} cos \left( \frac{360°\phi}{\lambda} \right), \tag{3}$$

where $\langle SST \rangle$ is the mean SST, $\Delta SST$ is the difference between the maximum SST and the minimum SST, and $\phi$ is latitude 275 in degrees. For $n = \pi R_E/6000 km$, which yields a radius of $R_E/n \approx R_E/3.336$, $\lambda = 180°$ corresponds to distance of 6000 km, to match the CRM configuration. If a smaller Earth radius of $R_E/4$ is used, as was used by some GCRMs in RCEMIP-I, $\lambda = 180°$ corresponds to a distance of approximately 5000 km. Smaller Earth radii than this are not recommended.

**Table 1.** RCEMIP-II Experiments

| Required Experiments | $\langle SST \rangle$ | $\Delta SST$ |
|---|---|---|
| MW_295dT1p25 | 295 K | 1.25 K |
| MW_300dT0p625 | 300 K | 0.625 K |
| MW_300dT1p25 | 300 K | 1.25 K |
| MW_300dT2p5 | 300 K | 2.5 K |
| MW_305dT1p25 | 305 K | 1.25 K |

| Optional Experiments | $\langle SST \rangle$ | $\Delta SST$ |
|---|---|---|
| MW_295dT0p625 | 295 K | 0.625 K |
| MW_295dT2p5 | 295 K | 2.5 K |
| MW_305dT0p625 | 305 K | 0.625 K |
| MW_305dT2p5 | 305 K | 2.5 K |

### 3.4 Radiative processes

The shortwave and longwave radiative heating rates are to be calculated interactively from the modeled state using a radiative transfer model. Trace gases are to be fixed and spatially uniform, according to Table 2 in Wing et al. (2018). The ozone profile is an analytic approximation of the horizontally uniform equatorial profile derived from the Aqua-Planet Experiment ozone climatology, given by Equation (1) in Wing et al. (2018). Aerosol effects are to be ignored. The incoming solar radiation is to be spatially uniform and constant in time; there is to be no diurnal nor seasonal cycle and every model grid point should receive the same incident radiation. Following Wing et al. (2018), a reduced solar constant of 551.58 W m$^{-2}$, a fixed zenith angle of 42.04°, and a fixed surface albedo of 0.07 should be used. The values of zenith angle and surface albedo are equal to the Equator to 20° and global average insolation-weighted values, respectively (Cronin, 2014).

### 3.5 Required simulations

The five required RCEMIP-II experiments are listed in Table 1, as are several optional experiments. Figure 1 shows the SST pattern in each experiment. In selecting the suite of $\Delta SST$ values, we took inspiration from, but did not attempt to exactly reproduce, observed SST gradients in the equatorial Pacific and Atlantic. Based on the 1950-2022 HadISST (Rayner et al., 2023) climatological SSTs, averaged between 5.5°S and 5.5°N, there is a difference between the warmest and coldest SSTs of ∼4 K over a distance of 14,344 km in the Pacific (between longitudes of 140.5° to -90.5°) and ∼2 K over a distance of 6,783 km in the Atlantic (between longitudes of -50.5° to 10.5°). In both basins, the climatological mean SST is ∼300 K. We desire to use an identical domain configuration as in RCEMIP-I, so that the simulations can be compared. This sets $\lambda = 6000$ km, to accommodate a full wavelength with the warmest SSTs at the center of the domain, doubly periodic lateral boundary

conditions, and the the `RCE_large` CRM domain length of $L_x \sim 6000$ km. Thus, a difference between the warmest and coldest SSTs of $\Delta SST$ = 0.83-0.89 K over 3000 km (half a wavelength) would be comparable to the observed SST gradients.

We considered three different values of $\langle SST \rangle$ (295, 300, and 305 K) and nine different values of $\Delta SST$ (0.625, 0.75, 1.0, 1.25, 1.5, 2.0, 2.5, 3.0, 5.0 K). Our central value of $\langle SST \rangle$ = 300 K is consistent with current observed mean SSTs in the equatorial Pacific and Atlantic. For the three values of $\Delta SST$ at $\langle SST \rangle$ = 300 K (Table 1, Figure 1), we selected values that resulted in distinctly different weak, moderate, and strong SST gradients, balancing similarity to observations with choices that have distinct spatial structures of convection and circulation. Our control experiment (`MW_300dT1p25`) thus has an SST gradient that is $\sim$45% stronger than observed. `MW_300dT0p65` and `MW_300dT2p5` consider SST gradients that are half and twice as strong as the control; `MW_300dT0p65` has an SST gradient that is $\sim$37% weaker than observed while `MW_300dT2p5` has an SST gradient that is $\sim$190% stronger than observed.

The `MW_295dT1p25` and `MW_305dT1p25` simulations, along with `MW_300dT1p25`, will reveal the effect of uniform warming on the large-scale circulation, convective aggregation, and other features of the simulated climate. This set of simulations can be compared to the corresponding `RCE_large` simulations with the same mean SSTs from RCEMIP-I, to evaluate the impact of the SST gradient and forced circulation on both the model mean state and response to warming. One slight caveat is that, assuming the weak temperature gradient condition, the domain mean free tropospheric temperature in the RCEMIP-II simulations will be set not by the the mean SST, but by the warmest SST (where deep convection is expected to occur). In the RCEMIP-I simulations, the SST is uniform. Thus, there will be a slight difference between the expected mean free tropospheric temperatures in the RCEMIP-II simulations and the RCEMIP-I simulations at the same mean SST.

The set of required RCEMIP-II simulations will also be used to study the response to warming with the SST gradient held constant, decreased, and increased, respectively. Changing $\Delta SST$ under mean warming to amplify or dampen the SST pattern will change the strength of the large-scale circulation and the forced component of aggregation, whereas when $\Delta SST$ is kept fixed, only changes in unforced aggregation will occur. This will facilitate improved understanding of the modulation of climate and hydrological sensitivity by convective aggregation, by having $\Delta SST$ as an external parameter that controls the forced component of aggregation. Varying $\Delta SST$ under mean warming will also permit us to assess how different models represent the pattern effect on climate feedbacks, which has not previously been examined in a model ensemble that includes models with explicit convection. RCEMIP-II will facilitate investigating the response of tropospheric stability and clouds to different patterns of warming.

## 4 Results

We performed test simulations using the System for Atmospheric Modeling (SAM, a CRM), version 6.11.2 (Khairoutdinov and Emanuel, 2013) and the Community Atmosphere Model (CAM, a GCM), version 6 (Danabasoglu et al., 2020). Similar to the paper describing the RCEMIP-I protocol (Wing et al., 2018), we show here sample results from some of these test simulations with SAM and CAM to motivate our choice of required simulations for the RCEMIP-II protocol and as an example of what those simulations might look like; this is not intended as a comprehensive comparison.

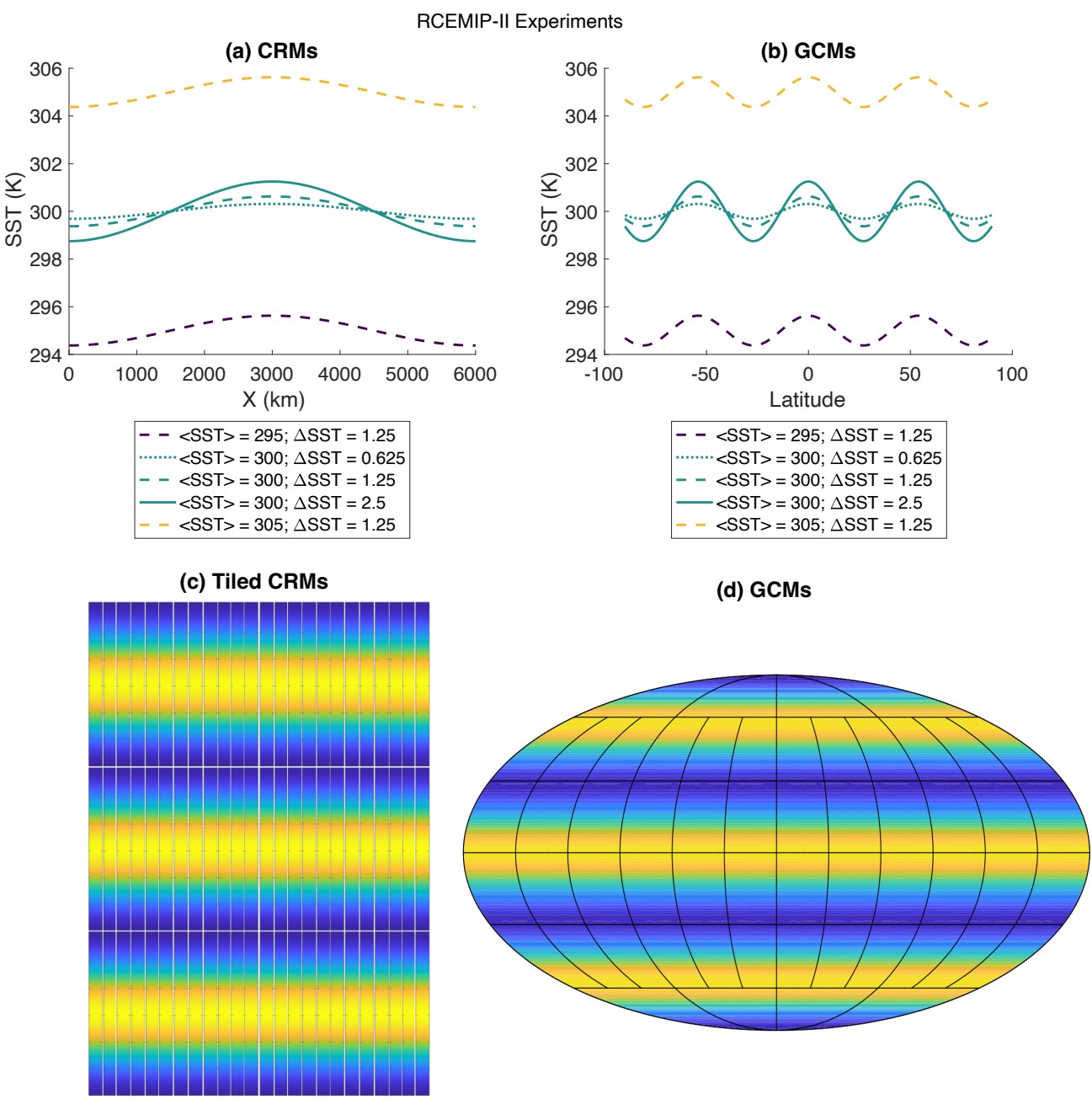

**Figure 1.** SST pattern for the five required RCEMIP-II experiments, for (a) a CRM with domain length $L_x = \lambda = 6000$ km; and (b) a GCM with real Earth radius and $\lambda = 54°$ following Equations 1-2. Panel (c) depicts the doubly periodic CRM domain as rotated and tiled 3 x 24 times, to conceptually match the GCM domain in panel (d), where yellow shading indicates warm SSTs.

The SAM simulations shown in Figures 2, 7, and 10 use Equation 1 to set the SST pattern as required for RCEMIP-II, with $L_x = \lambda = 6144$ km ($L_x = 6000$ km is not possible due to numerical and code optimization limitations). The SAM simulations used to test the value of $\Delta SST$ in Figures 4 and 6, and used to test the domain size in Figures 11, 12, and 13, were performed before Equation 1 was set and thus instead used $SST(x) = \langle SST \rangle + \frac{\Delta SST}{2} cos\left(\frac{2\pi}{L_x}\left(x + \frac{L_x}{2}\right)\right)$ with $\lambda = 6000$ km and $L_x = 6144$ km. While the details of an individual simulation with $\lambda = 6000$ km versus $\lambda = 6144$ km vary, the conclusions regarding the dependence on $\Delta SST$ and domain size do not qualitatively depend on this difference. Other than the SST boundary condition, the SAM simulations are identical to their configuration for RCEMIP-I.

The CAM simulations shown all use Equation 2 with $\lambda = 54°$, as required by RCEMIP-II. The CAM simulations use the version 6 physics package as in RCEMIP-I (Wing et al., 2020a; Reed et al., 2021) but the model code has been updated to improve some flaws in the cloud microphysics and ice nucleation (Zhu et al., 2022). Comprehensive climate simulations with the updated version of the Community Earth System Model, version 2 (CESM2) result in a lower equilibrium climate sensitivity, a more realistic Last Glacial Maximum, and a weaker shortwave cloud feedback than the version of CESM2 submitted to CMIP6 (Zhu et al., 2022), from which the version of CAM6 used in RCEMIP-I was drawn (Reed et al., 2021). Therefore, we rerun the RCEMIP-I simulations with this updated version of CAM6 as well for comparison.

## 4.1 Convective Structure and Evolution

Figures 2 - 3 show the spatial structure in the `MW_300dT1p25` simulation. Deep convection and precipitation are generally absent from the regions of coldest SSTs. In SAM, where there is one peak in SST at the center of the domain, the precipitation and low values of outgoing longwave radiation (OLR) associated with cold cloud tops are located in the region of the warmest SSTs and their periphery. There are numerous convective systems within the envelope of warm SST, but there are also convective systems on the flanks of the warm SSTs. If one averages over a longer time scale, however, the strongest mean rising motion occurs over the warmest SSTs where the moist static energy (MSE) is largest, in association with an overturning circulation that spans the length of the domain, as shown in the bottom row of Figure 2. The overturning circulation has both shallow and deep components. This is consistent with prior work, which has shown that mock-Walker simulations may develop stacked overturning circulations (Grabowski et al., 2000; Yano et al., 2002; Larson and Hartmann, 2003; Liu and Moncrieff, 2008; Silvers and Robinson, 2021) due to interactions between radiation and detrained condensate and water vapor (Nuijens and Emanuel, 2018; Sokol and Hartmann, 2022; Lutsko and Cronin, 2023).

In CAM, an SST pattern with three peaks in latitude is used to match the spatial scale of the SST pattern used by SAM. Accordingly, the CAM results show three latitudinal bands of precipitation. However within these latitudinal bands, there are certain longitudes with more precipitation than others. This varies with time (only a single time is shown in the figure) and is similar to the zonal contraction of convection in the simulations of Müller and Hohenegger (2020). While the long-term average circulation in CAM exhibits ascent over the peaks of SST there is substantial spatial and short-term variability among the CAM simulations. The longitudinal variations in circulation, including both ascent and descent within regions of peak SST, are apparent in the precipitation, pressure velocity, and OLR (Figure 3).

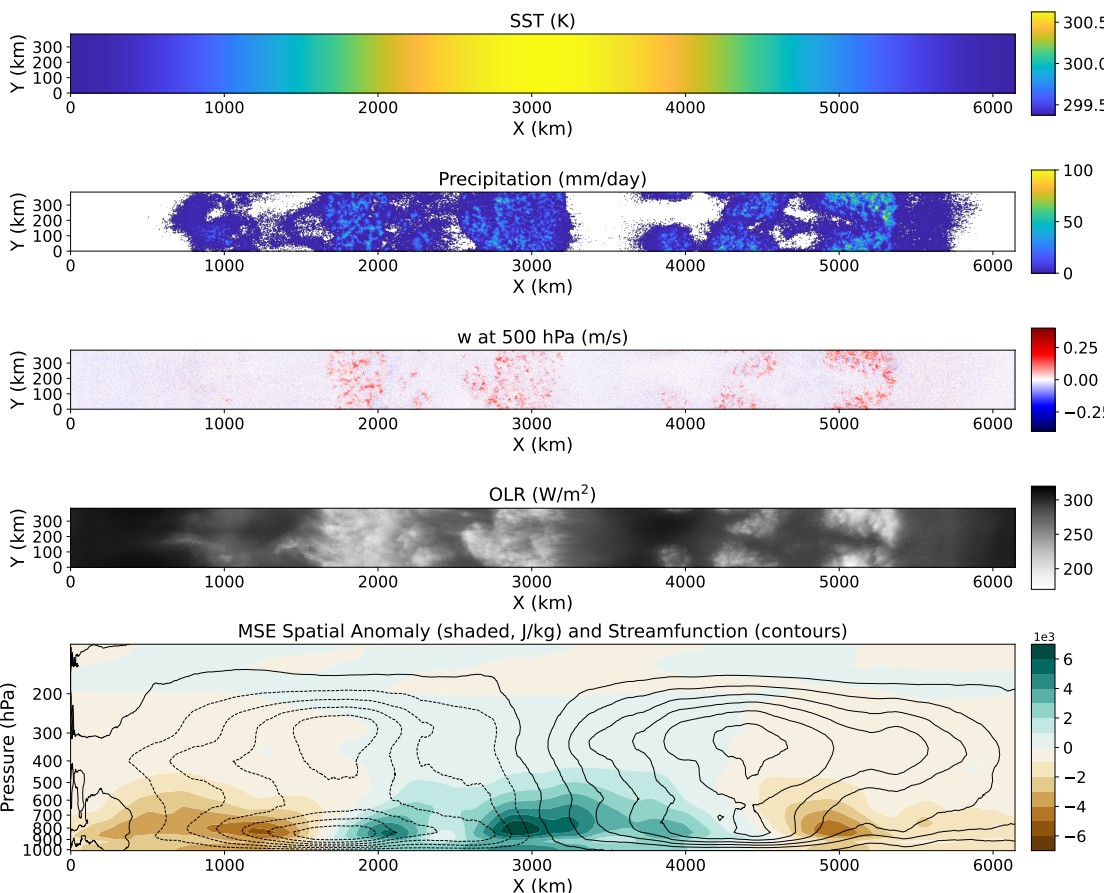

**Figure 2.** Daily-mean SST (K), precipitation (mm d$^{-1}$), vertical motion at 500 hPa (m s$^{-1}$), and outgoing longwave radiation (OLR; W m$^{-2}$) at day 150, and moist static energy (MSE; J kg$^{-1}$) anomaly from the spatial mean and streamfunction averaged over days 100 - 200 in the `MW_300dT1p25` simulation with SAM. Streamfunction is calculated by integrating y-averaged $\rho w$ across the x-axis of the simulation, with contours every 500 kg m$^{-1}$ s$^{-1}$.

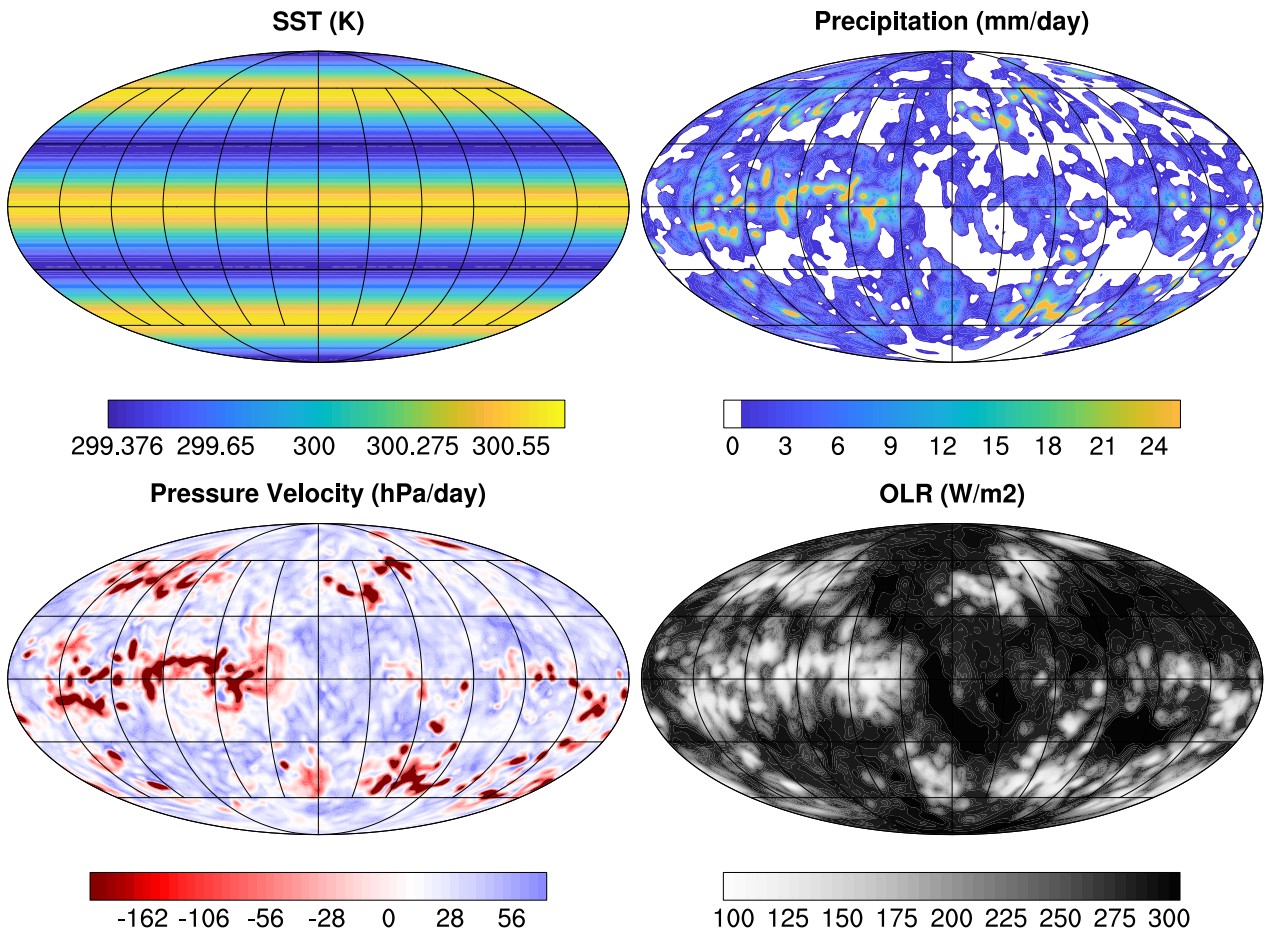

**Figure 3.** SST (K), precipitation (mm day$^{-1}$), vertical pressure velocity on the 500 hPa pressure level (hPa d$^{-1}$), and the outgoing longwave radiation (W m$^{-2}$), averaged over 1 day in the third year of the `MW_300dT1p25` simulation with CAM. Note that the colorbars are saturated for visualization purposes; the maximum and minimum values are 93.94 and 7.73 x 10$^{-7}$ mm day$^{-1}$ for precipitation, -1263.09 and 118.18 hPa d$^{-1}$ for vertical velocity, and 102.57 and 315.85 W m$^{-2}$ for OLR, respectively.

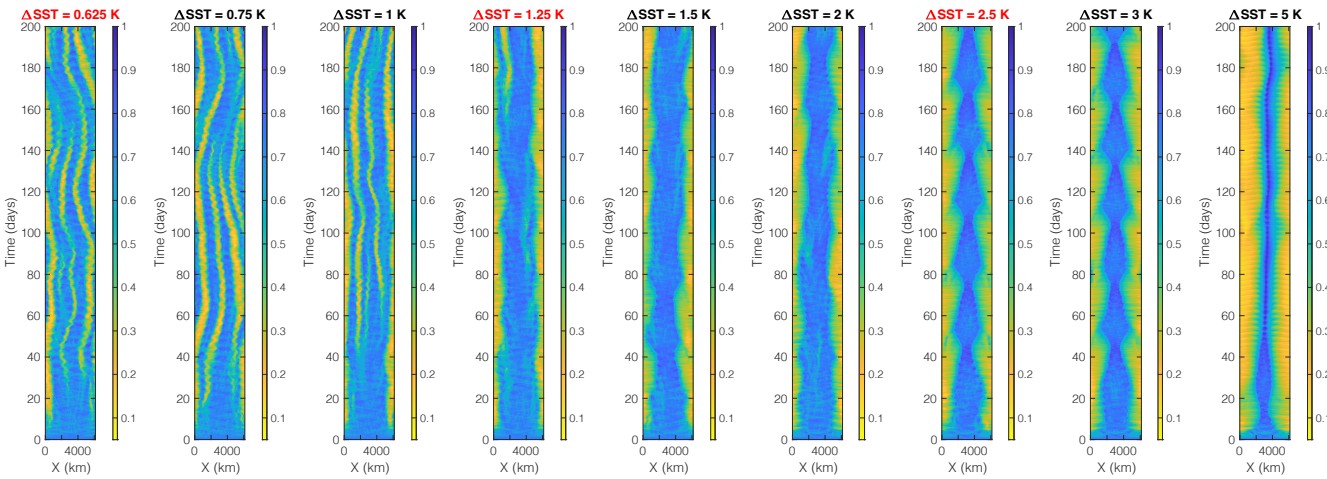

**Figure 4.** Hovmöller diagrams of y-averaged column relative humidity in the simulations with $\langle SST \rangle = 300$ and $\Delta SST = 0.625, 0.75, 1,$ 1.25, 1.5, 2, 2.5, 3, and 5 K, with SAM. The values of $\Delta SST$ that are required for RCEMIP-II are highlighted in red. Note that these SAM simulations, used to test the value of $\Delta SST$, were performed with $SST(x) = \langle SST \rangle + \frac{\Delta SST}{2} cos \left( \frac{2\pi}{L_x} \left( x + \frac{L_x}{2} \right) \right)$ with $\lambda = 6000$ km and $L_x = 6144$ km, rather than Equation 1 and $\lambda = L_x$ as required for RCEMIP-II.

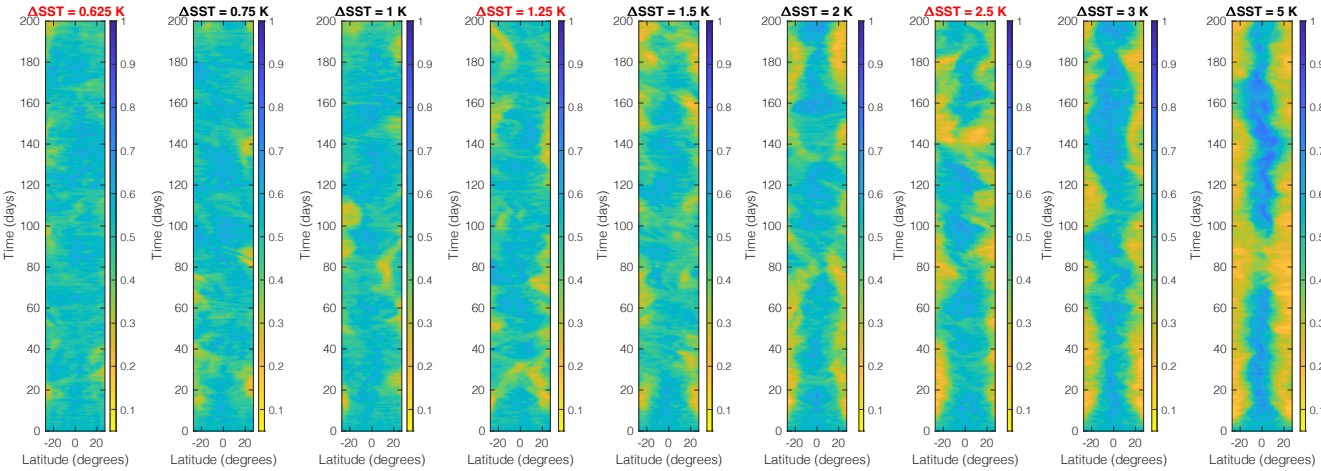

**Figure 5.** Hovmöller diagrams of column relative humidity averaged over $4°$ of longitude over the first 200 days of the simulations with $\langle SST \rangle = 300$ and $\Delta SST = 0.625, 0.75, 1, 1.25, 1.5, 2, 2.5, 3,$ and 5 K, with CAM. The values of $\Delta SST$ that are required for RCEMIP-II are highlighted in red.

Figures 4-5 show the temporal evolution of column relative humidity in a set of simulations with $\langle SST \rangle$ = 300 K and different values of $\Delta SST$. Column relative humidity is defined as the ratio of the water vapor path to the saturated water vapor path. Localization of convection associated with the development of dry and moist patches emerges within the first twenty days of the simulations, though it takes more than fifty days for the simulations to reach a statistical equilibrium. The localized dry patches are more spatiotemporally intermittent in CAM (Figure 5) than SAM (Figure 4), especially at the lower values of $\langle SST \rangle$. However, note that Figure 5 depicts the average across a random 4° of longitude, to average over a region roughly consistent with the SAM domain. Other moist and dry patches are found at other longitudes (and latitudes) in the SAM simulation, though Hovmöller diagrams of the zonal mean column relative humidity are qualitatively similar to Figure 5, albeit smoother. While only the first 200 days are plotted, the CAM simulation is run for over 1000 days. The structure of column relative humidity is remarkably consistent to what is shown in Figure 5 over the full simulation. The only exception is that in the `MW_300dT5`, the moist band near the equator breaks down around day 350, before re-emerging by day 450 (not shown). This could indicate the moist band moving to a different longitude, or it could suggest a temporary reorganization of atmospheric moisture content due to unpredictable convective interactions, as seen in similar simulations by Silvers and Robinson (2021) and speculated to be a result of parameterized convection.

The differences across different values of $\Delta SST$ is discussed below in Section 4.2, but here we note that all simulations display rich variability in the structure of the dry and moist regions (and convection). The SAM simulations in particular (Figure 4) depict disturbances that propagate along the long axis of the domain in both directions, with propagation speeds consistent with convectively coupled gravity waves. There are instances in which a dry band emerges from within a broader moist band, splitting the latter into two moist band, as well as instances in which two moist bands merge together. There is also lower frequency variability; for example, the location of the moist and dry regions in the SAM simulations with $\Delta SST$ = 0.625 K and $\Delta SST$ = 0.75 K oscillate back and forth with time. There are also fluctuations in the size of the moist, convective region, which contracts and expands along the long axis of the domain with a period of $\sim$30-40 days. This is most prominent in the SAM simulations with $\Delta SST$ = 2.5 K and $\Delta SST$ = 3 K but is also seen to a lesser extent in the CAM simulation at those values of $\Delta SST$ as well as both the SAM and CAM simulations at other values of $\Delta SST$.

These low-frequency oscillations can have a substantial influence on domain mean quantities such as the top of atmosphere radiative fluxes (Figures 6 - 7). In the SAM simulations with strong SST gradients (Figure 6c), the OLR and net shortwave flux at the top of the atmosphere vary on $\sim$30 day time scales with amplitudes of $O(10)$ W m$^{-2}$. This behavior has also been seen in similar mock-Walker simulations with other CRMs, in which the amplitude of the fluctuations can be larger (tens of W m$^{-2}$) depending on the model, microphysics settings, and other parameters (Guy Dagan, Andrew Williams, Peter Hill, Nick Lutsko, personal communication). Amongst the test simulations examined here with SAM and CAM, it seems that the oscillations are less extreme and less regular in CAM than they are in SAM (Figure 6). It is not clear whether this is due to the coarser grid spacing and parameterization of convection in CAM or its larger domain and spherical geometry.

We do not attempt to explain the origins of these low-frequency oscillations here, nor their dependence on $\langle SST \rangle$ and $\Delta SST$. Their physical mechanisms and model dependence is a likely target of analysis across the full suite of RCEMIP-II simulations, once complete, and is thus beyond the scope of this protocol paper. In terms of defining the protocol, however,

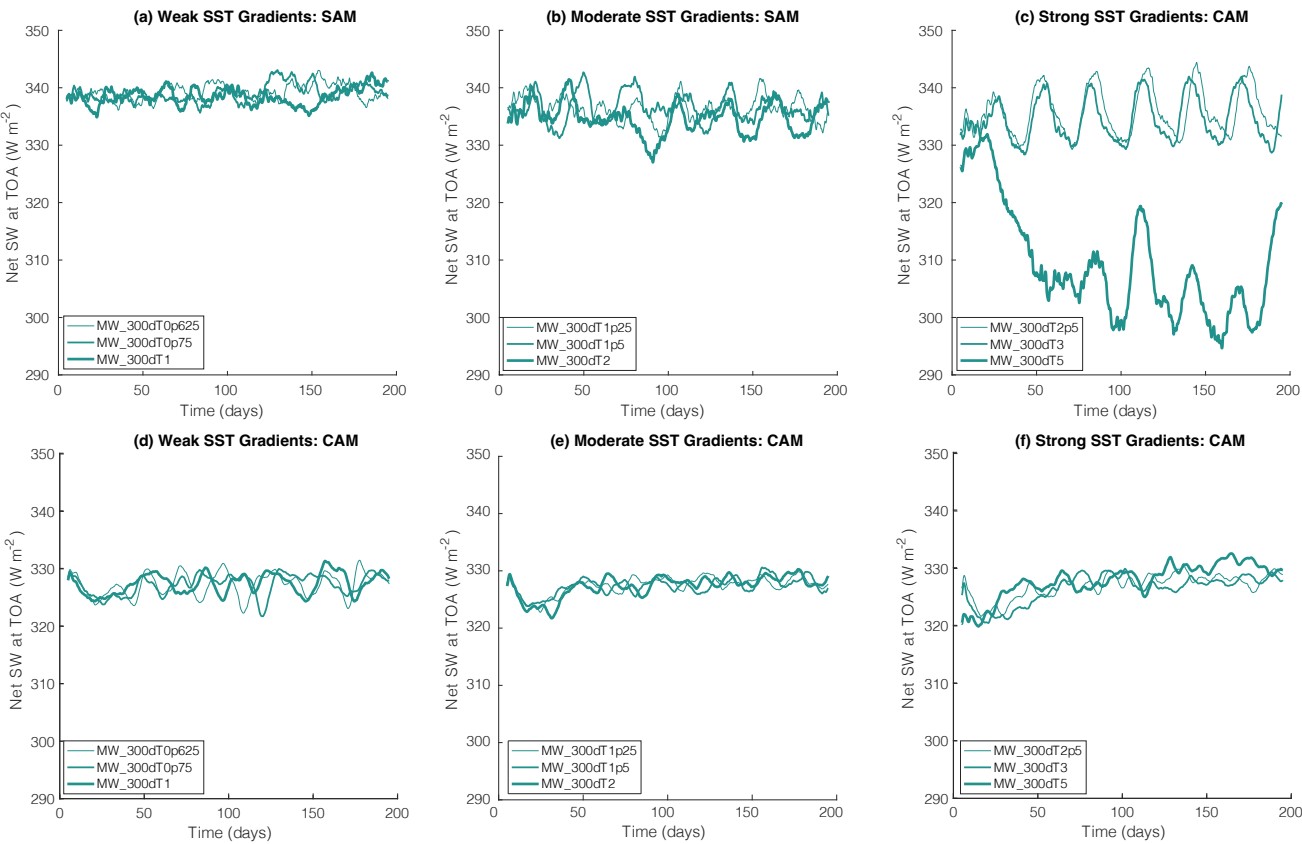

**Figure 6.** Domain mean net shortwave flux at the top of the atmosphere (W m$^{-2}$) in the simulations with (top row; panels a-c) SAM and (bottom row; panels d-f) CAM at $\langle SST \rangle$ = 300 K and (left column; panels a and d) weak ($\Delta SST$ = 0.625 K, 0.75 K, 1 K), (middle column; panels b and e) moderate ($\Delta SST$ = 1.25 K, 1.5 K, 2 K) and (right column; panels c and f) strong ($\Delta SST$ = 2.5 K, 3 K, 5 K) SST gradients. In each panel, a five-day running mean is shown with line thickness increasing with the value of $\Delta SST$. Note that these SAM simulations, used to test the value of $\Delta SST$, were performed with $SST(x) = \langle SST \rangle + \frac{\Delta SST}{2} cos\left(\frac{2\pi}{L_x}\left(x + \frac{L_x}{2}\right)\right)$ with $\lambda$ = 6000 km and $L_x$ = 6144 km, rather than Equation 1 and $\lambda = L_x$ as required for RCEMIP-II.

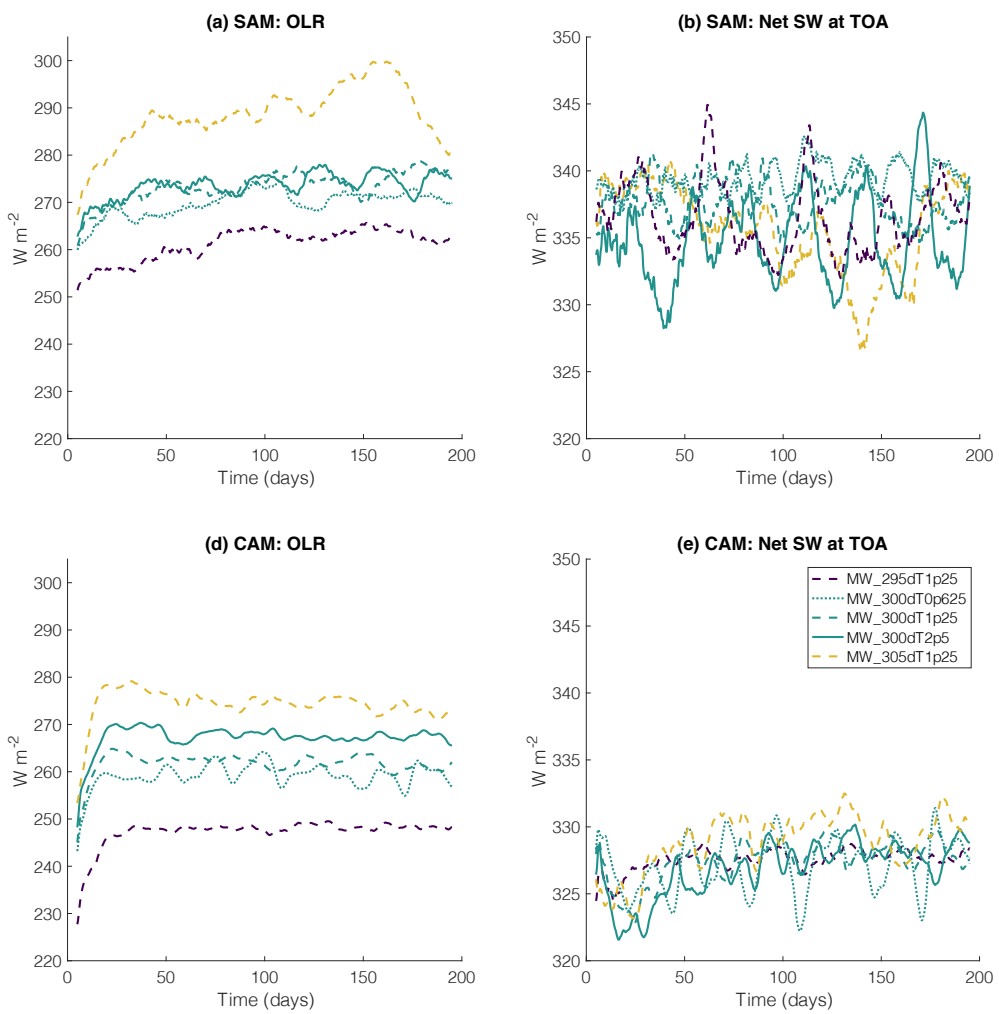

**Figure 7.** Domain mean (left column; panels a and c) outgoing longwave radiation (OLR; W m$^{-2}$) and (right column; panels b and d) net shortwave flux at the top of the atmosphere (W m$^{-2}$) in the simulations required for RCEMIP-II (`MW_295dT1p25`, `MW_300dT0p625`, `MW_300dT1p25`, `MW_300dT2p5`, and `MW_305dT1p25`) with (top row; panels a and b) SAM and (bottom row; panels c and d) CAM over the first 200 days of the simulation. A five-day running mean has been applied to all data, note the different axes in the left and right column.

the presence of these low-frequency oscillations suggests that longer simulations than in RCEMIP-I (which consisted of 100 day simulations for the CRMs) will be needed to achieve robust equilibrium statistics in the RCEMIP-II mock-Walker configuration. Our test simulations indicate that simulations of at least 200 days in length for the CRMs, in which the first 75 days are excluded from the equilibrium averaging period, should be sufficient to average over multiple cycles of the ~30 day oscillations and achieve robust statistics, while still being computationally feasible. If an individual model develops oscillations of even lower frequency, longer simulations with that model may be needed.

## 4.2 Sensitivity to $\Delta SST$

Figures 4-5 indicate that the moist, convective region narrows and the range of CRH values in the domain increases as the SST gradient increases. SAM systematically exhibits a larger range of CRH values than CAM. The simulations tend to divide into three groups, with weak ($\Delta SST$ = 0.625, 0.75, and 1 K), moderate ($\Delta SST$ = 1.25, 1.5, and 2 K), and strong ($\Delta SST$ = 2.5, 3, and 5 K) SST gradients, though these groups are more distinct for SAM than CAM. In the group with weak SST gradients ($\Delta SST$ = 0.625, 0.75, and 1 K) there are alternating moist and dry regions across the entire domain in the SAM simulation (Figure 4), reminiscent of the self-aggregation of convection seen in RCE simulations with the same geometry but uniform SST (Wing et al., 2020a). This suggests that $\Delta SST \leq 1$ K provides such a weak SST gradient that the spatial structure of convection is not influenced by the SST pattern. The circulations set up by the intrinsic self-aggregation are thus stronger than those forced by the SST gradient. This is also apparent in Figure 8, in which the the domain-scale overturning circulation is weak in the simulations with $\Delta SST \leq 1$ K. Compared to the `RCE_large` simulations with SAM from RCEMIP-I, in which $\Delta SST = 0$ K and self-aggregation generates multiple circulation cells across the domain, the streamfunction in the SAM simulations with $\Delta SST \leq 1$ K are weaker because the multiple aggregated regions have more variability in their spatial location and thus their circulations are damped in the time average.

In the group with moderate SST gradients ($\Delta SST$ = 1.25, 1.5, and 2 K), there is typically one moist, convecting region centered over the warmest SSTs with some variability in its spatial extent (Figures 4-5). In these simulations with SAM, the SST gradient has forced a domain-spanning overturning circulation (left column of Figure 8). In the group with strong SST gradients ($\Delta SST$ = 2.5, 3, and 5 K), stronger low-frequency oscillations emerge (particularly in SAM) and the convective region is increasingly narrow. In SAM, the overturning circulation and MSE anomalies are strongest in these simulations (left column of Figure 8). A tight connection between rising branches of the overturning circulations and positive MSE anomalies is apparent across all SAM simulations. In simulations with strong and moderate SST gradients, the positive MSE anomalies are generally co-located with the warmest SSTs at the center of the domain, though off-centered positive anomalies are found in a few simulations. The different circulation structures across different values of $\Delta$SST collapse if one instead considers a moisture-sorted circulation (Bretherton et al., 2005; Muller and Held, 2012). To compute the moisture-sorted circulation, CRH is smoothed to a grid scale of 60 km, $w$ and MSE are averaged over each percentile of CRH (in 1% bins), and $\rho w$ is integrated from the 0th to 100th percentile of CRH. All simulations exhibit a moisture-sorted circulation from dry to moist regions with both shallow and deep components (right column of Figure 8). The shallow overturning circulation has been shown to contribute to convective self-aggregation (Muller and Held, 2012; Muller and Bony, 2015; Coppin and Bony, 2015).

Though the differences are muted, the strength of the moisture-sorted circulation tends to be slightly weaker with larger $\Delta SST$ or larger $\langle SST \rangle$ (not shown), which is consistent with Silvers et al. (2023). This is in contrast to the increase in the strength of the physical circulation in SAM with $\Delta SST$ in the left column of Figure 8. The time-mean physical circulation represents the circulation forced by the prescribed SST gradient, while the moisture-sorted circulation emphasizes transport between dry and moist regions, regardless of how they are organized in space.

As in the SAM simulations, the MSE anomalies grow stronger and the regions of vertical ascent are centered over the peaks of SST for all of the CAM simulations (Figure 9). All of the mock-Walker simulations have stronger overturning circulations than the `RCE_large` simulation from RCEMIP-I (Figure 9, top panel). Pairs of opposite-signed streamfunction centered over the SST maxima are most well-defined in the simulations with the strongest SST gradients, but systematic variations of the streamfunction as $\Delta SST$ changes are less clear in CAM, relative to SAM. We think this is in part due to the large regions of descending motion that can form within a region of peak SST (as in Figure 3). It is also possible that the overturning circulations from each of the three peak SST regions interact with each other to either strengthen or weaken the mean circulation. It should be noted that the stacked circulations seen in SAM and previous literature with CRMs, do not appear in the CAM circulations.

The low-frequency oscillations discussed above are quite regular in the SAM simulations with $\Delta SST$ = 2.5 and 3 K, but are more extreme and more irregular with $\Delta SST$ = 5 K. The `MW_300dT5` SAM simulation exhibits oscillations in the domain-mean net shortwave flux at the top of the atmosphere of up to $\sim$20 W m$^{-2}$ but the magnitude varies (Figure 6c). This in part motivates us to avoid the use of such strong SST gradients in our selection of the required RCEMIP-II simulations. In the CAM simulations, the low-frequency oscillations are less prevalent and have less dependence on $\Delta SST$ (Figure 6). Another reason to exclude simulations with $\Delta SST$ = 5 K is that they, along with other simulations that have very warm SSTs in at least part of the domain (such as those with $\langle SST \rangle$ = 305 K) take longer to reach equilibrium, at least in SAM (Figures 6-7). `MW_305dT1p25` is a required simulation, but at least in this particular model, simulations longer than 200 days may be required for it to reach equilibrium.

Since the response of clouds to warming is one of the themes of RCEMIP, we also examine the sensitivity of cloud amount to the prescribed $\Delta SST$. Figure 10 shows cloud fraction profiles for $\Delta SST$ = 0.625 K, 1.25 K, and 2.5 K, as well as the corresponding `RCE_large` simulations from RCEMIP-I, with uniform SST ($\Delta SST$ = 0 K) for reference. The dependencies of cloud fraction on $\Delta SST$ generally hold across the entire set of $\Delta SST$ values examined, not just those shown in Figure 10. In SAM, increasing $\Delta SST$ increases the low cloud fraction due to the presence of colder SSTs and stronger subsidence in the cold region. A more nuanced examination of clouds using ISCCP histograms demonstrated that the SAM mock-Walker simulations have more numerous, thicker, deeper low clouds (perhaps indicative of stratocumulus) than their RCEMIP-I counterparts with uniform SST (Stauffer, 2023). Increasing $\Delta SST$ also tends to decrease the high cloud fraction. In CAM, $\Delta SST$ has little effect on low clouds, except perhaps in the simulations at $\langle SST \rangle$ = 295 K (Figure 10d). As in SAM, increasing $\Delta SST$ decreases the high cloud fraction in CAM (Figure 10), except in those simulations with $\langle SST \rangle$ = 305 K (Figure 10f), which have much larger high cloud fractions ($\sim$ 0.8) that tend to increase with increasing $\Delta SST$ (not shown). CAM6 is known to generate a lot of thin, high clouds over warm SSTs in RCEMIP-I (Reed et al., 2021), which is seen also here in RCEMIP-II.

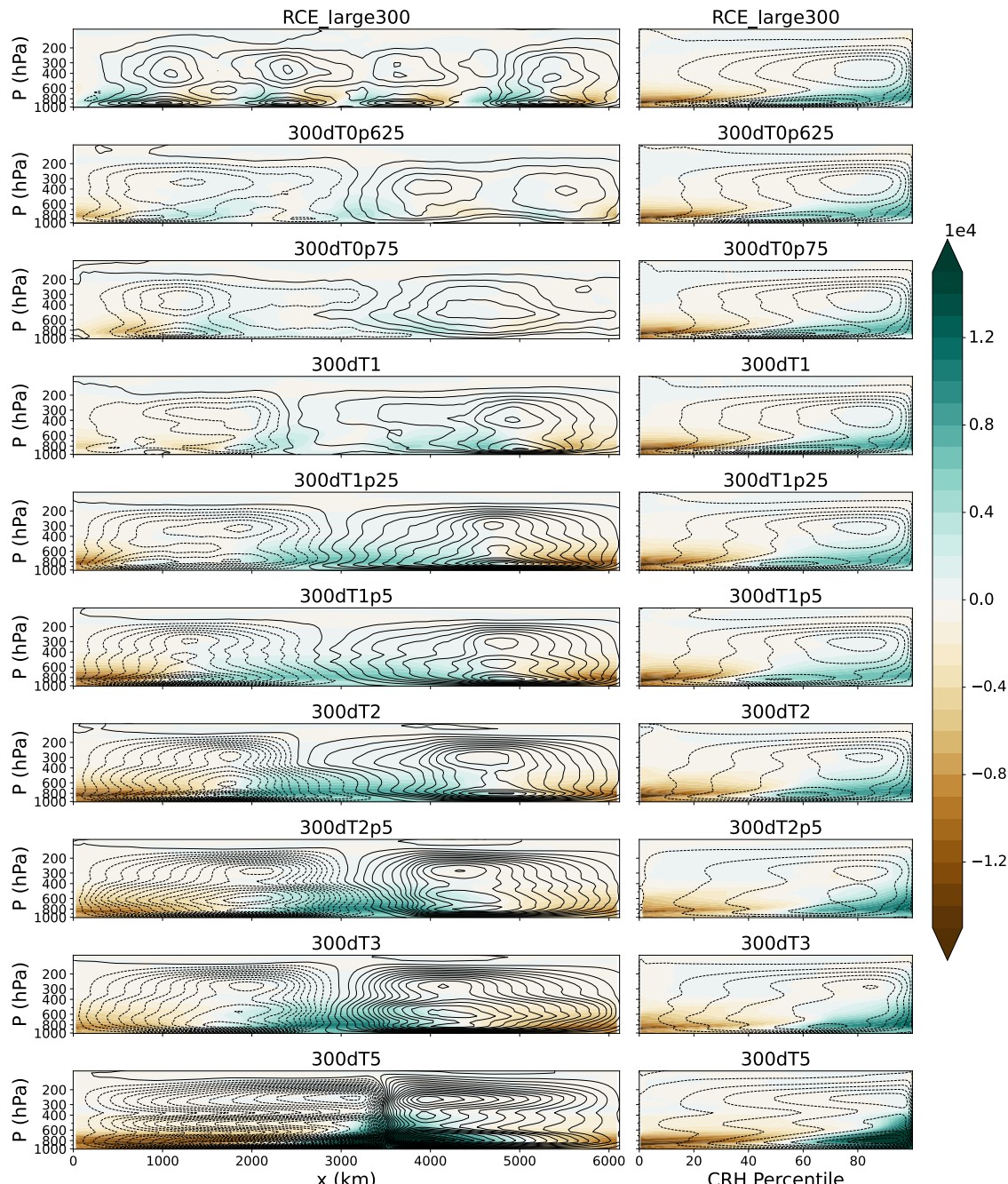

**Figure 8.** Streamfunction and moist static energy (MSE; shading; J kg$^{-1}$) in the SAM simulations. MSE is plotted as anomalies from the spatial mean. The left column shows the streamfunction in physical space, as in Figure 2, with contours every 500 kg m$^{-1}$ s$^{-1}$. The right column shows the streamfunction in moisture space, with contours every 0.05 kg*%ile m$^{-2}$ s$^{-1}$. The top row shows the `RCE_large300` simulations from RCEMIP-I with uniform SST ($\Delta SST = 0$ K), averaged over days 75-100, while the other rows show mock-Walker simulations with various $\Delta SST$, averaged over days 100-200. Note that these SAM simulations, used to test the value of $\Delta SST$, were performed with $SST(x) = \langle SST \rangle + \frac{\Delta SST}{2} cos\left(\frac{2\pi}{L_x}\left(x + \frac{L_x}{2}\right)\right)$ with $\lambda$ = 6000 km and $L_x$ = 6144 km, rather than Equation 1 and $\lambda = L_x$ as required for RCEMIP-II.

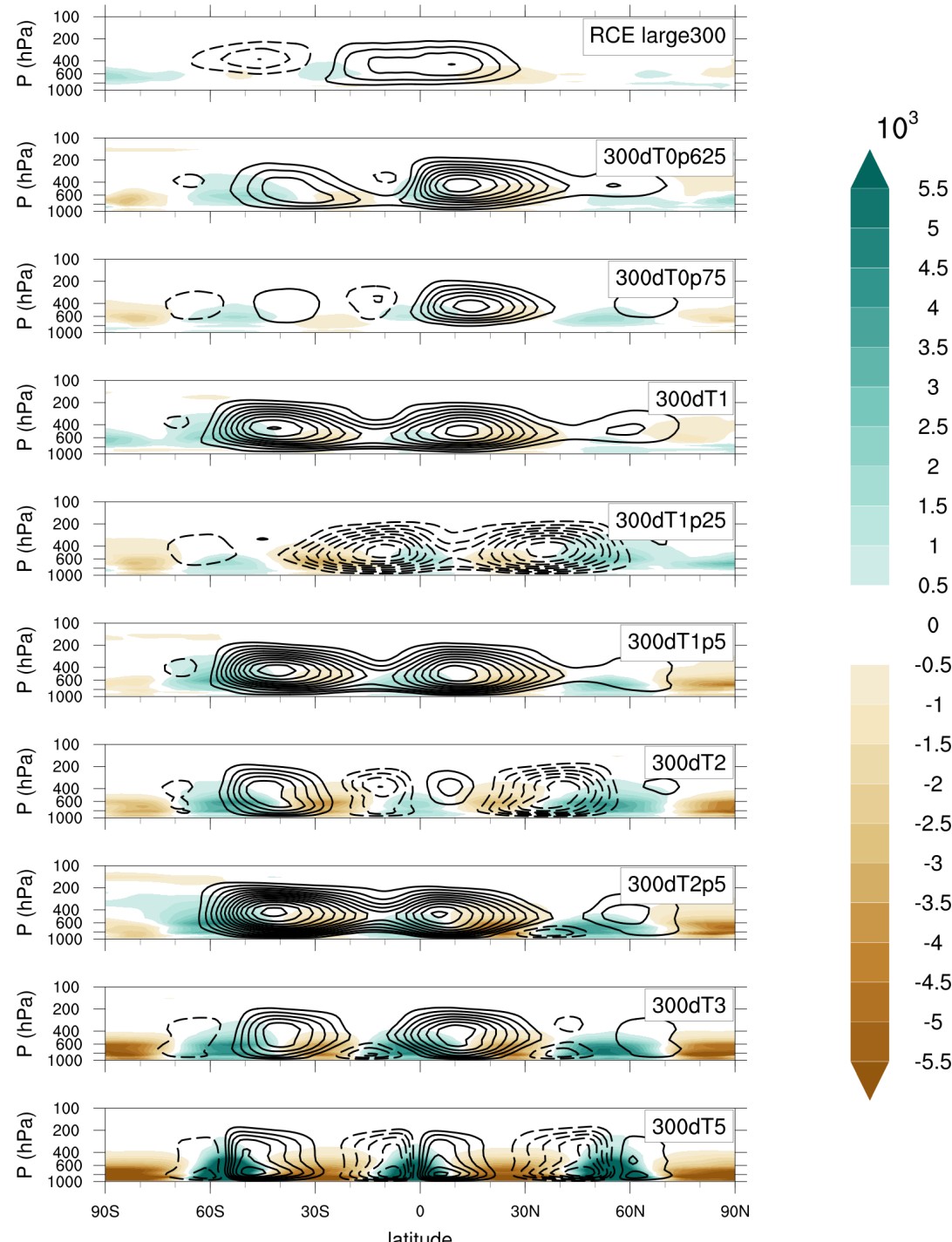

**Figure 9.** Streamfunction and moist static energy (MSE; shading; J kg$^{-1}$) in the CAM simulations. MSE is plotted as anomalies from the spatial mean. The streamfunction is plotted with contours every $4 \times 10^{10}$ kg s$^{-1}$, following typical convention with global models. Positive and negative streamfunction values are represented by solid and dashed lines, with the zero contour omitted. The top row shows the RCE_large300 simulations from RCEMIP-I with uniform SST ($\Delta SST = 0$ K). The remaining rows show mock-Walker simulations with various $\Delta SST$. All panels show data averaged over 25 days.

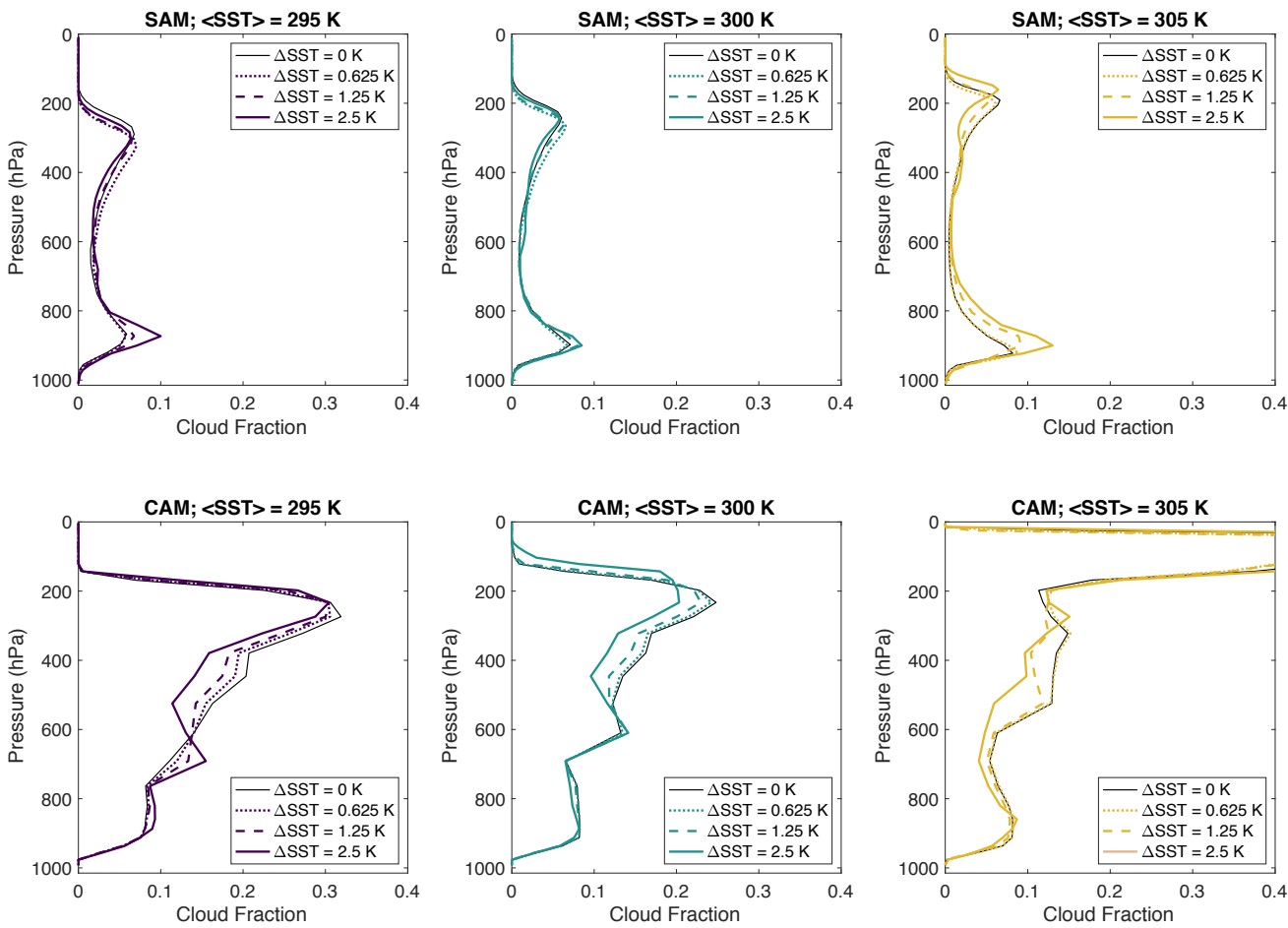

**Figure 10.** Domain- and time- mean profiles of cloud fraction, excluding the first 100 days of simulation, for simulations with $\langle SST \rangle$ of (left) 295, (middle) 300, and (right) 305 K. The corresponding `RCE_large` simulations from RCEMIP-I, with uniform SST ($\Delta SST = 0$ K) are plotted in the solid black line. The line style indicates the value of $\Delta SST$ (see legend).

The response of clouds to warming is similar across all values of $\Delta SST$ (Figure 10). The upward shift and decrease in high cloud fraction is robust and consistent with the response in the RCEMIP-I simulations with uniform SST (Stauffer and Wing, 2022). There is some suggestion of an increase in low cloud fraction with warming, particularly in SAM. A detailed calculation of the cloud feedback using cloud radiative kernels and its decomposition into contributions from changes in cloud amount, altitude, and optical depth indicated that the SAM mock-Walker simulations have cloud feedbacks of the same sign but larger

magnitudes than their RCEMIP-I counterparts (Stauffer, 2023).

### 4.3  Sensitivity to Domain Size

The test simulations with SAM described above follow the domain configuration for CRMs specified in Section 3.2.1. The domain is the same as that used for the SAM `RCE_large` simulations in RCEMIP-I; there are 2048 x 128 grid points in the horizontal with a grid spacing of 3 km, resulting in a horizontal domain that is 6144 km x 384 km, or an aspect ratio of

16:1. There are 74 vertical levels following Table 3 of Wing et al. (2018). The experimental design for RCEMIP-II calls for the domain configuration to be the same as in RCEMIP-I, to allow for a clean comparison as well as keep the simulations computationally inexpensive. However, the results could be sensitive to both the long and short dimensions of the domain. For example, prior mock-Walker studies have found that the structure of precipitation and vertical profiles of cloudiness are sensitive to the long dimension of the domain (Silvers and Robinson, 2021; Bretherton et al., 2006). In order to provide some

context for the results in our chosen domain configuration, we performed a few test simulations with SAM with different domain sizes: (1) A `wide` domain, which is twice as big in the short dimension and has an aspect ratio of 8:1 (2048 x 256 grid points; 6144 km x 768 km); (2) A `long` domain, which is twice as big in the long dimension and has an aspect ratio of 32:1 (4096 x 128 grid points; 12,288 km x 384 km); and (3) A long and wide (`longwide`) domain, which is twice as big in both dimensions so has the same 16:1 aspect ratio as the control simulation (4096 x 256 grid points; 12,288 km x 768

485    km). All simulations used to test the domain size utilize $SST(x) = \langle SST \rangle + \frac{\Delta SST}{2} cos \left( \frac{2\pi}{L_x} \left( x + \frac{L_x}{2} \right) \right)$ with $\lambda = 6000$ km and $L_x = 6144$ km, rather than Equation 1 and $\lambda = L_x$ as required for RCEMIP-II. The `long` and `longwide` simulations include two wavelengths of the SST pattern, but maintain the same SST gradient as the control domain. We conduct these domain size sensitivity tests for the `MW_300dT1p25` and `MW_300dT2p5` simulations (`longwide` is only performed for the `MW_300dT1p25` simulation).

The `MW_300dT1p25wide` and `MW_300dT2p5wide` simulations exhibit similar spatial structures of convection to that in their narrower counterparts (`MW_300dT1p25` and `MW_300dT2p5`), as seen in the CRH field in Figures 11-12, as well as the precipitation, OLR, and vertical velocity fields (not shown). The wide simulations are able to fit more convective cells across their short dimension, but otherwise the convection is similarly confined to the warmest SSTs and exhibits a similar temporal evolution (Figures 11a,b and 12a,b). This indicates that, at least in this situation with doubly periodic boundary

conditions, mock-Walker simulations with different domain widths but the same domain length exhibit qualitatively similar behaviors. When comparing the `MW_300dT1p25longwide` simulation to the `MW_300dT1p25long` simulation, the spatial structures are also generally similar, though it takes longer for the `MW_300dT1p25longwide` to evolve to match the `MW_300dT1p25long` simulation.

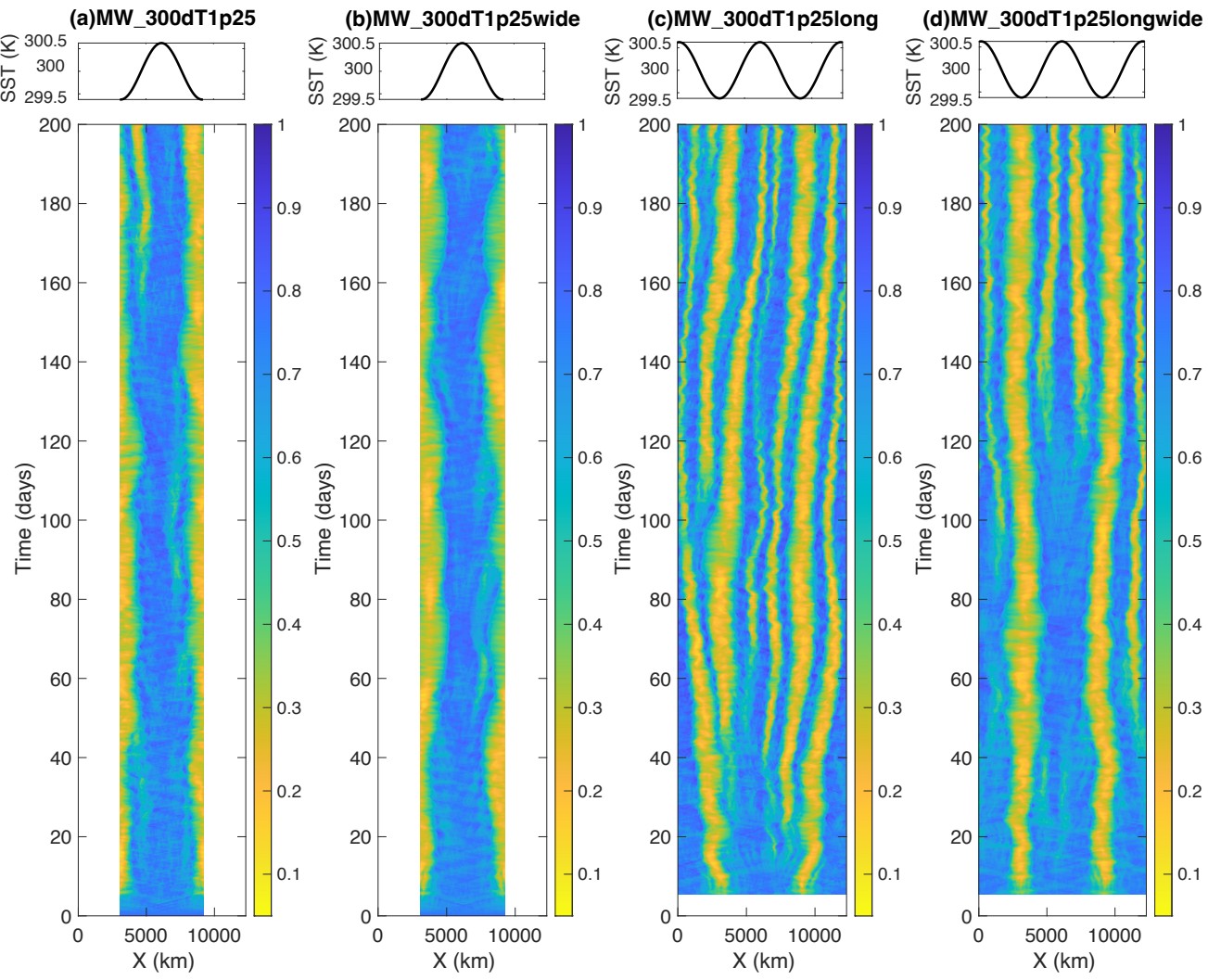

**Figure 11.** Hovmöller diagrams of y-averaged column relative humidity in the `MW_300dT1p25` simulation with SAM for (a) the standard domain size, (b) a domain that is twice as wide in the y-dimension, (c) a domain that is twice as long in the x-dimension, and (d) a domain that is twice as wide and twice as long. Note that these SAM simulations, used to test the domain size, were performed with $SST(x) = \langle SST \rangle + \frac{\Delta SST}{2} cos\left(\frac{2\pi}{L_x}\left(x + \frac{L_x}{2}\right)\right)$ with $\lambda$ = 6000 km and $L_x$ = 6144 km, rather than Equation 1 and $\lambda = L_x$ as required for RCEMIP-II. The top panel shows the SST pattern. Note that the output for the first five simulation days in panels (c) and (d) was corrupted and thus is not plotted.

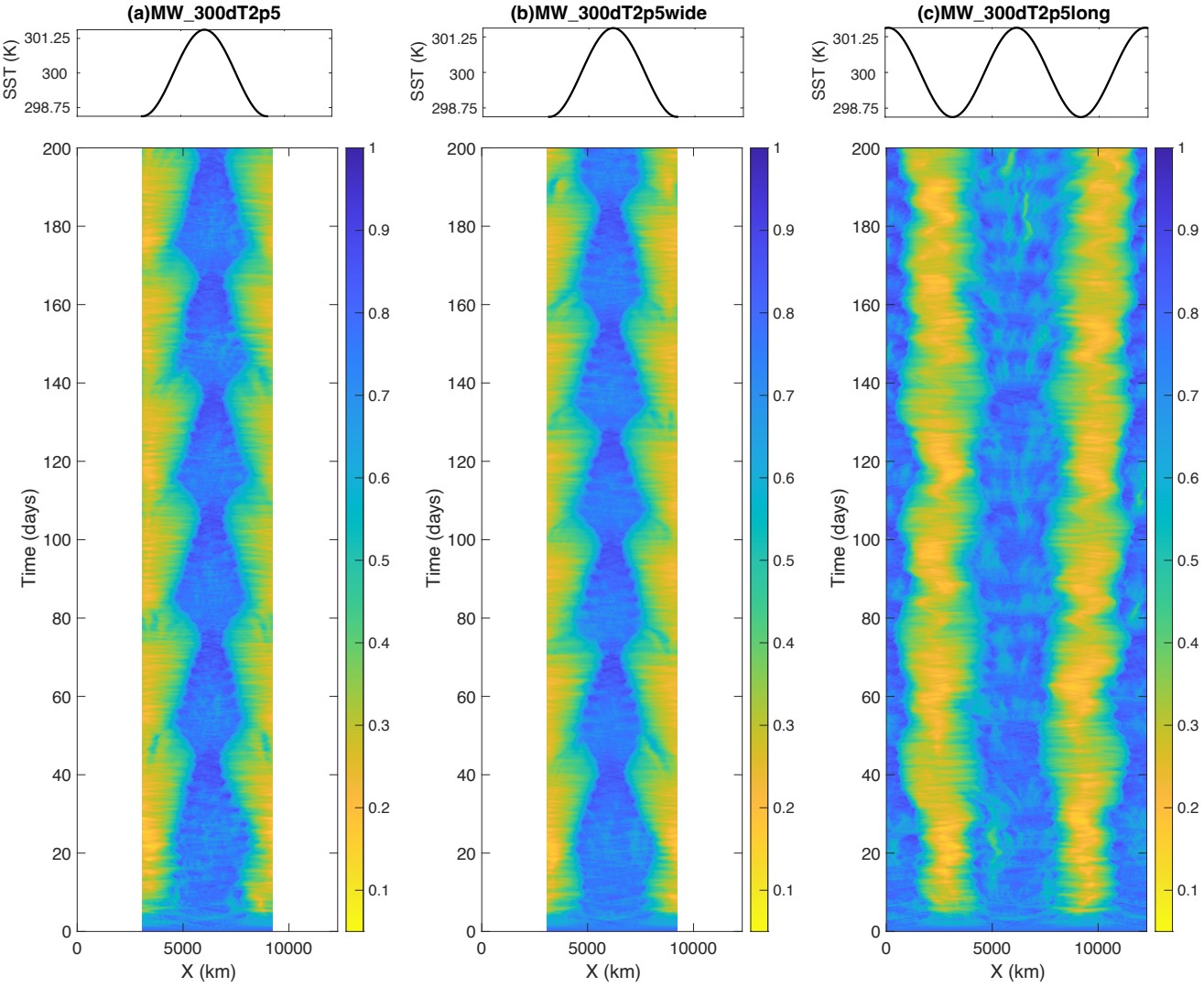

**Figure 12.** Hovmöller diagrams of y-averaged column relative humidity in the `MW_300dT2p5` simulation with SAM for (a) the standard domain size, (b) a domain that is twice as wide in the y-dimension, (c) a domain that is twice as long in the x-dimension. Note that these SAM simulations, used to test the domain size, were performed with $SST(x) = \langle SST \rangle + \frac{\Delta SST}{2} cos\left(\frac{2\pi}{L_x}\left(x + \frac{L_x}{2}\right)\right)$ with $\lambda = 6000$ km and $L_x = 6144$ km, rather than Equation 1 and $\lambda = L_x$ as required for RCEMIP-II. The top panel shows the SST pattern.

The `MW_300dT2p5long` simulation exhibits similar spatial structures and temporal evolution of convection to that in its
shorter counterpart `MW_300dT2p5`, except the pattern is repeated twice (Figure 12a,c). That is, the moist convecting regions
are confined to the regions with warmest SSTs at the middle and left and right edges of the domain. There are a few differences:
the central warm region is not quite as uniformly moist in the `long` simulation, and the low-frequency oscillations in the size
of the moist region are less notable. However, to first order, the `MW_300dT2p5long` simulation appears to be an extension
of the `MW_300dT2p5` simulation for another wavelength.

The `MW_300dT1p25long` simulation, however, has quite different behavior than its shorter counterpart (`MW_300dT1p25`).
Initially the moist convecting regions are confined to the regions with warmest SSTs, but unlike the control `MW_300dT1p25`
simulation, this is not maintained in the `long` simulation (Figure 11a,c). Instead, convection and precipitation begin to also
develop on the periphery of the warm SSTs and eventually, narrower bands emerge with no preference to occur over the regions
of the domain where the SST is maximized. In this regard, the `MW_300dT1p25long` simulation is similar to the simulations
with weaker SST gradients (i.e., `MW_300dT0p625`) and reminiscent of self-aggregation of convection with uniform SST. This
domain length dependence is similar to that found in the mock-Walker simulations of Silvers and Robinson (2021), though in
their longer simulations they utilized the same $\Delta SST$ between the center of the domain and the edges, resulting in a weaker
SST gradient. Thus, it is somewhat surprising that the breakdown of convection into narrow bands spanning the whole domain
occurs in our long simulation, in which the SST gradient is unchanged. It suggests that $\Delta SST = 1.25$ K is near the threshold
for when the circulation forced by the SST gradient is strong enough to overcome those of intrinsic self-aggregation, and when
provided by a longer domain with multiple wavelengths of SST, intrinsic self-aggregation outweighs the forced circulation.
In the `MW_300dT1p25` simulation with the control domain size, there is also some suggestion that convection is beginning
to break down into narrower bands right at the end of the simulation (Figure 11a), so if given enough time that simulation
may also evolve to be dominated by intrinsic self-aggregation. In the the `MW_300dT1p25longwide` simulation, the moist
convecting regions are confined to the regions with warmest SSTs for the first ∼100 days of the simulation, but they eventually
breakdown into narrow bands spanning the whole domain as in `MW_300dT1p25long` (Figure 11c,d).

Despite the differences in convective structures discussed above, the simulations with different domain width generally have
similar mean statistics (Figure 13). The `wide` simulations at both $\Delta SST = 1.25$ K and $\Delta SST = 2.5$ K have similar mean values
and temporal variability as their narrower counterparts. The simulations with different domain length exhibit more differences.
The `MW_300dT2p5long` simulation has lower amplitude and somewhat less regular, lower amplitude low-frequency oscil-
lations than its shorter counterpart, though roughly similar mean values. The precipitable water in the `MW_300dT1p25long`
and `MW_300dT1p25longwide` simulations diverges to much lower values than their shorter counterparts, as higher frac-
tions of the domain are occupied by dry areas. Consistent with the lower precipitable water, the OLR also tends to be higher in
these simulations.

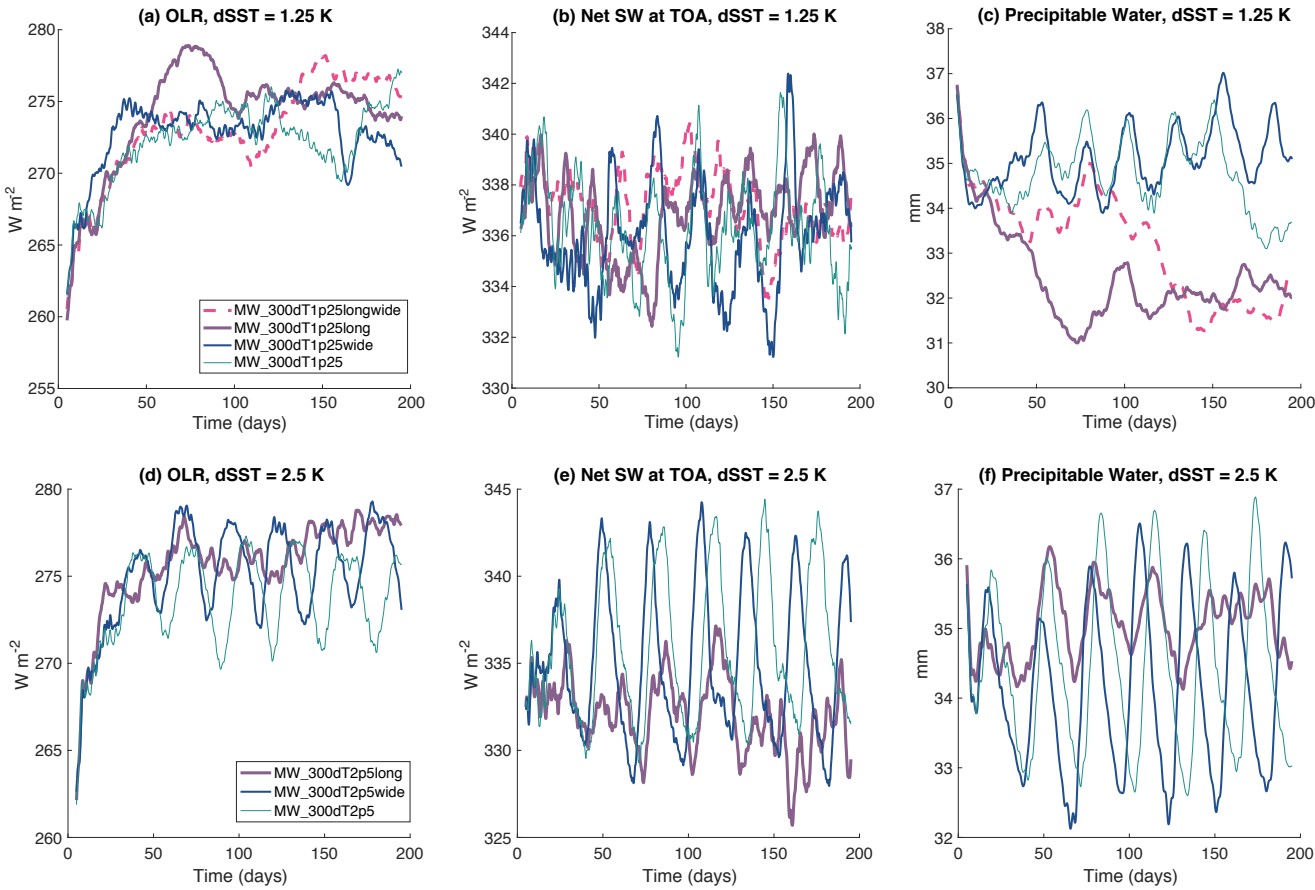

**Figure 13.** Domain mean (left column; panels a and d) outgoing longwave radiation (OLR; W m$^{-2}$), (middle column; panels b and e) net shortwave flux at the top of the atmosphere (W m$^{-2}$), and (right column; panels c and f) precipitable water (mm) in simulations with SAM with $\langle SST \rangle$ = 300 K and (top row; panels a-c) $\Delta SST$ = 1.25 K and (bottom row; panels d-f) $\Delta SST$ = 2.5 at different domain sizes. The domain sizes include the standard domain size (thin teal solid line), a domain that is twice as wide (medium blue solid line), a domain that is twice as long (thick purple solid line), and a domain that is both twice as wide and twice as long (thick pink dash-dotted line). A five-day running mean has been applied to all data. Note that these SAM simulations, used to test the domain size, were performed with $SST(x) = \langle SST \rangle + \frac{\Delta SST}{2} cos \left( \frac{2\pi}{L_x} \left( x + \frac{L_x}{2} \right) \right)$ with $\lambda$ = 6000 km and $L_x$ = 6144 km, rather than Equation 1 and $\lambda = L_x$ as required for RCEMIP-II.

## 5  Discussion and Next Steps

In summary, a mock-Walker configuration with a specified, sinusoidal SST boundary condition is proposed for RCEMIP-II. This is intended to provide a partial constraint on the structure of convection while still allowing for rich interactions between convection, clouds, and circulations. RCEMIP-II will build on the success of RCEMIP-I to facilitate deeper understanding of cloud-circulation coupling and convective aggregation and its role in climate.

After testing several equations for the SST boundary condition and nine different values of $\Delta SST$ with both SAM, a CRM, and CAM, a GCM, we selected five experiments to be required for RCEMIP-II (Table 1, Figure 1) with the experimental design as described in Section 3. The protocol is designed to allow for comparison between CRMs on a limited-area Cartesian domain and GCMs on the global sphere. The selection of the required $\Delta SST$ values balances similarity to observed SST gradients in the equatorial Pacific and Atlantic with choices that, in our test simulations, exhibit different spatial structures of convection and circulation. As was the philosophy for RCEMIP-I, the set of five required simulations for RCEMIP-II was carefully selected to facilitate addressing the scientific objectives while minimizing the computational expense (and thus maximizing participation). The SST boundary condition is the only difference in the set-up between RCEMIP-II and RCEMIP-I. While this enables comparison between RCEMIP-I and RCEMIP-II, we welcome participation in RCEMIP-II from models that did not participate in RCEMIP-I.

In addition to providing example results for the five required RCEMIP-II simulations, we also test the sensitivity to $\Delta SST$ and the domain geometry. As $\Delta SST$ increases, the moist, convective region narrows and becomes increasingly confined to the warmest SSTs. With small values of $\Delta SST$, there are alternating moist and dry region across the entire domain, reminiscent of the self-aggregation of convection seen in RCE simulations with uniform SST. At larger values of $\Delta SST$, low-frequency oscillations in the extent of the moist, convective region and associated variability in domain mean quantities emerge. While low cloud fraction tends to increase and high cloud fraction tends to decrease with increasing $\Delta SST$, the response of clouds to warming is similar across all values of $\Delta SST$. Simulations with different domain sizes generally have similar mean statistics. The convective structures are relatively insensitive to the width (short horizontal dimension) of the CRM domain, but can be sensitive to the length (long horizontal dimension), depending on the value of $\Delta SST$.

The breakdown to self-aggregated convection that spans the entire domain with longer time or larger length simulations (Section 4.3) is a potential concern for the choice of $\Delta SST$ = 1.25 K as representing the moderate SST gradient regime in the set of required RCEMIP-II simulations. It is possible that in some models, the behavior with $\Delta SST$ = 1.25 K could look similar to that in $\Delta SST$ = 0.625 K, as the transition between regimes is likely model dependent. While we considered instead selecting $\Delta SST$ = 0.75, 1.5, and 3 K as the set of required simulations, since $\Delta SST$ = 1.5 K might be more securely in the moderate SST gradient range than $\Delta SST$ = 1.25 K, differences across models are likely larger than those due to a 0.25K $\Delta SST$ difference. Our chosen $\Delta SST$ values of 0.625, 1.25, and 2.5 K result in SST gradients that are more symmetrically weaker and stronger than observed than the alternatives. Furthermore, in RCEMIP-I, SAM tended to be one of the most strongly self-aggregated models (Wing et al., 2020a), which could indicate that the dominance of intrinsic self-aggregation over the SST-forced circulation could hold to higher $\Delta SST$ values in SAM than in other models. Indeed, there is no guarantee

that $\Delta SST = 0.625$ will result in a self-aggregation-like regime in models that have weaker self-aggregation tendencies. While intermodel differences in which regime each $\Delta SST$ value belongs to may complicate analysis, identifying where these regime transitions occur across the spectrum of models is a goal of RCEMIP-II. To aid in this, while only the top five simulations listed in Table 1 are required, we encourage participants to conduct simulations at additional $\Delta SST$ values near regime transitions if possible as well as with different $\langle SST \rangle$ values, as listed as optional simulations in the bottom of Table 1.

In addition to the objectives related to the simulated mean state, the response of clouds to warming, and the role of convective aggregation in climate described in Section 2, and the analysis plans described in Section 3.5, there are numerous other avenues of investigation that could be explored in the full RCEMIP-II ensemble. By including both models with explicit convection and those with parameterized convection, RCEMIP-II maintains the ability to determine how behavior depends on the representation of convection. Other possible lines of inquiry include determining the physical mechanisms leading to low-frequency oscillations and their dependence on $\Delta SST$, investigating the development of stacked overturning circulations, and investigating what controls the transition between weak, moderate, and strong SST gradient regimes.

While we selected mock-Walker simulations for RCEMIP-II, that does not preclude other types of experiments from being performed by individuals or small groups of models, or being led as full intercomparisons as an offshoot of RCEMIP. Indeed, other community efforts utilizing the RCEMIP set-up to investigate other questions have been proposed. One effort involves repeating the RCEMIP-I simulations with aerosol-cloud interactions (Lorian and Dagan, 2023), to compare the response of clouds to aerosol perturbations at equilibrium under a wide range of SST values (RCEMIP-ACI). Such simulations could also be performed using the RCEMIP-II protocol to investigate aerosol-cloud interactions in the presence of a forced circulation. There has also been some interest in performing RCE (uniform thermal forcing) simulations with global models with rotation (e.g. Silvers et al. (2024)), which would generate convectively-coupled equatorial waves and tropical cyclones. In this way, the RCEMIP-I and RCEMIP-II protocols are a foundation upon which auxiliary investigations with modified experimental designs can be built. A potential future phase III of RCEMIP could focus on adding a slab mixed layer and interactive SSTs to the RCEMIP-I and RCEMIP-II configurations. While adding substantial complexity, this would allow the convective aggregation and cloud-circulation coupling to be influenced by ocean-atmosphere interactions, which is one of the primary physical processes currently missing from the RCEMIP set-up.

One potential limitation of the mock-Walker RCEMIP-II simulations is that the test simulations, as well as prior work demonstrate the potential for complex behaviors. For example, the emergence of intrinsic unforced self-aggregation that can overwhelm the forced circulation, low-frequency variability, and stacked overturning circulations. While these are all targets of investigation for RCEMIP-II, it is likely that these behaviors will differ across models and thus may complicated interpretation. For this reason, we considered pairing the required mock-Walker simulations with fully interactive radiation with additional simulations with fixed, horizontally and vertically uniform radiative cooling. Prior work has suggested that this can suppress stacked-overturning cells (Grabowski et al., 2000; Wofsy and Kuang, 2012; Sokol and Hartmann, 2022; Lutsko and Cronin, 2023). Horizontally uniform radiative cooling has also been shown to suppress self-aggregation (Wing et al., 2017). However, there are numerous nuances regarding how to best implement this and the design of such simulations requires testing that is beyond the scope of this current paper. Performing both sets of simulations may also be prohibitively difficult for some models,

**Table 2.** List of models planning to participate in RCEMIP-II.

| Model Type | Model | |
|---|---|---|
| CRM | DALES | Dutch Atmospheric Large Eddy Simulation (Heus et al., 2010) |
| CRM | DAM | Das Atmosphaerisch Modell (Romps, 2008) |
| CRM | FV3 | GFDL-FV3 CRM (Zhou et al., 2019) |
| CRM | ICON | ICOsahedral Nonhydrostatic Model - Sapphire (Hohenegger et al., 2023) |
| CRM | MESO-NH | MESO-NH v5.6 (Lac et al., 2018) |
| CRM | UKMO-RA1-T | UK Met Office Idealized Model v11.0 (Stratton et al., 2018) |
| CRM | RAMS | Regional Atmospheric Modeling System (Cotton et al., 2003) |
| CRM | SAM-1MOM | System for Atmospheric Modeling, 1-moment microphysics (Khairoutdinov and Randall, 2003) |
| CRM | SAM-M2005 | System for Atmospheric Modeling, M2005 microphysics (Morrison et al., 2005) |
| CRM | SAM-P3ice | System for Atmospheric Modeling, P3ice microphysics (Morrison and Milbrandt, 2015; Gasparini et al., 2022) |
| CRM | SCALE | Scalable Computing for Advanced Library and Environment v5.2.5 (Nishizawa et al., 2015; Sato et al., 2015) |
| CRM | SCREAMv0 | Simple Cloud-Resolving E3SM Atmosphere Model (Caldwell et al., 2021; Bogenschutz et al., 2023) |
| CRM | SNAP | Simulating Nonhydrostatic Atmosphere on Planets (Li and Chen, 2019) |
| CRM | VVM | Vector Vorticity Model (Wu et al., 2019) |
| GCRM | ICON | ICOsahedral Nonhydrostatic Model - Sapphire (Hohenegger et al., 2023) |
| GCRM | NICAM | Non-hydrostatic Icosahedral Atmospheric Model v16.3 (Satoh et al., 2014) |
| GCRM | SAM | Global System for Atmospheric Modeling (Khairoutdinov et al., 2022) |
| GCM | CAM5 | Community Atmosphere Model version 5 (Neale et al., 2012) |
| GCM | CAM6 | Community Atmosphere Model version 6 (Danabasoglu et al., 2020) |
| GCM | CNRM-CM6 | CNRM-CM6-1 - Atmosphere component (Roehrig and coauthors, 2020; Voldoire et al., 2019) |
| GCM | EXOCUBED | ExoCubed (Chen and Li, 2024) |
| GCM | E3SM-MMFv2 | Super-parameterized Energy Exascale Earth System Model (Hannah et al., 2020) |
| GCM | E3SMv3 | Energy Exascale Earth System Model (E3SM Project, 2024) |
| GCM | FV3-AM4 | GFDL-FV3 with AM4 Physics (Zhao et al., 2018a, b) |
| GCM | IPSL-CM6 | LMDZ6A version (Hourdin and coauthors, 2020) |
| GCM | MIROC6 | Model for Interdisciplinary Research on Climate (Tatebe et al., 2019) |
| GCM | SP-CAM | Super-Parameterized Community Atmosphere Model (Randall et al., 2016) |
| GCM | UKMO-GA7.1 | Met Office Unified Model Global Atmosphere v7.1 (Walters et al., 2019) |

and we desire to maintain the ability for many groups to participate. Therefore, we do not request or provide a protocol for

fixed radiation simulations at this time, but will continue to investigate their feasibility and optimal design for inclusion in future work, in which such additional simulations would be a valuable complement to the required RCEMIP-II simulations described here.

Table 2 shows the list of models that intend to contribute to RCEMIP-II. It includes 18 models that participated in RCEMIP-I and 9 models that did not. This list may grow with participation from additional modeling groups and scientists across the

world. We also welcome multiple configurations of a given model (i.e., the same model with various microphysics schemes). Appendix A details the output specification for RCEMIP-II, which closely follows that of RCEMIP-I with a few changes and additions. The additions are requested to facilitate analysis that was not possible with the RCEMIP-I data, and are divided into required and optional output requests.

*Code availability.* Analysis scripts used to generate the figures are available in a Zenodo archive at https://doi.org/10.5281/zenodo.11043720

(Wing, 2024a). The version of the System for Atmospheric Modeling (Khairoutdinov and Randall, 2003) used here is publicly available

*Data availability.* Model configuration files and a subset of the model data needed to reproduce the figures is available in a Zenodo

archive at https://doi.org/10.5281/zenodo.10137266 (Wing and Silvers, 2023). We thank the German Climate Computing Center (DKRZ) for hosting the standardized RCEMIP-I data (Wing et al., 2020b), which is publicly available online at http://hdl.handle.net/21.14101/d4beee8e-6996-453e-bbd1-ff53b6874c0e. The standardized RCEMIP-II data will also be hosted and made publicly available in the DKRZ archive once all the RCEMIP-II simulations have been completed and submitted.

## Appendix A: Output Specification

The RCEMIP-II output request closely follows that of RCEMIP-I, as described by (Wing et al., 2018) and its corresponding corrigendum. We highlight here a few variables to pay careful attention to and a few changes and additions.

### A1 Output Variables

In all tables, the italicized variables are non-standard outputs, all others are standard CMIP6 output. Bolded variables are new compared to RCEMIP-I. The variables with a (-)[1] symbol are outputs for GCMs only. The output should be "CMOR-ized",

such that the output variable names and units are the same as in CMIP6, as listed in the below tables.

In addition to the listed variables, the horizontal coordinates, vertical coordinate, and time coordinate should also be output. The time coordinate should be in units of days since the beginning of the simulation. Output should be submitted on a $x$-$y$- (for CRMs) or latitude-longitude (for global models) grid. In the vertical, the variables should be on model levels and the necessary information to compute pressure on model levels should be provided. If your model does not employ pressure levels (i.e., it

uses height levels or a type of hybrid level), please also output the domain- and time-mean values of pressure on your model levels, for approximate plotting purposes. Ideally this would be included in all files, but it is especially useful in the 1D files.

CRMs should output all variables, including 3D variables, over the full simulation. GCMs should output 0D, 1D, 2D variables over the full simulation and 3D variables over last 200 days of simulation. GCRMs should output all variables over the full simulation, but only upload 0D, 1D, and 2D variables to the RCEMIP data repository (3D variables should be archived

locally). **Note that the 3D output request is different than RCEMIP-I.**

Tables A1-A4 list the *required output*. Table A1 indicates the list of zero-dimensional domain-averaged variables (functions of $t$ only) that are to be computed and output as hourly averages. Table A2 indicates the list of one-dimensional domain-averaged profiles (functions of $z$ and $t$) that are to be computed and output as hourly averages. Table A3 indicates the list of two-dimensional variables (functions of $x$, $y$, and $t$) to output, as hourly averages. All models should output *tabot, uabot*, and *vabot*

(the air temperature, eastward wind, and northward wind, respectively, at the lowest model level). Those models which routinely estimate and output 2m air temperature, 10m eastward wind, and 10m northward wind (*tas, uas*, and *vas*, respectively), should

also output those variables. Models that use height coordinates should output vertical velocity, *wa500*, whereas models that use pressure-based coordinates should output omega, *wap500*. Table A4 indicates the list of three-dimensional variables to output, as instantaneous 6-hourly snapshots. Models that use height coordinates should output vertical velocity, *wa*, and pressure, *pa*, whereas models that use pressure-based coordinates should output omega, *wap*, and geopotential height *zg*.

Tables A5 lists *optional, but recommended output*, which consists of additional 3-D variables to be output as 6-hourly instantaneous snapshots, two 2-D variables to be output as 30-minute instantaneous snapshots (in contrast to the hourly averages requested in Table A3) and additional 2-D variables, including those associated with the frozen moist static energy (FMSE) budget. The 30-minute instantaneous output will facilitate tracking of mesoscale convective systems and the FMSE budget output will facilitate diagnosing physical mechanisms of convective aggregation. We also add as optional output several microphysical variables, which would facilitate analysis of microphysically-driven changes in deep convection, clouds, and climate. Since different microphysics schemes might predict a different set of variables, models should only output those quantities that are applicable to them. We also request *tntm*, the tendency of air temperature due to microphysical latent heating, as an optional 3-D output to facilitate heat budget analysis. This is the term $L(c - e)$ where $c$ is the condensation rate and $e$ is the evaporation rate. *crvi* is the 2-D mass-weighted vertical integral of the gross condensation rate, which should be computed as the mass-weighted vertical integral of the negative part of the microphysical tendency of water vapor. We note that if water vapor (or equivalent) is not a prognostic variable, this might not be feasible. **The optional requests for the microphysical variables and the 30-minute instantaneous 2-D variables are new compared to RCEMIP-I.**

## A2  Cloud fraction

We request the diagnosis of a global cloud fraction profile (*cldfrac_avg*) that includes all clouds and is the fraction of the entire domain covered by cloud at a given height (Table A2). This 1-D variable should be a function of vertical level and time. Following Stauffer and Wing (2022), the presence of a cloud in CRMs (or models without a cloud scheme) should be defined where the mixing ratio of the total cloud condensate (cloud liquid water + cloud ice) is greater than 1 x $10^{-5}$ g g$^{-1}$. **Note that this is different from the original RCEMIP-I definition in Wing et al. (2018)**. GCMs or other models with cloud schemes should continue to provide the cloud fraction as output from the cloud scheme. For GCMs, we also request the output of a total cloud fraction for each grid column as the 2-D variable *cl*, which is a function of $x$,$y$, and $t$ (Table A3), as well as the output of cloud fraction for each grid column as a function of height, as the 3-D variable *cldfrac*, which is a function of $x$, $y$, $t$, and vertical level (Table A4). Please output cloud fraction as a fraction (between 0 and 1), not a percentage out of 100. **The request of the 3-D variable *cldfrac* for GCMs is new compared to RCEMIP-I**.

## A3  Cloud water variables

Following the conventions of CMIP6 (see http://clipc-services.ceda.ac.uk/dreq/index.html for variable descriptions), several of the cloud water variables have confusing variable names. The 0-D variable *clwvi_avg* (Table A1 and the 2-D variable *clwvi* (Table A3) represent the condensed water path which includes both cloud ice and cloud liquid water. The 1-D variable *clw_avg* (Table A2) and the 3-D variable *clw* (Table A4) represent only the cloud liquid water.

## A4   Relative humidity

When computing relative humidity (the 1-D variable *hur_avg*, Table A2), care should be taken to compute the saturation with respect to liquid for temperatures above freezing and with respect to ice for temperatures below freezing. The formulas for saturation vapor pressure should follow those used in a given model's thermodynamics. If the model's thermodynamics interpolates between saturation over liquid and saturation over ice for temperatures near freezing, that should also be followed.

## A5   Radiative heating rates and fluxes

The tendency of air temperature due to shortwave and longwave radiative heating assuming clear-sky should be output as 3-D variables (*tntrscs*, *tntrlcs*), in addition to the total radiative heating tendencies (*tntrs*, *tntrl*); see Table A4. **The request of the 3-D variables *tntrscs* and *tntrlcs* is new compared to RCEMIP-I**. The domain average shortwave and longwave radiative heating rate profiles should also be output as 1-D variables for all-sky and clear-sky conditions (*tntrs_avg*, *tntrl_avg*, *tntrscs_avg*, *tntrlcs_avg*; see Table A2). The downwelling and upwelling shortwave and longwave radiative fluxes, at the surface and top of atmosphere, and for all-sky and clear-sky conditions, are requested as 2-D variables (Table A3).

## A6   CFMIP Observational Simulator Package (COSP)

COSP simulator outputs are requested for the ISCCP simulator, for GCMs that have COSP available as a diagnostic package. This will facilitate comparison with observations as well as the calculation and decomposition of cloud feedbacks (Zelinka et al., 2012a, b, 2013). More information about COSP can be found on the COSP website (https://cfmip.github.io). The ISCCP simulator provides pseudo-retrievals of cloud top pressure (CTP) and cloud optical thickness (tau) (Klein and Jakob, 1999; Webb et al., 2001). The following ISCCP simulator outputs are requested: hourly averages of ISCCP 2-D diagnostics (*cltisccp*: cloud fraction, *albisccp*: cloud albedo, and *pctisccp*: cloud top pressure; see Table A3) and 6-hourly diagnostics of the ISCCP CTP-tau histograms (*clisccp*: cloud area percentage in the 7 pressure and 7 optical depth bins). *clisccp* is listed in Table A4 because it is requested every 6 hours, but to clarify, we expect one histogram for each horizontal grid point. CRMs are not required to provide ISCCP simulator output, but if they have the capability to run COSP, they may provide the domain-mean ISCCP histogram as a comparison to planned offline calculations, either as a function of time or averaged over the last 100 days. **The request for ISCCP simulator output for GCMs is new compared to RCEMIP-I**.

## A7   Frozen Moist Static Energy Budget

Four optional 2D variables associated with the FMSE budget are requested in Table A5. If these variables (functions of $x$, $y$, and $t$) are diagnosed online in the model, their values may be output as hourly averages. If they are diagnosed offline from the instantaneous 3D output, they may be provided as instantaneous 6-hourly snapshots. For example, if *tnfmse*, the tendency of FMSE, is diagnosed offline, it should be diagnosed from instantaneous *fmse* output. If it is diagnosed online from instantaneous variables, its hourly average can then be output.

FMSE is defined as $h = c_p T + gz + L_v q - L_f q_{ice}$. The values of $c_p, g, L_v$, and $L_f$ used by the model formulation should be used to compute $h$. $q_{ice}$ is the mass fraction of all ice phase condensates (cloud ice, snow, etc...). The mass-weighted vertical integral of FMSE is given by:

$$\widehat{h} = \int_0^{z_{\text{top}}} (c_p T + gz + L_v q - L_f q_{ice}) \rho \, \mathrm{d}z, \tag{A7}$$

or, in pressure coordinates,

$$\widetilde{h} = \frac{1}{g} \int_{p_{\text{top}}}^{p_{\text{sfc}}} (c_p T + gz + L_v q - L_f q_{ice}) \, \mathrm{d}p. \tag{A8}$$

Care should be taken to make sure the same limits of integration are used at all times/locations. The mass-weighted vertical integral of the advective tendency of FMSE (*advfmse*) is given by

$$\int_0^{z_{\text{top}}} \left( u \frac{\partial h}{\partial x} + v \frac{\partial h}{\partial y} + w \frac{\partial h}{\partial z} \right) \rho \, \mathrm{d}z. \tag{A9}$$

Ideally, FMSE would be diagnosed online and each model's advection scheme used to advect it. For instance, one could diagnose FSME from the prognostic variables just before and just after they are advected, and then the difference could be taken as a measure of the FMSE advective tendency. If this is not possible we ask that groups make their best effort to estimate these terms. The spatial variance of the mass-weighted vertical integral of frozen moist static energy is computed using the squared anomalies from the horizontal mean of the mass-weighted vertical integral of moist static energy ($\widehat{h}$). Its tendency (*tnfmsevar*) is given by

$$\frac{\partial}{\partial t} \left( \int_0^{z_{\text{top}}} h \rho \, \mathrm{d}z \right)'^2 \tag{A10}$$

where $'$ indicates an anomaly from the horizontal mean.

**Table A1.** 0D hourly-averaged variables ($t$). Italicized variables are *not* standard CMIP output.

| Variable Name | Description | Units |
|---|---|---|
| pr_avg | domain avg. surface precipitation rate | kg m$^{-2}$ s$^{-1}$ |
| hfls_avg | domain avg. surface upward latent heat flux | W m$^{-2}$ |
| hfss_avg | domain avg. surface upward sensible heat flux | W m$^{-2}$ |
| prw_avg | domain avg. water vapor path | kg m$^{-2}$ |
| *sprw_avg* | domain avg. saturated water vapor path | kg m$^{-2}$ |
| clwvi_avg | domain avg. condensed water path (cloud ice + cloud liquid) | kg m$^{-2}$ |
| clivi_avg | domain avg. ice water path (cloud ice) | kg m$^{-2}$ |
| rlds_avg | domain avg. surface downwelling longwave flux | W m$^{-2}$ |
| rlus_avg | domain avg. surface upwelling longwave flux | W m$^{-2}$ |
| rsds_avg | domain avg. surface downwelling shortwave flux | W m$^{-2}$ |
| rsus_avg | domain avg. surface upwelling shortwave flux | W m$^{-2}$ |
| rsdscs_avg | domain avg. surface downwelling shortwave flux - clear sky | W m$^{-2}$ |
| rsuscs_avg | domain avg. surface upwelling shortwave flux - clear sky | W m$^{-2}$ |
| rldscs_avg | domain avg. surface downwelling longwave flux - clear sky | W m$^{-2}$ |
| rluscs_avg | domain avg. surface upwelling longwave flux - clear sky | W m$^{-2}$ |
| rsdt_avg | domain avg. TOA incoming shortwave flux | W m$^{-2}$ |
| rsut_avg | domain avg. TOA outgoing shortwave flux | W m$^{-2}$ |
| rlut_avg | domain avg. TOA outgoing longwave flux | W m$^{-2}$ |
| rsutcs_avg | domain avg. TOA outgoing shortwave flux - clear sky | W m$^{-2}$ |
| rlutcs_avg | domain avg. TOA outgoing longwave flux -clear sky | W m$^{-2}$ |

**Table A2.** 1D hourly-averaged variables ($z$,$t$). Italicized variables are *not* standard CMIP output.

| Variable Name | Description | Units |
|---|---|---|
| ta_avg | domain avg. air temperature profile | K |
| ua_avg | domain avg. eastward wind profile | m s$^{-1}$ |
| va_avg | domain avg. northward wind profile | m s$^{-1}$ |
| hus_avg | domain avg. specific humidity profile | kg/kg |
| hur_avg | domain avg. relative humidity profile | % |
| clw_avg | domain avg. mass fraction of cloud liquid water profile | kg/kg |
| cli_avg | domain avg. mass fraction of cloud ice profile | kg/kg |
| *plw_avg* | domain avg. mass fraction of precipitating liquid water profile | kg/kg |
| *pli_avg* | domain avg. mass fraction of precipitating ice profile | kg/kg |
| theta_avg | domain avg. potential temperature profile | K |
| thetae_avg | domain avg. equivalent potential temperature profile | K |
| tntrs_avg | domain avg. shortwave radiative heating rate profile | K s$^{-1}$ |
| tntrl_avg | domain avg. longwave radiative heating rate profile | K s$^{-1}$ |
| tntrscs_avg | domain avg. shortwave radiative heating rate profile - clear sky | K s$^{-1}$ |
| tntrlcs_avg | domain avg. longwave radiative heating rate profile - clear sky | K s$^{-1}$ |
| *cldfrac_avg* | global cloud fraction profile | |

**Table A3.** 2D hourly averaged variables ($x$,$y$,$t$). Italicized variables are *not* standard CMIP output. Bolded variables are a new request compared to RCEMIP-I. Variables with a ($^!$) symbol are required only for models with parameterized convection.

| Variable Name | Description | Units |
|---|---|---|
| pr | surface precipitation rate | kg m$^{-2}$ s$^{-1}$ |
| hfls | surface upward latent heat flux | W m$^{-2}$ |
| hfss | surface upward sensible heat flux | W m$^{-2}$ |
| rlds | surface downwelling longwave flux | W m$^{-2}$ |
| rlus | surface upwelling longwave flux | W m$^{-2}$ |
| rsds | surface downwelling shortwave flux | W m$^{-2}$ |
| rsus | surface upwelling shortwave flux | W m$^{-2}$ |
| rsdscs | surface downwelling shortwave flux - clear sky | W m$^{-2}$ |
| rsuscs | surface upwelling shortwave flux - clear sky | W m$^{-2}$ |
| rldscs | surface downwelling longwave flux - clear sky | W m$^{-2}$ |
| rluscs | surface upwelling longwave flux - clear sky | W m$^{-2}$ |
| rsdt | TOA incoming shortwave flux | W m$^{-2}$ |
| rsut | TOA outgoing shortwave flux | W m$^{-2}$ |
| rlut | TOA outgoing longwave flux | W m$^{-2}$ |
| rsutcs | TOA outgoing shortwave flux - clear sky | W m$^{-2}$ |
| rlutcs | TOA outgoing longwave flux -clear sky | W m$^{-2}$ |
| prw | water vapor path | kg m$^{-2}$ |
| *sprw* | saturated water vapor path | kg m$^{-2}$ |
| clwvi | condensed water path (cloud ice + cloud liquid) | kg m$^{-2}$ |
| clivi | ice water path (cloud ice) | kg m$^{-2}$ |
| psl | sea level pressure | Pa |
| tas | 2m air temperature | K |
| tabot | air temperature at lowest model level | K |
| uas | 10m eastward wind | m s$^{-1}$ |
| vas | 10m northward wind | m s$^{-1}$ |
| uabot | eastward wind at lowest model level | m s$^{-1}$ |
| vabot | northward wind at lowest model level | m s$^{-1}$ |
| *wa500* or *wap500* | vertical velocity or omega at 500 hPa | m s$^{-1}$ or Pa s$^{-1}$ |
| cl$^!$ | total cloud fraction of grid column | |
| pr_conv$^!$ | surface convective precipitation rate | kg m$^{-2}$ s$^{-1}$ |
| **albisccp**$^!$ | ISCCP mean cloud albedo | |
| **cltisccp**$^!$ | ISCCP total cloud cover | % |
| **pctisccp**$^!$ | ISCCP mean cloud top pressure | Pa |

**Table A4.** 3D instantaneous 6-hourly variables ($x$,$y$,$z$,$t$).Italicized variables are *not* standard CMIP output. Bolded variables are a new request compared to RCEMIP-I. Variables with a ($^!$) symbol are required only for models with parameterized convection.

| Variable Name | Description | Units |
|---|---|---|
| clw | mass fraction of cloud liquid water | g/g |
| cli | mass fraction of cloud ice | g/g |
| *plw* | mass fraction of precipitating liquid water | g/g |
| *pli* | mass fraction of precipitating ice | g/g |
| ta | air temperature | K |
| ua | eastward wind | m s$^{-1}$ |
| va | northward wind | m s$^{-1}$ |
| hus | specific humidity | g/g |
| *wa* or wap | vertical velocity or omega | m s$^{-1}$ or Pa s$^{-1}$ |
| *pa* or zg | pressure or geopotenial height | Pa or m |
| *tntrs* | tendency of air temperature due to shortwave radiative heating | K s$^{-1}$ |
| *tntrl* | tendency of air temperature due to longwave radiative heating | K s$^{-1}$ |
| *tntrscs* | tendency of air temperature due to shortwave radiative heating - clear sky | K s$^{-1}$ |
| *tntrlcs* | tendency of air temperature due to longwave radiative heating - clear sky | K s$^{-1}$ |
| *cldfrac*$^!$ | cloud fraction | |
| mc$^!$ | convective mass flux | kg m$^{-2}$ s$^{-1}$ |
| tntc$^!$ | tendency of air temperature due to moist convection | K s$^{-1}$ |
| **clisccp**$^!$ | ISCCP cloud area percentage in optical depth and pressure bins | % |

**Table A5.** Optional output variables. Italicized variables are *not* standard CMIP output. Bolded variables are a new request compared to RCEMIP-I.

| Variable Name | Description | Units |
| --- | --- | --- |
| 6-hourly instantaneous 3-D variables ($x,y,z,t$) | | |
| ***tntm*** | tendency of air temperature due to microphysical latent heating | K s$^{-1}$ |
| **reffclw** | effective radius of cloud liquid water (in-cloud) | $\mu$m |
| **reffcli** | effective radius of cloud ice (in-cloud) | $\mu$m |
| ***cdnc*** | number concentration of cloud liquid water particles (in-cloud) | cm$^{-3}$ |
| ***icnc*** | number concentration of cloud ice particles (in-cloud) | cm$^{-3}$ |
| | | |
| 30-minute instantaneous 2-D variables ($x,y,t$) | | |
| **rlut_inst** | TOA outgoing longwave flux | W m$^{-2}$ |
| **pr_inst** | surface precipitation rate | kg m$^{-2}$ s$^{-1}$ |
| | | |
| Hourly 2-D variables ($x,y,t$) | | |
| *fmse* | mass-weighted vert. integral of FMSE | J m$^{-2}$ |
| *advfmse* | mass-weighted vert. integral of advective tendency of FMSE | J m$^{-2}$ s$^{-1}$ |
| *tnfmse* | tendency of mass-weighted vert. integral of FMSE | J m$^{-2}$ s$^{-1}$ |
| *tnfmsevar* | tendency of spatial variance of mass-weighted vert. integral of FMSE | J$^2$ m$^{-4}$ s$^{-1}$ |
| ***crvi*** | mass-weighted vert. integral of gross condensation rate | kg m$^{-2}$ s$^{-1}$ |

*Author contributions.* AAW led the writing of the text. AAW and LGS performed and analyzed simulations. All authors contributed to discussing the goals and specifications of RCEMIP-II and editing the text.

*Competing interests.* No competing interests are present.

*Acknowledgements.* AAW acknowledges support from NSF grants AGS-2140419 and AGS-1830724. LGS and KAR acknowledge support from NSF grant AGS-2327958. KAR was also partially supported by the Department of Energy Office of Science award number DE-SC0016605 "A Framework for Improving Analysis and Modeling of Earth System and Intersectoral Dynamics at Regional Scales" The authors thank Tim Cronin, Guy Dagan, Peter Hill, Nick Lutsko, Andrew Williams, and the participants of the RCEMIP breakout session at the 2023 CFMIP-GASS Meeting in Paris for helpful discussions, to Catherine Stauffer for assistance with initial analysis of the test simulations, and to Graham O'Donnell for calculating streamfunction in the SAM simulations and generating Figures 2 and 8. We also thank Peter Bogenschutz, Jean-Pierre Chaboureau, Guy Dagan, Stephan de Roode, Blaž Gasparini, Walter Hannah, Keiichi Hashimoto, Peter Hill, Cathy Hohenegger, Chris Holloway, Fredrik Jansson, Todd Jones, Marat Khairoutdinov, Gabrielle Leung, Cheng Li, Shuhei Matsugishi,

Hiroaki Miura, David Randall, Camille Risi, Romain Roehrig, David Romps, Masaki Satoh, Lorenzo Tomassini, Sue van den Heever, Chien-Ming Wu, Tomoro Yanase, and Ming Zhao for agreeing to perform RCEMIP-II simulations with the models listed in Table 2. We also thank Marat Khairoutdinov for maintaining and providing the SAM cloud-resolving model, which was used to perform simulations in this paper, and two anonymous reviewers for in depth reviews that led to improvements in the manuscript. We would like to acknowledge high-performance computing support from Cheyenne (doi:10.5065/D6RX99HX) provided by NCAR's Computational and Information Systems Laboratory, sponsored by the National Science Foundation (CISL, 2019).

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
