# Peer review of "RCEMIP-II: Mock-Walker Simulations as Phase II of the Radiative-Convective Equilibrium Model Intercomparison Project"

_Geoscientific Model Development, 2023_

## Referee Comment (RC2)

Wing et al. have suggested a protocol for the second phase of the Radiative Convective Equilibrium Model Intercomparison Project (RCEMIP), following on from the Phase One experiments which were motivated in their 2017 paper, also in GMD. My general thoughts are that the protocol is poorly motivated, and neglects many of the (arguably more promising) avenues in which a Phase II of RCEMIP could explore in favor of a relatively untested setup, the 'mock-Walker' simulation.

In my opinion, the following issues mean that the current protocol is unfit for publication in its current form.

**Issue #1: approach to constraining model diversity**

The 2017 protocol paper (RCEMIP-I) outlined a series of small and large domain CRM/LES simulations, in addition to suggested runs with global models and single-column models. One of the key takeaways from RCEMIP-I is that there is substantial diversity in the simulated RCE state even in the absence of convective self-aggregation.

This point is acknowledged in the current manuscript, which notes that: "While several robust results emerged across the spectrum of models that participated in the first phase of RCEMIP (RCEMIP-I), two points that stand out are (1) the strikingly large diversity in simulated climate states and (2) the strong imprint of convective self-aggregation on the climate state." This diversity in RCEMIP-I is also acknowledged to be driven by "representations of convection, microphysics, turbulence, and dynamical cores".

However, the current manuscript does little to tackle the question of inter-model diversity in the RCE state and instead primarily focuses on point (2) stating that "...the wide range in the degree of self-aggregation and the lack of consensus in its temperature dependence is a barrier to understanding."

This emphasis is counter to the vision laid out in the RCEMIP-I protocol (and contrary to the response given to the RCEMIP-I reviewers, quotations from which will be cited below). Originally, RCEMIP-I was presented as an opportunity to explore inter-model diversity, with RCEMIP-II being an opportunity to try to understand/narrow that diversity through the use of simplified radiation/microphysics schemes. For example, many of the reviewers for RCEMIP-I strongly suggested the use of simplified radiation and/or microphysics schemes as a way to better understand the diversity of RCE states and their response to warming. A list of these is presented below:

(Isaac Held): "*I would strongly encourage you to reconsider and ask groups to run with a standard microphysical mechanism in addition to their model's microphysics. Otherwise, there is a good chance that the diversity of simulations will be dominated by the diversity of microphysical assumptions. (For example, we know that in RCEs assumptions about ice fall-speeds will exert a strong control on the cirrus climate.)*"

(Levi Silvers): "*It is not essential, but I think it would be useful to be more precise about a second set of non-required (Tier 2) experiments. This could be written in such a way that modeling centers wishing to participate with minimal effort are not thus discouraged from participating, but that more ambitious modeling centers or individuals could clearly push farther into the project in a coordinated way. My suggestions for further experiments would be: 1. Rotating RCE 2. GCMs in RCE mode with convective parameterization turned off 3. RCE with cloud RCE off (COOKIE type experiments) 4. Kessler physics across the hierarchy of models*"

(Anonymous reviewer IV): "*...the resulting equilibrium state and the clustering may look very different in the different CRMs. It will thus be very difficult to compare the different*

*models and to identify the root for the differences (radiation scheme, microphysics parametrizations, . . .). An even simpler setup for the models could therefore be useful to identify, which schemes are responsible for the differences. As suggested by the authors and brought forward by Jeevanjee et al. (2017) a simplified microphysics scheme could be one option. A further option could be to simplify the longwave radiative cooling, as e.g. described in Muller and Bony (2015).*"

(Nadir Jeevanjee): "*As advocated for by Isaac and other reviewers, there is also interest here in using simplified (Kessler) microphysics in our RCE setup. Such a scheme already exists in development branches of our code.*"

Furthermore, this was explicitly stated as a potential target for the second phase of RCEMIP (Sec 6.2 of 2017 protocol paper):

"*Additional simulations could be performed to assess the sensitivity to dynamical core, radiation scheme, microphysics scheme, boundary layer scheme, convective scheme (in the case of models with parameterized convection), and the sensitivity to various parameters in those schemes (such as the entrainment parameter in a convective scheme).*"

Additionally, in the reply to Reviewer IV of RCEMIP-I (similar comments exist in the reply to other reviewers):

"*We agree that large differences could result from differences in physical parameterizations. However, we think that it is useful to first determine the full range of RCE simulations and then proceed to test the parameterization sensitivity by imposing a simple microphysics scheme on all models in the second phase of RCEMIP.*"

From these statements it is clear that simplified experiments were anticipated to be necessary/useful by reviewers for understanding inter-model diversity in the simulated RCE state. This was also acknowledged by the authors and explicitly suggested as a route forward for Phase II of RCEMIP. The need for idealized experiments seems even more necessary now that the RCEMIP-I simulations have demonstrated such an extreme diversity in the simulated RCE state (perhaps even larger than expected).

Furthermore, since RCEMIP-I it has been fairly well-established that relevant quantities such as high cloud feedbacks are extremely sensitive to microphysics. For example, both Wing et al. (2020) and Stauffer and Wing (2022) demonstrated that about one-third of models do not exhibit an 'iris' effect, and in fact have an *increase* in high cloud fraction under warming despite a decrease in clear-sky divergence at the anvil level. The disconnect between radiatively driven divergence and high-cloud fraction explored in Seeley et al (2019), Beydoun et al. (2021) and Jeevanjee (2023) (among others), and one key takeaway from those papers is that the lifetime of detrained cloud condensate is a crucial determinant of high-cloud fraction and the anvil cloud feedback. It is thus extremely confusing that the authors do not explore the possibility of using simplified microphysics schemes, or at least require that all models output microphysical process rates and (where relevant) particle size distributions, which would be trivial for most models, and allow for a better understanding of this crucial feedback.

While implementing idealized microphysical schemes is a potential burden on modeling centers (though some models such as FV3 already have the option to use a Kessler-style microphysics package), it is quite simple to replace interactive radiative transfer with a prescribed radiative cooling rate. For example, Paulius and Garner (2006) use a simple 1.5K/day radiative cooling rate up to a fixed tropopause temperature. Such an approach is already widely used in the RCE literature. Additionally, a slightly more realistic approach, which better captures the effect of warming on radiative fluxes, is to prescribe the

radiative flux divergence in temperature coordinates ($\partial_T F$) as being linear in temperature (following Eq. 2 of Jeevanjee and Zhou (JAMES, 2022)). Such an approach was also used by Seeley and Wordsworth (2023).

My point here is not simply that we should revisit Phase One experiments, for which simplified runs would be helpful, but also that runs with simplified physics will likely be *even more necessary* in RCEMIP-II if the authors do intend on using 'mock-Walker' simulations. This is because it is now well-established (although not cited in the protocol paper…) that the 'mock-Walker' simulations are prone to develop stacked overturning circulations (e.g. Grabowski et al., 2000; Yano et al., 2002; Larson and Hartmann, 2003; Liu and Moncrieff , 2008; Silvers and Robinson, 2021), due to the interactions between radiation and detrained condensate/water vapor (e.g. Nuijens and Emanuel, 2018; Sokol and Hartmann, 2022; Lutsko and Cronin, 2023). An example of which, from experiments with the ICON model, is shown below (domain is ~3000x100km, with dx=2km, but similar results are obtained for the standard RCE_large setup):

[Figure]

Figure 1: Time-averaged streamfunction (red=clockwise motion; blue=anticlockwise motion) and cloud liquid water content in our control simulation, driven by a zonal SST gradient between the centre and edges of the domain.

As shown in Fig. 1, the ICON model exhibits a pronounced 'triple-cell' structure, with three stacked overturning circulations. This structure differs between models, with some simulating a triple-cell structure (ICON), others a double-cell (e.g. Silvers and Robinson, 2021). Furthermore, the onset of these stacked overturning structures is itself dependent on temperature (Lutsko and Cronin, 2018, 2023). In both their 2018 experiments (2D CRM) and in their 2023 experiments (3D CRM), Lutsko and Cronin found that the circulation transitions from a single- to double-cell structure at mean SSTs of ~300K. Although this number is likely model-dependent, it is worrying that in their SAM simulations the transition occurred right in the middle of the SSTs considered as part of the Phase Two RCEMIP protocol.

This model- and SST-dependent transition between different stacked circulation structures will certainly complicate analysis and interpretation of the proposed RCEMIP-II experiments. In light of this I strongly disagree with the statement on L103 that mock-Walker

simulations will "...reduce the diversity of simulated climates and provide a clearer tie to observations…"

Previous studies have shown that the stacked-overturning cells can be effectively suppressed by prescribing uniform (or otherwise simplified) radiative cooling profiles (e.g., Grabowski et al., 2000; Wofsy and Kuang, 2012; Sokol and Hartmann, 2022; Lutsko and Cronin, 2023). This is effective at suppressing these features because they are largely driven by sharp vertical gradients in the longwave cooling rate, caused by sharp gradients in moisture near the melting line. These moisture gradients are themselves associated with complex interactions between convection, microphysics and radiation; near the melting line, evaporation of detrained condensate and melting of ice form a stable layer which promotes further detrainment of condensate, whose enhanced radiative appears to draw out more condensate (Nuijens and Emanuel, 2018; Sokol and Hartmann, 2022).

Overall, one of the most salient features of mock-Walker circulations is their stacked overturning circulations and the elevated moist layers associated with them. Both of these appear to be governed by complex interactions between microphysics (with both the liquid and ice phase playing a key role), convection and radiation. Hence, it seems highly unlikely that these mock-Walker simulations will "provide a constraint on convection and circulation" or "narrow the intermodel spread", as hypothesized in the protocol.

I find it somewhat unlikely the authors will give up on mock-Walker simulations altogether, but to claim (as they do in the manuscript) that they will narrow model diversity is quite naïve. If they do insist on using mock-Walker simulations (see point #2, below for more on this), then it is highly recommended that they pair such simulations with simplified radiation/microphysics runs to help narrow some of the model diversity and aid interpretation and understanding of the results. If it is deemed too complex to impose a uniform microphysics scheme, then models should instead be required to output comprehensive microphysical diagnostics.

**Issue #2: the choice of mock-Walker simulations over other options**

The RCEMIP-I protocol named a number of possible routes forward for Phase Two. For example, introducing a mixed-layer ocean to ensure a closed surface energy budget, or introducing rotation (f-plane / beta-plane / full spherical geometry) in order to simulate tropical cyclones and equatorial waves. Both of these options have been explored in single-model studies prior to and following on from RCEMIP-I, and thus it is confusing why the authors decided to use the mock-Walker setup instead of these alternative approaches. For example, investigating tropical cyclones in f-plane simulations has been richly explored in a number of single-model papers, but each with differing setups. This seems like a natural avenue for RCEMIP, especially given that tropical cyclones are an incredibly impactful phenomena which we frequently observe in the real-world, as opposed to the "stacked" overturning circulations of mock-Walker cells (which have no obvious observational analogue).

During the RCEMIP breakout session at CFMIP 2023, a number of participants expressed interest in rotating RCE experiments. It is thus extremely strange to me that the current protocol paper does not justify its emphasis on mock-Walker simulations and argue for their benefit *in comparison to* rotating RCE (or indeed other configurations suggested by the RCEMIP Phase One paper).

---

## Editor Comment (EC2)

Dear Dr. Wing and Co-authors,

Thank you again for submitting the RCEMIP-II experiment description paper for potential publication in GMD. Since there is still time for community commentary and a dialogue between yourselves and the reviewers (and any community commentators), I want to take a brief moment to weigh in on the reviews already received. To start, I want to recognize and appreciate the amount of thought and effort that clearly went in to devising this experimental protocol, to testing it on SAM and CESM, and to writing this paper.

Both Anonymous Reviewers raise some potentially serious concerns about the mock-Walker simulations proposed: particularly about their ability to clarify the origins of intermodel differences in convective aggregation. In particular, Anonymous Reviewer #2 raises the broader question of why mock-Walker simulations were chosen as the experimental direction for RCEMIP-II. Based on my read of the paper, the justification for the focus on mock-Walker simulations comes from the following train of logic: (1) RCEMIP-I used homogenous SSTs (line 53), (2) RCEMIP-I analyses revealed large sensitivities to "convection, microphysics, turbulence, and dynamical cores" (lines 55-56), and (3) the second phase of RCEMIP specifies idealized, but more realistic SSTs (lines 57-59). An key line in this section ("sensitivities that may have been masked in other intercomparisons by dynamical constraints", lines 56-57) suggests that the motivation for prescribed, heterogenous SSTs might be to trigger a dynamical circulation in the simulations that constrains the convective aggregation response. That said, I do think this motivation could use more clarification. Also, I want to highlight Anonymous Reviewer #2's concern that there might be a wide variety in dynamical responses to the imposed SSTs that could make it more difficult to understand the variety of RCE states that models exhibit.

Since the GMD discussion period is still open, I am going to refrain from making a decision or a recommendation at this point. Rather, I would encourage you and your co-authors to take advantage of the open-discussion feature of GMD to interact with the reviewers. For example, it might help to clarify: the specific question(s) that this MIP intends to address, the process for deciding on which question(s) RCEMIP-II should address (e.g., did this stem from one or several RCEMIP group meetings?), and the logic for choosing mock-Walker simulations to address these questions. It might also be useful to invite others in the RCEMIP community to engage in the discussion here, since a MIP is only useful if multiple modeling groups commit to performing experiments for the MIP. You are also welcome to ask clarifying questions; I will encourage the reviewers to respond accordingly. If you would like more time in the discussion phase, please reach out and we can extend the discussion.

Alternatively, you are welcome to treat this is a more traditional peer-review process in which you wait for the GMD discussion to close and then provide a revised paper and a line-by-line response to the reviewer concerns. In that case, I would treat this as a 'major revision'. That said, I suspect that some dialog may end up helping you and your co-authors come to a consensus with the reviewers about modifications to the RCEMIP-II protocol and/or paper that best answer the questions about model responses to RCE that you intend to address with this next phase of RCEMIP.

With Kind Regards, Travis A. O'Brien Pronouns: he/him

Topical Editor Geoscientific Model Development, Assistant Professor Earth and Atmospheric Sciences Indiana University, Bloomington

---

## Author Comment (AC4)

**Initial Response to Reviewer Comments**

We are providing an initial response to the reviewer comments while the discussion period is still open, which is intended to provide more background on how mock-Walker simulations came to be selected for RCEMIP-II and justification for the experimental design. A line-by-line response to other aspects of the reviewer comments is reserved for after the discussion period has closed. If given the opportunity we would revise the manuscript to clarify the motivations for mock-Walker simulations as the choice for RCEMIP-II as discussed below.

**Evolution of Phase II:**

The authors of this paper began discussing possible directions for a second phase in Spring 2021, shaped by the initial findings and ongoing analysis of RCEMIP-I by ourselves and other groups. Mock-Walker simulations emerged as our preference, based on their ability to satisfy the four principles discussed below in response to reviewer #2, and the fact that numerous groups in the community were already performing such simulations, with a variety of interesting, yet complex, results. We then inquired with some of the other core RCEMIP participants regarding their thoughts about this proposal. After hearing positive feedback as to the value of this choice and its suitability as a way to move forward, as well as willingness to perform the simulations, we reached out to everyone that contributed simulations to phase I of RCEMIP. Again, we received positive feedback and willingness to contribute, such that by the time A. Wing submitted an NSF proposal in summer 2021 proposing to perform mock-Walker simulations as a phase II of RCEMIP, 18 models had indicated that they would participate.

By summer 2022, Wing's NSF proposal had been funded and mock-Walker simulations were presented at the CFMIP meeting in Seattle and the 2nd Model Hierarchies Workshop at Stanford University as the idea for phase II. Through informal conversations at these meetings and communication with others in the community who had previously or were actively performing their own mock-Walker simulations, the detailed protocol began to take shape. At the end of 2022, mock-Walker simulations were announced to the broader RCEMIP community as the planned phase II, as were other potential opportunities to utilize the RCEMIP set-up to investigate other phenomena, such as tropical cyclones in rotating RCE simulations and cloud-aerosol interactions. The former received interest from a handful of people, while the latter, led by Guy Dagan (Hebrew University) received interest from about 10 CRMs and has subsequently proceeded as an offshoot of RCEMIP (RCEMIP-ACI). Throughout 2023, we tested and optimized the mock-Walker protocol and simulation design. Throughout, we sought to listen to input from the community and address any concerns they brought up. For example, we performed the domain size tests presented in this paper in response to a concern from a community member that there may be sensitivities to both the domain length and width. We switched from prescribing the same wavelength of the SST perturbation across CRMs of slightly different domain lengths to enforcing a wavelength always equal to the domain length, after a RCEMIP participant pointed out some numerical and non-physical artifacts of the former option. At the Joint CFMIP-GASS Meeting in Paris in July 2023, we held a RCEMIP breakout discussion. This was an opportunity to hear further feedback from the community before the protocol was finalized. The main topics of conversation were (1) exactly what delta-SST values to use; (2) the GCM configuration; and (3) the output request. In response to this, we performed 9 additional sensitivity tests to help us determine the delta-SST values as indicated in this paper. We added additional output variables to meet the analysis needs of the community. We re-considered the GCM configuration, though we ended up back at our original proposal (the one presented in this paper) as the best way to make the CRM and GCM set-ups consistent. As reviewer #2 notes, there was also some interest expressed in rotating RCE experiments, but we elected to stick with mock-Walker simulations as the choice for the official phase II for the reasons discussed below. We have received a much smaller volume of interest in rotating RCE as the framework for RCEMIP II, compared with the mock-Walker simulations. As pointed out by the editor, a MIP is only useful if multiple modeling groups commit to performing the experiments. At the time of the submission of this protocol paper in December 2023, 17 people in addition to the authors of this paper had expressed enthusiasm about the proposed mock-Walker simulations and their willingness to contribute simulations. The 20 models planning to participate are listed in Table 2 of the paper. Since submitting this paper, an additional ~10 people have expressed interest in participating. While the lengthy period of testing and optimization was necessary to ensure a robust protocol and address issues pointed out by the community, further delay in beginning phase II risks losing the momentum we have built.

**Reviewer #1**

**Maturity of the mock-Walker set-up:**

In designing the protocol for the mock-Walker simulations as RCEMIP-II, we tested different versions of the equation for prescribed SSTs, the choice of delta-SSTs, the CRM domain length and width, maintaining different aspects of the SST pattern across CRMs of slightly different domain length, and how to make the CRM and GCM set-ups consistent. While there are always more sensitivity tests that could be done, the protocol was not decided upon lightly and is instead the result of several years of consideration and optimization. As described above regarding the evolution of phase II, in multiple cases, we performed additional tests and even changed the protocol in response to comments from the RCEMIP community. Therefore, we feel like the testing presented in this paper in combination with the substantial body of literature on mock-Walker simulations over past decades makes it appropriate to do a mock-Walker intercomparison. Further tests in the context of one or a few models could continue to be performed in parallel with the broader intercomparison, as needed.

**SST gradient vs. absolute SSTs:**

Reviewer #1 makes a good point that it is unclear which SST quantity matters most for the climate of mock-Walker simulations. From the perspective of the weak temperature gradient approximation, the absolute SST contrast (maxSST - minSST) and the maximum SST is what ought to matter to the dynamics. The SST gradient (dSST/dx) might plausibly set horizontal flow

speeds under Lindzen-Nigam type arguments, and if that is the case, the SST laplacian ought to matter for vertical motion and precipitation. Ideally all three of these parameters would be kept fixed across the models, but due to computational limitations on domain size this is not possible. This issue is one that we considered at length, discussed with other members of the RCEMIP community, and tested extensively in preparing the RCEMIP-II protocol.

We discussed two possible options in the paper: (1) enforcing that the wavelength equals the domain length and (2) enforcing the same wavelength (6000 km) regardless of domain length. Option (1) keeps the absolute SST contrast (delta-SST), maximum SST, and mean SST the same but leads to slightly different SST gradients. Option (2) leads to slightly different mean SSTs, a discontinuous SST distribution at the boundaries, and the projection of the prescribed SST forcing onto all scales, introducing substantial noise at higher wavenumbers.

After testing in one CRM, SAM, we elected to go with option (1). Even though this choice could cause differences in the results due to differences in domain length, these differences would at least result from a physical reason (a different SST gradient) which we felt was preferable to the non-physical artifacts present in option (2). Adjusting the value of delta-SST, as the reviewer suggests, would cause differences in the absolute SST contrast and maximum SSTs. We feel that keeping the absolute SST contrast and maximum SSTs consistent (our chosen option) is the most elegant and plausibly what matters most for dynamics, precipitation, clouds, etc... based on weak temperature gradient arguments. It is also the simplest to implement.

The differences in the SST gradient that would result from different domain lengths are small compared to the differences in SST gradient from choosing 0.625 K, 0.75 K, or 1 K as the "weak gradient", for instance. So, while we cannot rule out that the differences in domain size could contribute to differences across models, we believe it would be a small effect. This is supported by the testing we did in SAM, in which our results did not qualitatively depend on the choice of option (1) or option (2). We could attempt to determine the influence of domain length on the results by assessing if models with a larger domain length behave systematically differently from those with a smaller domain length. While we acknowledge that it could be difficult to disentangle the relative contributions of the domain length difference and other aspects of model physics and numerics to intermodel differences, this difficulty would also be present if we were to employ any of the other options. The only thing we can do is try to make the set-up as uniform as possible given computational limitations and avoid the imposition of non-physical artifacts.

**GCM set-up:**

Regarding the reviewer's concern about the GCM set-up, we considered and tested alternate geometries of the SST pattern, which was a subject of discussion in the RCEMIP breakout session at the 2023 Joint CFMIP-GASS Meeting in Paris. We ultimately elected to utilize zonal bands of hot and cold SSTs to ensure the closest possible correspondence to the set-up in the doubly periodic long-channel CRM domain (chosen to be identical to the RCEMIP-I domain), including the mean SST, maximum SST, and SST gradient. Given the double periodicity, the CRM

domain should be conceptualized as being infinitely repeated in both dimensions, which would then (other than the sphericity) make it analogous to the GCM set-up. This is demonstrated in the figure below, in which the SSTs in the GCM are shown on the left and the SSTs in the CRM are shown on the right, in which the CRM has been rotated and tiled 24 times in one dimension and 3 times in the other, to emulate the GCM domain. Note that since these simulations are non-rotating, the choice of which direction is x- and which direction is y- does not matter.

Since one of the core principles of RCEMIP is to be able to compare limited area CRMs and GCMs, we chose to confine the warm SSTs in only one direction in both model types. The GCM set-up is similar to that used in Mueller and Hohenegger (2020), which utilized zonally homogenous but meridionally varying SSTs. Convective self-aggregation within the warm latitude bands is indeed possible, as the reviewer suggests. Zonal contraction of convection was seen in Mueller and Hohenegger (2020) and in our test simulations with CAM. While this could complicate interpretation, it will also be interesting to see how the degree to which this "intrinsic" self-aggregation emerges on top of the forced convergence varies across models. The ability to study self-aggregation both in the context of SST gradients and constant SST is one of our motivations for this particular setup and creates an additional connecting point both with the RCEMIP experiments and the tropical oceans of Earth.

**Reviewer #2**

In selecting an experimental design for phase II of RCEMIP, we sought to follow the following principles, in the spirit of the design of RCEMIP-I:

- 1) The ability to directly compare CRMs and GCMs
- 2) Ease of implementation, to encourage the broadest possible participation
- 3) Permitting continued investigation of the three themes of RCEMIP while moving a step up the model hierarchy
- 4) Providing some sort of constraint on convection.

Several different possibilities were suggested in the RCEMIP-I protocol paper (Wing et al. 2018) and were considered by the author, as paths forward for a second phase. A mock-Walker configuration, as proposed, meets all of these criteria. It does not prohibit other possible studies but simply represents a practical direction that both our conversations with colleagues and our work over the past few years has naturally taken. Above we reviewed the process by which the proposed mock-Walker protocol emerged. Here, we discuss how mock-Walker simulations satisfy these four principles, whereas other possible options for a second phase of RCEMIP do not.

**Mock-Walker:**

The proposed mock-Walker experimental design maintains an identical set-up to RCEMIP-I with the exception of a simple, prescribed SST pattern. This is easy to implement and allows for direct comparison with RCEMIP-I. Care was taken to maintain consistent values of mean SST, maximum SST, and SST gradient between the CRM and GCM domains, as much as possible. These characteristics satisfy **principles 1 and 2**.

From its inception, RCEMIP has been motivated in part by a desire to better understand how the balance between convection and radiation interacts with large-scale circulations (Wing et al., 2018). However, the only large-scale circulations present in RCEMIP-I are those generated by self-aggregation. One of our motivations for selecting mock-Walker simulations is a desire to broaden the range of dynamical regimes and cloud types that can be simulated by moving one step up the model hierarchy from RCE. Interactions between convection and a large-scale circulation that is forced by SST anomalies have direct analogues on Earth to the ITCZ, the Walker Circulation, and the Hadley Circulation (in a non-rotating context). The presence of subsiding circulations consistently occurring over regions of cooler SST also allows for the possibility of simulations that include stratocumulus clouds. Initial tests with SAM indicate that the mock-Walker simulation does contain optically and geometrically thicker low-clouds than RCEMIP-I (Stauffer 2023, PhD thesis). The SST gradient, combined with an overturning circulation, also allow for the possibility of modeling the transition between shallow and deep convective clouds. Additional dynamical regimes and cloud types could increase the sensitivity of the model results to details of the parameterization schemes, but we think that RCEMIP-I and a mock-Walker based RCEMIP-II will provide an excellent dataset for investigating many interesting question and will serve as a good reference for further study.

The mock-Walker configuration thus allows investigation on the three themes of RCEMIP (robustness of simulated mean state, response of clouds to warming and climate sensitivity, and dependence of convective aggregation on temperature) in a framework focused on cloud-circulation coupling that is one step up the model hierarchy from RCE, satisfying **principle 3.** The characteristic that makes RCE an idealized model configuration is its homogeneous boundary conditions (uniform surface temperature and insolation). Thus to move one step up the model hierarchy from RCE closer to the real world, we need to relax the idealization of the boundary conditions. A prescribed sinusoidal SST pattern as in the proposed mock-Walker set-up achieves this by providing a still idealized but more realistic (i.e., heterogeneous)

boundary condition, while maintaining all other aspects of the original RCEMIP set-up. It brings in *one* of the dynamical instabilities that is present in the real world but was not present in the original RCEMIP – a circulation forced by an SST gradient. This provides a clearer tie to observations than the original RCEMIP simulations with uniform SSTs.

The prescribed SST gradient in the mock-Walker simulations provides forcing for low-level convergence towards the warmest SSTs and will drive a large-scale circulation that provides a dynamical constraint on the location and spatial pattern of convection, satisfying **principle 4**. Reviewer #2 argues that the prescribed SST gradient will not provide a constraint on convection and circulation or narrow the intermodel spread. It is possible that we are overly optimistic about the degree to which the prescribed SST gradient will reduce the diversity of simulated climates, but we maintain that a prescribed SST gradient of at least moderate strength will provide a dynamically-forced organizing constraint on the convection and circulation relative to uniform SSTs. The ability of SST gradients to dynamically constrain/organize the convection is clear in our present paper as well as numerous previous studies (e.g. Grabowski et al., 2000; Tompkins, 2001; Bretherton et al., 2006; Lutsko and Cronin, 2018; Silvers and Robinson, 2021). We believe that the extent to which the prescribed SST gradient constrains convection and circulation in an environment of complex interactions between moist convective processes, radiation, and microphysics is a question worthy of investigation across an ensemble of models.

**Rotation:**

We do not feel that it would be possible to satisfy **principle 1** while including rotation. Based on the author's own experience, and the abundant prior literature, f-plane RCE simulations in a limited area CRM domain are quite different from rotating RCE simulations in a GCM (realistic rotation on the sphere with uniform thermal forcing). In our view, f-plane simulations in a limited area domain, while richly explored and yielding valuable insights on many questions, preclude investigation of essential questions about tropical cyclone frequency or genesis rate, since at low-f a single TC is artificially squeezed into the domain size provided and at high-f the number of TCs is controlled by the maximum packing. GCM simulations with uniform rotation have similar issues. Rotating RCE on the sphere would be a promising set-up for an intercomparison about tropical cyclones, but there is no obvious CRM analog other than a global CRM or perhaps a large beta-plane, but the latter would entail a different domain set-up to RCEMIP-I and would be more computationally expensive (in opposition to principle 2). While adding rotation does move up the model hierarchy of complexity from non-rotating RCE (supporting **principle 3**), the themes that would be investigated are likely tropical cyclone-focused. Though such questions are of great interest in general, they are different from the current themes of RCEMIP (opposing principle 3). Rotation allows for additional dynamical interactions that could provide a constraint on convection (principle 4), though it is not clear to what extent. The same sensitivities to microphysical and radiative parameterizations that Reviewer #2 is concerned about in the context of mock-Walker simulations would also likely be present in rotating simulations.

Interactive SST:

Performing simulations with interactive SST involves jumping further up the model hierarchy. While an important step towards the real world, RCE simulations with interactive SSTs have been studied in far less detail than mock-Walker simulations. Slab mixed layer oceans of even relatively shallow depth take many hundreds of days to reach equilibrium (Cronin and Emanuel, 2013). This greatly increases the computational expense, particularly for CRMs (in opposition to **principle 2**). To reach our goal of relaxing the idealization of uniform SSTs, we chose mock-Walker simulations over interactive SSTs as the next step partly for this pragmatic reason, and partly because of our interest in the scientific questions that open up once we have a system with a forced circulation. We are open to revisiting the idea of interactive SSTs if we make it to a phase three of RCEMIP :-)

**Simplified physics:**

Simplified radiation/microphysics schemes was another possible direction for a second phase of RCEMIP. Imposing simplified physics schemes can in principle be done in both CRMs and GCMs, satisfying **principle 1.** However, a simplified microphysics scheme would be significantly more complicated to implement in most models, opposing principle 2. Simplified physics would provide a further constraint on convection (supporting part of principle 3), but would move down the model hierarchy towards more idealization, not less, and, depending on the types of simplifications, could remove some phenomena of interest, such as self-aggregation, as topics of investigation (opposing principle 3). Furthermore, the more the physics is modified to be simpler, the further the models diverge from their parent models. In the case of GCMs, the ability to learn about the comprehensive version of the model from more idealized configurations was a strength of RCEMIP (e.g., Reed et al. 2021). This would be less likely with the use of simplified physics. Simplified physics would likely provide a constraint on convection (supporting **principle 4**). But while we admit that it is more complicated, we prefer a dynamical constraint on convection (as in the mock-Walker set-up) to a constraint provided by removing physical processes (as in simplified physics), because the former is more consistent with how convection is constrained in the real world.

Our view of the value of simplified physics has evolved since we originally suggested in the RCEMIP-I protocol paper and the response to reviewers there that it would be needed. We believe that simplified physics is valuable when trying to isolate the minimal ingredients necessary for a particular physical mechanism, typically within the context of experiments within an individual model. With an individual model, mechanism denial experiments, in which specific mechanisms are methodically removed through targeted simplifications, are also an excellent tool for determining the role of particular processes. We are sure that assessing the sensitivity to dynamical core, radiation scheme, microphysics scheme, boundary layer scheme, convective scheme, and the sensitivity to various parameters in those schemes would likely lead to both interesting and informative results. However, we feel that it is more suitable and tractable to do this in investigations with a single model or related group of models, rather than a large intercomparison.

In a model intercomparison, simplifying some of the physics could be useful for ruling out particular sources of intermodel spread. However, in our opinion the goal of an intercomparison is NOT to constrain models so much that they are forced to agree. When a robust result emerges from an intercomparison *in spite of* great diversity in model physics, this provides much stronger evidence for this behavior than if it is found when the physics has been constrained to be the same. It indicates that in order for this result to emerge, it must be the result of a very fundamental physical mechanism that is *not* dependent on the details of physics parameterizations. In addition, model diversity in an intercomparison provides an opportunity to explain the intermodel spread, not in terms of a particular model detail, but in terms of robust physical mechanisms and theory. For example, Wing and Singh (2023) used zero-buoyancy plume theory to explain the intermodel spread in stability and humidity in RCEMIP.

That being said, we do recognize the reviewer's point that the mock-Walker simulations may develop odd behaviors (stacked overturning circulations, low-frequency variability, etc...) that likely will differ across models and complicate interpretation. We will consider pairing the proposed simulations with simplified radiation runs to aid understanding, but we have not yet figured out how this could be implemented and still satisfy the four guiding principles outlined above. Compared with simplified microphysics, simplified radiation would be easier to implement and the comparison with fully interactive radiation would be instructive. However, we worry in particular about principle 2, as even if they are easy to configure, requesting more simulations may reduce the ability and willingness of groups to participate. We would want to retain the simulations with full physics to satisfy principle 3 and facilitate direct comparison between RCEMIP-I (uniform SST) and RCEMIP-II (prescribed SST gradient) simulations, and so simplified radiation simulations would be in addition to the full physics simulations, rather than instead of them. It is also not clear to us which prescription of radiation would be most appropriate. A fixed cooling rate is easy and would be consistent across models, but could result in a mean cooling rate that is different than in the simulations with interactive radiation which presents its own issues of interpretation. One way to address this is to spatially homogenize the radiative cooling at each level, or prescribe a profile of radiative cooling determined from the horizontal- and time-mean of the simulations with interactive radiation. These options would decouple the radiative cooling from the condensate and moisture and would be internally consistent, avoiding a mean bias. However, the former would still allow time dependence and both options would yield different radiative cooling in each model and may still contain sharp vertical gradients and thus may not remove stacked-overturning cells. Prescribing radiative flux divergence in temperature coordinates seems a bit more complicated to implement, though we admittedly have not tried this before. With any of these choices, we would need to perform additional test simulations, and it is not possible to complete simulations that are directly comparable with the simulations presented in the paper because the computer on which they were performed (Cheyenne) no longer exists. We are also hesitant to commit to including simplified radiation simulations unless we receive assurance from a majority of the now ~30 groups interested in participating in RCEMIP-II that they would be willing to perform these additional simulations.

---

## Author Response (AR1)

Response to Anonymous Referee #1

Our responses are in blue.

This manuscript proposes a second phase of RCEMIP, focused on "mock-Walker" simulations. These simulations include hot/cold patches, which force convection to aggregate over part of the domain, alleviating some of the issues convective aggregation adds to uniform-SST RCE simulations. As well as describing the proposed set-ups, the text also presents some initial results from mock-Walker simulations with a cloud-resolving model (CRM) and a global model (GCM).

The first phase of RCEMIP was very successful, and phase 2 is certainly warranted. I appreciate the authors' attempt to get this going; however, I have some concerns about the proposed set-ups given here:

- the study of mock-Walker simulations is less "mature" than that of RCE simulations were going into RCEMIP-!, which leaves some nagging questions about the set-up. The authors note that for computational reasons the SST gradient will vary slightly across the models for the range of widths proposed here. While the variation is small across models (3% or less for most models), one could equally prescribe a set-up in which \Delta T is adjusted to keep the same gradient across the models. The issue is it is unclear whether it is the gradient or the absolute SSTs which matter most for the climate of mock-Walker simulations. My guess is it's the warm pool SSTs, which would support the current proposal, but to my knowledge this has not been systematically investigated. There is some effort here to explore sensitivity to domain size etc., and it might be a good idea to do a more thorough investigation. Without this, it may be difficult to do an "intermodal comparison" as differences in model climate could be due to model physics or to differences in domain size.

*Maturity of the mock-Walker set-up:*

In designing the protocol for the mock-Walker simulations as RCEMIP-II, we tested different versions of the equation for prescribed SSTs, the choice of delta-SSTs, the CRM domain length and width, maintaining different aspects of the SST pattern across CRMs of slightly different domain length, and how to make the CRM and GCM set-ups consistent. While there are always countless more sensitivity tests that could be done, the protocol was not decided upon lightly and is instead the result of several years of consideration and optimization, as well as engagement of the RCEMIP community. As described in the initial response to reviewers regarding the evolution of phase II, in multiple cases, we performed additional tests and even changed the protocol in response to comments from the RCEMIP community. Therefore, based on the testing presented in this paper in combination with the substantial body of literature on

mock-Walker simulations over past decades, our opinion is that this configuration is both scientifically interesting for RCEMP-II and is better tested than the other potential RCEMIP-II candidate experiments. Further tests in the context of one, or a few models could continue to be performed in parallel with the broader intercomparison, as needed (as is typical for MIPs). While the lengthy period of testing and optimization we have undertaken was necessary to ensure a robust protocol and address questions from the community, further delay in beginning phase II risks losing the momentum we have built.

*SST gradient vs. absolute SSTs:*

Reviewer #1 makes a good point that it is unclear which SST quantity matters most for the climate of mock-Walker simulations. From the perspective of the weak temperature gradient approximation, the absolute SST contrast (maxSST - minSST) and the maximum SST is what ought to matter to the dynamics (Sobel and Bretherton 2000; Bretherton and Sobel 2002). The SST gradient (dSST/dx) might plausibly set horizontal flow speeds, based on Lindzen and Nigam (1987), and if that is the case, the SST laplacian ought to matter for vertical motion and precipitation. Ideally all three of these parameters would be kept fixed across the models, but due to computational limitations on domain size this is not possible. This issue is one that we considered at length, discussed with other members of the RCEMIP community, and tested extensively in preparing the RCEMIP-II protocol.

We discussed two possible options in the paper: (1) enforcing that the wavelength equals the domain length and (2) enforcing the same wavelength (6000 km) regardless of domain length. Option (1) keeps the absolute SST contrast (delta-SST), maximum SST, and mean SST the same but leads to slightly different SST gradients. Option (2) leads to slightly different mean SSTs, a discontinuous SST distribution at the boundaries, and the projection of the prescribed SST forcing onto all scales, introducing substantial noise at higher wavenumbers.

After testing in one CRM, SAM, we elected to go with option (1). Even though this choice could cause differences in the results due to differences in domain length, these differences would result from a physical reason (a different SST gradient) which we felt was preferable to the non-physical artifacts present in option (2). Adjusting the value of delta-SST, as the reviewer suggests, would cause differences in the absolute SST contrast and maximum SSTs. Keeping the absolute SST contrast and maximum SSTs consistent (our chosen option) is the most elegant and the simplest to implement. It is plausibly what matters most for dynamics, precipitation, and clouds based on weak temperature gradient arguments.

The differences in the SST gradient that would result from different domain lengths are small compared to the differences in SST gradient from choosing 0.625 K, 0.75 K, or 1 K as the "weak gradient", for instance. While we cannot rule out that differences in domain size could contribute to differences across models, we expect a small impact. This is supported by the testing we did in SAM, in which our results did not qualitatively depend on the choice of option (1) or option (2). Once all the RCEMIP-II simulations are

completed, we could attempt to determine the influence of domain length on the results by assessing if models with a larger domain length behave systematically differently from those with a smaller domain length. While we acknowledge that it could be difficult to disentangle the relative contributions of the domain length difference and other aspects of model physics and numerics to intermodel differences, this difficulty would also be present if other configurations were selected for RCEMIP-II. Our goal with the RCEMIP-II protocol has been to make the set-up as uniform as possible given computational limitations and to avoid the imposition of non-physical artifacts.

Lindzen, R.S. and Nigam, S. (1987): On the role of sea surface temperature gradients in forcing low-level winds and convergence in the tropics. J. Atmos. Sci., 44, 2418-2438.

Sobel, A.H. and C.S. Bretherton (2000): Modeling tropical precipitation in a single column. J. Climate, 13, 4378-4392.

Bretherton, C.S. and A.H. Sobel (2002): A simple model of a convectively-coupled Walker circulation using the weak temperature gradient approximation. J. Climate, 15, 2907-2920.

We have revised Section 3.3.1 to provide further discussion and justification for our choice.

- I was very surprised by the GCM set-up, which consists of zonal bands of hot and cold SSTs. I haven't seen this set-up before, and don't have a good feel for it. I would have expected the authors to propose uniform SSTs with hot and cold patches on the equator. Mock-Walker simulations confine the convection in x and y; here it is only confined in y. There could be similar issues with convective aggregation within the warm latitude bands as are seen in RCE simulations, and I would also expect large internal variability, which could interfere with the interpretation of model results.

We considered and tested alternate geometries of the SST pattern, which was a subject of both internal discussion and at the RCEMIP breakout session at the 2023 Joint CFMIP-GASS Meeting in Paris. We ultimately elected to utilize zonal bands of hot and cold SSTs to ensure the closest possible correspondence to the set-up in the doubly periodic long-channel CRM domain (chosen to be identical to the RCEMIP-I domain), including the mean SST, maximum SST, and SST gradient. It should be noted that positioning the SST warm patches periodically along the equator of GCMs would either necessitate changing the radius of the globe or abandoning the idea of maintaining a physical wavelength of approximately 6000 km for comparison to the CRMs. We have chosen to vary SSTs in only one dimension partly for simplicity but also partly because the deep tropics on Earth can be approximated in this way. If the SST warm patches were shifted by 90 degrees as suggested by the reviewer, interpretation of the variability would be complicated due to a changing width of the SST pattern as the poles are

approached. Furthermore, given the double periodicity, the CRM domain should be conceptualized as being infinitely repeated in both dimensions, which would then (other than the sphericity) make it analogous to the GCM set-up. This is demonstrated in the figure below (and now added to Figure 1 in the paper), in which the SSTs in the GCM are shown on the left and the SSTs in the CRM are shown on the right, in which the CRM has been rotated and tiled 24 times in one dimension and 3 times in the other, to emulate the GCM domain. Note that since these simulations are non-rotating, there is no physical distinction between x,y, latitude, or longitude.

Since one of the core principles of RCEMIP is to be able to compare limited area CRMs and GCMs, we chose to confine the warm SSTs in only one direction in both model types. The GCM set-up is similar to that used in Müller and Hohenegger (2020), which also utilized zonally homogenous but meridionally varying SSTs. Convective self-aggregation within the warm latitude bands does occur, as the reviewer suggests. Zonal contraction of convection was seen in Müller and Hohenegger (2020) and in our test simulations with CAM.  This is one of the aspects of this set up that we are excited to explore as we think it corresponds well to observed convective aggregation. While it could complicate interpretation, it will also be interesting to see how the degree to which this "intrinsic" self-aggregation emerges on top of the forced convergence varies across models.  The ability to study self-aggregation both in the context of SST gradients and constant SST is one of our motivations for this particular setup and creates an additional connecting point both with the RCEMIP-I experiments and the tropical oceans of Earth.

Müller, S.K.M. and Hohenegger, C. (2020): Self-aggregation of convection in spatially-varying sea surface temperatures. J. Adv. Model. Earth Syst., 12, e2019MS001698.

We have revised section 3.2.2 to clarify this issue.

[Figure]

For some inspiration, the authors might want to check out the following papers from Dennis Hartmann's group:

Hartmann, D. L., and B. D. Dygert (2022), Global Radiative Convective Equilibrium With a Slab Ocean: SST Contrast, Sensitivity and Circulation, Journal of Geophysical Research: Atmospheres, 127(12), e2021JD036,400.

Larson, K., and D. L. Hartmann (2003), Interactions among cloud, water vapor, radiation, and large-scale circulation in the tropical climate. part ii: Sensitivity to spatial gradients of sea surface temperature., Journal of Climate, 16(10), 1441–1455.

Thank you for pointing us to these papers. We now reference them in the manuscript.

I mostly focused on the model set-ups, but also have a number of small comments/typos:

L13: SST's -> SSTs. I also think this should be "the same three mean SSTs". The sentence is a bit confusing because it mentions 5 simulations covering 3 mean SSTs and 3 SST gradients. I understand it after reading the paper, but for the abstract the authors might want to word things more carefully.

We have revised the sentence to specify the required simulations more explicitly. It now reads "RCEMIP-II will consist of five required simulations: three simulations with the same three mean SSTs as in RCEMIP-I but with an SST gradient, as well as two additional simulations at one of the mean SSTs with different values of the SST gradients." (Lines 13-14).

L14: Sentence starting "Under weak SST gradients" -> maybe a personal preference, but the commas feel misplaced in this sentence to me. I would put the first one after "domain" and delete the third one.

We have split the sentence into two: "Under weak SST gradients, unforced self-aggregation emerges across the entire domain. As the SST gradient increases, the convective region narrows and is confined to the warmest SSTs." (Lines 15-17).

L20: Suggest: "two; however, we also"

We have made the correction as suggested (Line 21).

L26: "Operational models" to me signals forecast models, not GCMs. Also should be "such as"

We have adjusted the end of the first paragraph for additional clarity (start at Line 27).

L28: Should this be "uncertainty"? This sentence also makes it seem like the tropics are the only source of bias/uncertainty in climate projections.

We have adjusted the end of the first paragraph for additional clarity (start at Line 27).

L30: Not sure how Held, 2005 is relevant here. I also don't know what it means to"effectively use" an Earth System model.

We have changed the sentence to "...complicated by a host of scale interactions that are challenging to effectively represent in comprehensive Earth system models." and removed the reference to Held (2005). (Lines 30-31).

L45 and afterwards: I encourage the authors to provide a bit more information about these robust responses. E.g., what is the response of deep convective clouds to warming? How is the existence of self-aggregation robust? Etc. These results are important motivations for the present paper and should be discussed in detail.

We have tweaked the description of the robust RCEMIP-I results to provide more of this information, but elect against adding more detailed discussion, which can be found within the referenced papers. (Line 47-53).

L53: is insolation the only forcing? Surely greenhouse gases also? Also, in this paragraph "sensitive"."sensitivities" is used a lot. Suggest re-writing.

Here we are referring only to the homogeneous thermal forcings at the boundaries. We have revised the sentence to state this more clearly: "RCEMIP-I prescribed homogeneous thermal forcings at the boundaries, which consisted of uniform sea surface temperature (SST) and insolation." (Line 58-59).

We have also reworded the third sentence of the paragraph to reduce usages of "sensitivities". It now reads: "The divergent behavior in RCEMIP-I reveals dependencies on representations of convection, microphysics, turbulence, and dynamical cores that may have been masked in other intercomparisons by dynamical constraints." (Line 60-63).

L57: "proposal" -> "propose"

Corrected. (Line 62).

L58: delete "now"

Corrected. (Line 63).

L71: I found it odd the authors wrote "compared to RCE", implying that mock-Walker simulations are not in RCE even though they are the basis of RCEMIP-II. Maybe the authors could clarify how to think about the set-ups in relation to RCE?

The mock-Walker simulations are not strictly in RCE, since they include a heterogeneous boundary condition and an externally forced large-scale circulation. So this intercomparison might be more accurately described as "mock-Walker MIP". But, we choose to instead consider them as a Phase II of RCEMIP rather than a new MIP since we investigate the same scientific themes using an identical model configuration (with the exception of the SST boundary condition) to that of RCEMIP-I and we can use the RCEMIP-I results as a sort of control case. In a sense, we are extending RCEMIP by relaxing the assumption of strict RCE.

We have adjusted our description to clarify this on lines 64, 75-79, and in section 2.

L84: Suggest replacing "provided" with "described"

Corrected. (Line 89).

L90: Sentence starting "In considering", should it be "(1) how strikingly large"…and "(2) how strong"?

Corrected. (Line 105-106)

L101: what does "its" refer to here? In general, I found this paragraph a little muddled with frequent use of "coupling", "clouds" and "circulation". Suggest streamlining.

"Its" was referring to convective aggregation (convective aggregation and its role in climate). We have rewritten the paragraph to clarify and expanded section 2. (Lines 101-104, and section 2 in general).

L108: Another personal preference, but I feel writing should stand on its own, and discourage the authors' use of bold text.

We have removed the bold text. (Lines 176-178).

L138: I would encourage the use of LES if anyone has the resources!

We have added this. (Lines 198-200).

L163: I would capitalize "Cartesian"

We have corrected this. (Line 226).

L165: Suggest: "between the maximum and minimum SST".

We have corrected this.  (Line 228).

L180: Why did you switch from 2\pi to 360 degrees here? (Also equation 3)

Because we wrote the equation as a function of latitude in degrees, whereas equation (1) is a function of x in distance.

Table 1: I encourage the authors to have 5 "core" experiments (listed here) and 4 additional experiments (295/0.625,295/2.5, 305/0.625,305/2.5). It would be great if some groups ran more experiments

We have added optional experiments to the table as suggested.

L198: Should this be: "there is to be no diurnal…", to keep the tense consistent?

We have corrected this. (Line 284).

L228: The authors should refer to the (large) literature on the pattern effect in this paragraph, which might also provide some insight into what to expect in terms of cloud feedbacks.

We have added discussion of the pattern effect and relevant references to Section 2 (in particular, see lines 123-145) and have revised our discussion of it in Section 3.5

L252: Have the simulations been updated, or has the model code?

The model code has been updated and the simulations shown in the paper were run using that updated model code. We have revised the sentence to make clear that the model code has been updated. (Line 339).

Figure 3: This doesn't look much like a Walker Circulation to me…It might be interesting to plot the overturning stream function to get a better feel for what the flow is doing.

In this simulation (MW_300dT1p25) the precipitation is located in the region of the warmest SSTs and their periphery, but absent from the coldest SSTs. The convection is more widespread than one might expect because the SST gradient is apparently not strong enough to constrain it to only the warmest SSTs.

We have added plots of the streamfunction in Figure 2, 8, and 9, and discussion of the circulation as suggested.

L315: Suggest re-writing sentence starting "This suggests" (remove "doesn't "care"").

We have rewritten the sentence to say that the spatial structure of convection is not influenced by the SST pattern. (Lines 412-416).

L329: This is a good example of my concern that mock-Walker simulations are not ready for this kind of comparison. This is a required simulation, but it takes longer than 200 days (the prescribed simulation length) to reach equilibrium. Maybe in other models all simulations need longer to equilibrate.

The presence of low-frequency oscillations and thus the need for a long simulation time is one of the elements that initially concerned us, since 200 days is already a significant numerical cost for some CRMs. Through communication with colleagues, 200 days seems to be sufficient for several different models. But since we can't rule out that other models might need longer, the protocol dictates that CRMs perform simulations for *at least* 200 days (Section 3.2.1). We have revised this to specify that longer simulations may be needed to reach equilibrium ("The simulations are to be performed for at least 200 days, or longer if needed to reach equilibrium." Line 452-453).

L433: "the behavior of \Delta SST"?

We have corrected this. (Line 556).

L461: Should this be "SSTs"?

Corrected, (Line 585).

Response to Anonymous Referee #2

Our responses are in blue.

Wing et al. have suggested a protocol for the second phase of the Radiative Convective Equilibrium Model Intercomparison Project (RCEMIP), following on from the Phase One experiments which were motivated in their 2017 paper, also in GMD. My general thoughts are that the protocol is poorly motivated, and neglects many of the (arguably more promising) avenues in which a Phase II of RCEMIP could explore in favor of a relatively untested setup, the 'mock-Walker' simulation.

In our initial response to reviewer comments, we provided more background on the evolution of Phase II of RCEMIP and how the mock-Walker experimental design came to be. We have revised the manuscript to provide better motivation for the choice of mock-Walker simulations as Phase II of RCEMIP. We believe that our initial response, revised manuscript, and further point-by-point response here provide strong justification for the proposed protocol.

We disagree with the reviewer's characterization of mock-Walker simulations as a relatively untested setup. In designing the protocol for the mock-Walker simulations as RCEMIP-II, we tested different versions of the equation for prescribed SSTs, the choice of delta-SSTs, the CRM domain length and width, maintaining different aspects of the SST pattern across CRMs of slightly different domain length, and how to make the CRM and GCM set-ups consistent. While there are always countless more sensitivity tests that could be done, the protocol was not decided upon lightly and is instead the result of several years of consideration and optimization. As described in our initial response to reviewer comments regarding the evolution of phase II, in multiple cases, we performed additional tests and even changed the protocol in response to comments from the RCEMIP community. Therefore, we feel like the testing presented in this paper in combination with the substantial body of literature on mock-Walker simulations over past decades makes it appropriate to do a mock-Walker intercomparison. Further tests in the context of one or a few models could continue to be performed in parallel with the broader intercomparison, as needed (as is common for MIPs). While there are certain behaviors that emerge in the mock-Walker simulations for which a comprehensive physical understanding is currently lacking, we feel that RCEMIP-II is an opportunity to explore those behaviors across a wide range of models.

In my opinion, the following issues mean that the current protocol is unfit for publication in its current form.

**Issue #1: approach to constraining model diversity**

The 2017 protocol paper (RCEMIP-I) outlined a series of small and large domain CRM/LES simulations, in addition to suggested runs with global models and single-column models. One of the key takeaways from RCEMIP-I is that there is substantial diversity in the simulated RCE state even in the absence of convective self-aggregation.

This point is acknowledged in the current manuscript, which notes that: "While several robust results emerged across the spectrum of models that participated in the first phase of RCEMIP (RCEMIP-I), two points that stand out are (1) the strikingly large diversity in simulated climate states and (2) the strong imprint of convective self-aggregation on the climate state." This diversity in RCEMIP-I is also acknowledged to be driven by "representations of convection, microphysics, turbulence, and dynamical cores".

However, the current manuscript does little to tackle the question of inter-model diversity in the RCE state and instead primarily focuses on point (2) stating that "...the wide range in the degree of self-aggregation and the lack of consensus in its temperature dependence is a barrier to understanding."

In our original submission, we did not adequately explain the principles on which the experimental design for Phase II of RCEMIP was based. We have tried to correct that in our initial response to reviewers, our further response here, and in the revised manuscript. While choosing an experimental design that might narrow inter-model diversity was one of our initial goals, it is not our only goal.

The experimental design for RCEMIP-II follows four principles, in the spirit of the design of RCEMIP-I:

1) The ability to directly compare limited-area models with explicit convection and global climate models
2) Ease of implementation, to encourage the broadest possible participation
3) Continued investigation of the three themes of RCEMIP (robustness of the mean state, response of clouds to warming and climate sensitivity, dependence of convective self-aggregation on temperature), *while moving a step up the model hierarchy of complexity*
4) Providing an external constraint on convection.

This emphasis is counter to the vision laid out in the RCEMIP-I protocol (and contrary to the response given to the RCEMIP-I reviewers, quotations from which will be cited below). Originally, RCEMIP-I was presented as an opportunity to explore inter-model diversity, with RCEMIP-II being an opportunity to try to understand/narrow that diversity through the use of simplified radiation/microphysics schemes.

We would argue that RCEMIP-I was presented and executed as an opportunity to explore questions about tropical clouds, convective aggregation, and climate. Inter-model diversity was a component of that but not the sole target of investigation.

In the RCEMIP-I protocol paper (Wing et al., 2018), we suggested several different avenues as possible directions for future iterations of RCEMIP. Narrowing inter-model diversity through the use of simplified radiation/microphysics schemes was one possible future direction, but not the only one. Further, our views regarding the best choice for a second phase of RCEMIP have evolved since 2018. The selection of mock-Walker simulations for RCEMIP-II simply reflects a practical direction that both our conversations with colleagues and our work over the past few years has naturally taken.

We disagree that the choice of mock-Walker simulations as RCEMIP-II is counter to the vision laid out in the RCEMIP-I protocol paper. Instead, we feel that it follows the philosophy of our design of RCEMIP-I; in particular, principles 1 and 2 above, which were also key elements of RCEMIP-I.

For example, many of the reviewers for RCEMIP-I strongly suggested the use of simplified radiation and/or microphysics schemes as a way to better understand the diversity of RCE states and their response to warming. A list of these is presented below:

(Isaac Held): "*I would strongly encourage you to reconsider and ask groups to run with a standard microphysical mechanism in addition to their model's microphysics. Otherwise, there is a good chance that the diversity of simulations will be dominated by the diversity of microphysical assumptions. (For example, we know that in RCEs assumptions about ice fall-speeds will exert a strong control on the cirrus climate.)*"

(Levi Silvers): "*It is not essential, but I think it would be useful to be more precise about a second set of non-required (Tier 2) experiments. This could be written in such a way that modeling centers wishing to participate with minimal effort are not thus discouraged from participating, but that more ambitious modeling centers or individuals could clearly push farther into the project in a coordinated way. My suggestions for further experiments would be: 1. Rotating RCE 2. GCMs in RCE mode with convective parameterization turned off 3. RCE with cloud RCE off (COOKIE type experiments) 4. Kessler physics across the hierarchy of models*"

(Anonymous reviewer IV): "*...the resulting equilibrium state and the clustering may look very different in the different CRMs. It will thus be very difficult to compare the different models and to identify the root for the differences (radiation scheme, microphysics parametrizations, . . .). An even simpler setup for the models could therefore be useful to identify, which schemes are responsible for the differences. As suggested by the*

*authors and brought forward by Jeevanjee et al. (2017) a simplified microphysics scheme could be one option. A further option could be to simplify the longwave radiative cooling, as e.g. described in Muller and Bony (2015)."*

(Nadir Jeevanjee): *"As advocated for by Isaac and other reviewers, there is also interest here in using simplified (Kessler) microphysics in our RCE setup. Such a scheme already exists in development branches of our code."*

Furthermore, this was explicitly stated as a potential target for the second phase of RCEMIP (Sec 6.2 of 2017 protocol paper):

*"Additional simulations could be performed to assess the sensitivity to dynamical core, radiation scheme, microphysics scheme, boundary layer scheme, convective scheme (in the case of models with parameterized convection), and the sensitivity to various parameters in those schemes (such as the entrainment parameter in a convective scheme)."*

Simulations to assess the sensitivity to different physical schemes were in fact performed as part of RCEMIP-I (Wing et al. 2020).  For example, the UKMO Idealized Model v11.0 was run as a CRM in its RA1-T configuration, its RA1-T configuration without a cloud scheme, and in a configuration that instead used CASIM microphysics. Comparing UKMO-RA1-T and UKMO-CASIM permits one to investigate the sensitivity to microphysics scheme within a single model environment. WRF-GCM was run with six different convective parameterizations, so that sensitivity to convective scheme could be evaluated.  Reed et al. (2021) evaluated the differences between CAM5 and CAM6, which have different boundary layer, shallow convection, and cloud macrophysics schemes (but the same deep convection scheme). As indicated by Table 2 in the manuscript, there are already similar plans in place for RCEMIP-II; for example, SAM will be run using three different microphysics schemes.

Additionally, in the reply to Reviewer IV of RCEMIP-I (similar comments exist in the reply to other reviewers):

*"We agree that large differences could result from differences in physical parameterizations. However, we think that it is useful to first determine the full range of RCE simulations and then proceed to test the parameterization sensitivity by imposing a simple microphysics scheme on all models in the second phase of RCEMIP."*

From these statements it is clear that simplified experiments were anticipated to be necessary/useful by reviewers for understanding inter-model diversity in the simulated RCE state. This was also acknowledged by the authors and explicitly suggested as a route forward for Phase II of RCEMIP. The need for idealized experiments seems even

more necessary now that the RCEMIP-I simulations have demonstrated such an extreme diversity in the simulated RCE state (perhaps even larger than expected).

Our view of the value of simplified physics has evolved since we originally suggested in the RCEMIP-I protocol paper and the response to reviewers there that it would be needed. We believe that simplified physics is valuable when trying to isolate the minimal ingredients necessary for a particular physical mechanism, typically within the context of experiments within an individual model. Mechanism denial experiments, in which specific mechanisms are methodically removed from an individual model through targeted simplifications, are also an excellent tool for determining the role of particular processes. We are sure that a comprehensive assessment of the sensitivity to dynamical core, radiation scheme, microphysics scheme, boundary layer scheme, convective scheme, and the sensitivity to various parameters in those schemes would likely lead to both interesting and informative results. However, we feel that it is more suitable and tractable to do this in investigations with a single model or related group of models, rather than a large intercomparison.

In a model intercomparison, simplifying some of the physics could be useful for ruling out particular sources of intermodel spread. However, in our opinion the goal of an intercomparison is NOT to constrain models so much that they are forced to agree. When a robust result emerges from an intercomparison *in spite of* great diversity in model physics, this provides much stronger evidence for this behavior than if it is found when the physics has been constrained to be the same. It indicates that in order for this result to emerge, it must be the result of a very fundamental physical mechanism that is *not* dependent on the details of physics parameterizations. In addition, model diversity in an intercomparison provides an opportunity to explain the intermodel spread, not in terms of a particular model detail, but in terms of robust physical mechanisms and theory. For example, Wing and Singh (2023) used zero-buoyancy plume theory to explain the intermodel spread in stability and humidity in RCEMIP-I.

Furthermore, since RCEMIP-I it has been fairly well-established that relevant quantities such as high cloud feedbacks are extremely sensitive to microphysics. For example, both Wing et al. (2020) and Stauffer and Wing (2022) demonstrated that about one-third of models do not exhibit an 'iris' effect, and in fact have an *increase* in high cloud fraction under warming despite a decrease in clear-sky divergence at the anvil level. The disconnect between radiatively driven divergence and high-cloud fraction explored in Seeley et al (2019), Beydoun et al. (2021) and Jeevanjee (2023) (among others), and one key takeaway from those papers is that the lifetime of detrained cloud condensate is a crucial determinant of high-cloud fraction and the anvil cloud feedback.

Stauffer and Wing (2022) found that the majority of models exhibited a decrease in anvil cloud fraction in concert with radiatively-driven divergence. We interpret this finding, which occurs despite great model diversity, as strong support for radiatively-driven divergence as a first order control on anvil coverage. Stauffer and Wing (2022) acknowledges that this breaks down in some models, which indicates that other factors,

such as the lifetime of detrained condensate, also contribute.  Thus while we agree with the reviewer that changes in high clouds are sensitive to microphysics, the diversity of microphysics in RCEMIP-I can also be considered as a benefit, since a response that emerges across most models in that situation must be driven by a mechanism that is *not* contingent on the details of the microphysics.

It is thus extremely confusing that the authors do not explore the possibility of using simplified microphysics schemes, or at least require that all models output microphysical process rates and (where relevant) particle size distributions, which would be trivial for most models, and allow for a better understanding of this crucial feedback.

We have added microphysical variables as optional output in Table A5 as suggested.

While implementing idealized microphysical schemes is a potential burden on modeling centers (though some models such as FV3 already have the option to use a Kessler-style microphysics package), it is quite simple to replace interactive radiative transfer with a prescribed radiative cooling rate. For example, Paulius and Garner (2006) use a simple 1.5K/day radiative cooling rate up to a fixed tropopause temperature. Such an approach is already widely used in the RCE literature. Additionally, a slightly more realistic approach, which better captures the effect of warming on radiative fluxes, is to prescribe the radiative flux divergence in temperature coordinates ($\partial_T F$) as being linear in temperature (following Eq. 2 of Jeevanjee and Zhou (JAMES, 2022)). Such an approach was also used by Seeley and Wordsworth (2023).

As described in more detail below, we considered adding additional fixed radiation simulations, but ultimately decided not to request these simulations or provide a protocol for them at this time. We will continue to investigate their feasibility and optimal design for inclusion in future work. This is discussed in the revised paper, see the revised section 2 and Lines 576-588.

My point here is not simply that we should revisit Phase One experiments, for which simplified runs would be helpful, but also that runs with simplified physics will likely be *even more necessary* in RCEMIP-II if the authors do intend on using 'mock-Walker' simulations. This is because it is now well-established (although not cited in the protocol paper...) that the 'mock-Walker' simulations are prone to develop stacked overturning circulations (e.g. Grabowski et al., 2000; Yano et al., 2002; Larson and Hartmann, 2003; Liu and Moncrieff , 2008; Silvers and Robinson, 2021), due to the interactions between radiation and detrained condensate/water vapor (e.g. Nuijens and Emanuel, 2018; Sokol and Hartmann, 2022; Lutsko and Cronin, 2023). An example of which, from experiments with the ICON model, is shown below (domain is ~3000x100km, with dx=2km, but similar results are obtained for the standard RCE_large setup):

[Figure]

Figure 1: Time-averaged streamfunction (red=clockwise motion; blue=anticlockwise motion) and cloud liquid water content in our control simulation, driven by a zonal SST gradient between the centre and edges of the domain.

As shown in Fig. 1, the ICON model exhibits a pronounced 'triple-cell' structure, with three stacked overturning circulations. This structure differs between models, with some simulating a triple-cell structure (ICON), others a double-cell (e.g. Silvers and Robinson, 2021). Furthermore, the onset of these stacked overturning structures is itself dependent on temperature (Lutsko and Cronin, 2018, 2023). In both their 2018 experiments (2D CRM) and in their 2023 experiments (3D CRM), Lutsko and Cronin found that the circulation transitions from a single- to double-cell structure at mean SSTs of ~300K. Although this number is likely model-dependent, it is worrying that in their SAM simulations the transition occurred right in the middle of the SSTs considered as part of the Phase Two RCEMIP protocol.

We have added figures of streamfunction to the paper (Figure 2, Figure 8 and Figure 9) and added discussion of the circulation in Section 4.1 and 4.2. While we do see evidence of stacked overturning circulations in our test simulations for the SAM cases, but not for the CAM cases (Figure 9), in SAM (Figure 8), they are not as prominent as in the example in ICON provided above by the reviewer. Our simulations instead depict a predominant deep circulation with a shallow component. Thus the structure of the circulation is likely to be model dependent. While this could complicate interpretation, RCEMIP-II also presents an opportunity to study the nature of these circulations across a wide variety of models. RCEMIP-II can also serve as a reference point for future studies that include simplified experiments to study the underlying physics of these stacked cells.

This model- and SST-dependent transition between different stacked circulation structures will certainly complicate analysis and interpretation of the proposed RCEMIP-II experiments. In light of this I strongly disagree with the statement on L103 that mock-Walker simulations will "...reduce the diversity of simulated climates and provide a clearer tie to observations..."

We admit that we might have been a bit too optimistic about how much the mock-Walker simulations would reduce the diversity of simulated climates. However, we still expect, and find in our test simulations, that the prescribed SST gradient drives a large-scale circulation that provides a partial, dynamical constraint on the structure of convection compared to strict RCE with uniform SST. The extent to which it provide this constraint in an environment of complex interactions between moist convective processes, radiation, and microphysics will be a subject of investigation across the RCEMIP-II ensemble.

Even if the mock-Walker simulations do not perfectly reproduce observed tropical circulations, interactions between convection and a large-scale circulation that is forced by SST anomalies have direct analogous on Earth to the ITCZ, the Walker Circulation, and the Hadley Circulation (in a non-rotating context). This is a clear connection point to observations, and it is certainly clearer than strict RCE which has uniform SSTs. If different models produce different responses to the same SST gradient, we find this to be an interesting target of scientific investigation.

We have revised Section 2 as well as Section 1 (see e.g., line 76) to make the connection to observations clearer as well as soften our expectation that the mock-Walker simulations will reduce the diversity of simulated climates.

Previous studies have shown that the stacked-overturning cells can be effectively suppressed by prescribing uniform (or otherwise simplified) radiative cooling profiles (e.g., Grabowski et al., 2000; Wofsy and Kuang, 2012; Sokol and Hartmann, 2022; Lutsko and Cronin, 2023). This is effective at suppressing these features because they are largely driven by sharp vertical gradients in the longwave cooling rate, caused by sharp gradients in moisture near the melting line. These moisture gradients are themselves associated with complex interactions between convection, microphysics and radiation; near the melting line, evaporation of detrained condensate and melting of ice form a stable layer which promotes further detrainment of condensate, whose enhanced radiative appears to draw out more condensate (Nuijens and Emanuel, 2018; Sokol and Hartmann, 2022).

Overall, one of the most salient features of mock-Walker circulations is their stacked overturning circulations and the elevated moist layers associated with them. Both of these appear to be governed by complex interactions between microphysics (with both

the liquid and ice phase playing a key role), convection and radiation. Hence, it seems highly unlikely that these mock-Walker simulations will "provide a constraint on convection and circulation" or "narrow the intermodel spread", as hypothesized in the protocol.

I find it somewhat unlikely the authors will give up on mock-Walker simulations altogether, but to claim (as they do in the manuscript) that they will narrow model diversity is quite naïve.

As discussed in our response on the previous page, the SST gradient ought to provide through its forced large-scale circulation at least a partial dynamical constraint on convection. If it fails to do so in some models, this is scientifically interesting. The constraint might be weak when the SST gradient is weak, but a strong SST gradient may be able to narrow model diversity in certain quantities. Perhaps our hypothesis that the mock-Walker simulations would narrow model diversity (relative to strict RCE) was too naïve, so we have revised Section 2 as well as Section 1 (see e.g., line 76) to soften our expectation that the mock-Walker simulations will reduce the diversity of simulated climates.

If they do insist on using mock-Walker simulations (see point #2, below for more on this), then it is highly recommended that they pair such simulations with simplified radiation/microphysics runs to help narrow some of the model diversity and aid interpretation and understanding of the results. If it is deemed too complex to impose a uniform microphysics scheme, then models should instead be required to output comprehensive microphysical diagnostics.

We have carefully considered the reviewers suggestion to pair the mock-Walker simulations with simplified physics runs. We agree that the mock-Walker simulations may develop interesting and complex phenomena (stacked overturning circulations, low-frequency variability, etc…) that likely will differ across models and complicate interpretation.  These kinds of results further motivate the use of the mock-Walker simulations in a MIP and will provide an excellent base from which further and more focused experiments can be conducted by individual groups.

Given the diversity of models and model types in RCEMIP, implementing idealized microphysics schemes across the large range of participating models is not feasible in our opinion. Replacing interactive radiation with a prescribed cooling rate is more tractable, and beyond narrowing model diversity, the comparison between the interactive and fixed radiation could be instructive as a mechanism denial experiment. Instituting a vertically-uniform cooling rate decreasing to zero near the tropopause is likely the easiest option to implement, but even this is not as straightforward as it seems. Some models (particularly GCMs) also take surface downwelling radiative

fluxes as input into their surface layer scheme, so this would also have to be specified. In addition to the issue of the effect of warming on radiative fluxes mentioned by the reviewer above, there is also the issue that the mean radiative cooling would likely be different between the interactive and fixed radiation simulations. This introduces its own issues of interpretation.

However, even if fixed radiation simulations are easy to configure, requesting more simulations reduces the ability and willingness of groups to participate. We'd like to point out that the participants in RCEMIP are in many cases *not* the developers of the model they use, nor are they necessarily based at the modeling center that maintains that model. Therefore, any experimental design that requires substantial code changes greatly reduces the likelihood that it will be adopted.

We also need to retain the simulations with full physics to facilitate investigation of the RCEMIP themes and allow for direct comparison between RCEMIP-I (uniform SST) and RCEMIP-II (prescribed SST gradient) simulations. Requiring both would double the number of simulations expected which is too demanding and would limit participation. We considered suggesting vertically-uniform radiation simulations as additional *optional* simulations, but are not able to include this in the protocol without thorough testing of the set-up. Unfortunately, we also do not have the ability to perform test simulations without re-running all our simulations. For fair comparison, they should be performed on the same computing system, and the machine on which our previous simulations were performed (Cheyenne) is no longer operational. Re-running all our simulations and new fixed radiation test simulations is prohibitive, both in terms of computational expense and time. We have already undertaken a lengthy period of testing and optimization and further delay in beginning RCEMIP-II risks losing the momentum we have built.

We will continue to investigate the feasibility and optimal design for inclusion of fixed radiation simulations in future work. This is discussed in the revised paper, see Lines 588-601.

As evidence of the momentum we have built, and support for the proposed mock-Walker simulations, we note that at the time of the submission of this manuscript in December 2023, 17 people in addition to the authors had expressed enthusiasm about the project and their willingness to contribute simulations. Since then, additional people who would contribute 7 additional models have expressed interest in participating. We have updated Table 2 to reflect these new participants.

**Issue #2: the choice of mock-Walker simulations over other options**

The RCEMIP-I protocol named a number of possible routes forward for Phase Two. For example, introducing a mixed-layer ocean to ensure a closed surface energy budget, or

introducing rotation (f-plane / beta-plane / full spherical geometry) in order to simulate tropical cyclones and equatorial waves. Both of these options have been explored in single-model studies prior to and following on from RCEMIP-I, and thus it is confusing why the authors decided to use the mock-Walker setup instead of these alternative approaches. For example, investigating tropical cyclones in f-plane simulations has been richly explored in a number of single-model papers, but each with differing setups. This seems like a natural avenue for RCEMIP, especially given that tropical cyclones are an incredibly impactful phenomena which we frequently observe in the real-world, as opposed to the "stacked" overturning circulations of mock-Walker cells (which have no obvious observational analogue).

During the RCEMIP breakout session at CFMIP 2023, a number of participants expressed interest in rotating RCE experiments. It is thus extremely strange to me that the current protocol paper does not justify its emphasis on mock-Walker simulations and argue for their benefit *in comparison to* rotating RCE (or indeed other configurations suggested by the RCEMIP Phase One paper).

We have added to the paper a brief description of why we do not propose alternate options, like simple physics, rotating RCE, or mixed-layer ocean for RCEMIP-II (Section 2, Lines 154-171, as well as Section 5, Lines 588-601). This does not preclude such experiments from being performed by individual models or small groups of models or being led as a full intercomparison by other parties as an offshoot of RCEMIP. We'd also like to point out that the mock-Walker set-up has also been explored in many single-model studies prior to and after RCEMIP-I, including some referenced by the reviewer.

Here we provide a longer explanation, copied from the initial response to reviewers:

*Rotation:*

We do not feel that it would be possible to satisfy **principle 1** while including rotation. Based on the author's own experience, and the abundant prior literature, f-plane RCE simulations in a limited area CRM domain are quite different from rotating RCE simulations in a GCM (realistic rotation on the sphere with uniform thermal forcing). In our view, f-plane simulations in a limited area domain, while richly explored and yielding valuable insights on many questions, preclude investigation of essential questions about tropical cyclone frequency or genesis rate, since at low-f a single TC is artificially squeezed into the domain size provided and at high-f the number of TCs is controlled by the maximum packing. GCM simulations with uniform rotation have similar issues. Rotating RCE on the sphere would be a promising set-up for an intercomparison about tropical cyclones, but there is no obvious CRM analog other than a global CRM or perhaps a large beta-plane, but the latter would entail a different domain set-up to RCEMIP-I and would be more computationally expensive (in opposition to **principle 2**).

While adding rotation does move up the model hierarchy of complexity from non-rotating RCE (supporting **principle 3**), the themes that would be investigated are likely tropical cyclone-focused. Though such questions are of great interest in general, they are different from the current themes of RCEMIP (opposing **principle 3**). Rotation allows for additional dynamical interactions that could provide a constraint on convection (**principle 4**), though it is not clear to what extent. The same sensitivities to microphysical and radiative parameterizations that Reviewer #2 is concerned about in the context of mock-Walker simulations would also likely be present in rotating simulations. During the process of developing the protocol for RCEMIP-II, two of the authors did research (Silvers et al., 2024; GRL; https://doi. org/10.1029/2023GL105850) with both the CAM5 and CAM6 models with realistic (Earth-like) rotation and used the experiments from RCEMIP-I as the control experiments with which to compare the rotating cases. As stated above, this work did produce interesting results and helped us to better understand the impact of rotation on the hydrologic cycle, but it also strengthened our conclusion that at this time mock-Walker simulations are better suited as the core of RCEMIP-II.

*Interactive SSTs:*

Performing simulations with interactive SSTs involves jumping further up the model hierarchy. While an important step towards the real world, RCE simulations with interactive SSTs have been studied in far less detail than mock-Walker simulations. Slab mixed layer oceans of even relatively shallow depth take many hundreds of days to reach equilibrium (Cronin and Emanuel, 2013). This greatly increases the computational expense, particularly for CRMs (in opposition to **principle 2**). To reach our goal of relaxing the idealization of uniform SSTs, we chose mock-Walker simulations over interactive SSTs as the next step partly for this pragmatic reason, and partly because of our interest in the scientific questions that open up once we have a system with a forced circulation. We are open to revisiting the idea of interactive SSTs if we make it to a phase three of RCEMIP (but also reserve the right to consider something else).

*Simplified physics:*

Simplified radiation/microphysics schemes was another possible direction for a second phase of RCEMIP. Imposing simplified physics schemes can in principle be done in both CRMs and GCMs, satisfying **principle 1.** However, a simplified microphysics scheme would be significantly more complicated to implement in most models, opposing **principle 2**. Simplified physics would provide a further constraint on convection (supporting part of **principle 3**), but would move *down* the model hierarchy towards more idealization, not less, and, depending on the types of simplifications, could remove some phenomena of interest, such as self-aggregation, as topics of investigation (opposing **principle** 3). Furthermore, the more the physics is modified to be simpler, the further the models diverge from their parent models. In the case of GCMs, the ability to learn about the comprehensive version of the model from more idealized configurations was a strength of RCEMIP (e.g., Reed et al. 2021). This would be less likely with the use of simplified physics. Simplified physics would likely provide a constraint on

convection (supporting **principle 4**). But while we admit that it is more complicated, we prefer a dynamical constraint on convection (as in the mock-Walker set-up) to a constraint provided by removing physical processes (as in simplified physics), because the former is more consistent with how convection is constrained in the real world.

---

## Author Response (AR2)

Technical corrections:

* In your references, please update the reference to Stauffer and Wing (2023) (which currently shows as being in review, though I see that it appears to now be published)

We have updated the reference to Stauffer and Wing (2023). Stauffer and Wing (2024) is still in review, so we have removed the references to this paper.

* If possible, please provide DOI versions of the exact CESM and SAM versions used in this paper (and update the code availability section accordingly). If DOI versions do not exist, could you please determine whether it is compatible with the licenses of the models to upload the exact versions to zenodo and provide a DOI that way? If neither is possible, could you please amend the code availability section to indicate the reason that the model codes are not provided in a long-term archive (consistent with GMD policy "Where the authors cannot, for reasons beyond their control, publicly archive part or all of the code and data associated with a paper, they must clearly state the restrictions" https://www.geoscientific-model-development.net/policies/code_and_data_policy.html)

We have created Zenodo repositories with the SAM and CESM codes and have updated the code availability section of the paper accordingly.

* line 521-522: "the simulations with different domain sizes generally have similar mean statistics" <-- please consider rephrasing this, especially considering how much precipitable water differs at 1.25 K in the MW_300dT1p25longwide and MW_300dT1p25long

We have rephrased this to "Despite the differences in convective structures discussed above, the simulations with different domain width generally have similar mean statistics (Figure \ref{fig:statevol-domainsize}). The \texttt{wide} simulations at both $\Delta SST$ = 1.25 K and $\Delta SST$ = 2.5 K have similar mean values and temporal variability as their narrower counterparts. The simulations with different domain length exhibit more differences…"

* Figure 11 - it may help readers to more easily interpret the figure if panels are provided above a--d showing SST vs X for each simulation (much of the discussion surrounding Figure 11 focuses on the correspondence between high SST and convective regions

We have added SST panels in Figure 11 and Figure 12 as suggested.

* Figure 11 - there is unexplained white space at the bottom of panels c & d. I assume that the first five simulations days were omitted for some reason? Please indicate this in the text.

The output for the first five simulation days was corrupted. We have added a note about this to the caption.

* Figure 3 - the colorbars are saturated in precipitation, pressure velocity, and OLR panels. How extreme are the values in the simulation? If the scale shown is preferred for visualization purposes, please indicate the min/max values in the figures or in the caption

We selected the colorbar scale for visualization purposes, so we have indicated the min/max values in the caption. The maximum and minimum values for the precipitation are 93.94/7.73e-07 (mm/day), for the vertical velocity -1263.09/118.18 (hPa/day) and for the OLR 102.57/315.85 (W/m^2).

* Figure 2 - please consider reducing the range of the color scale for w: barely anything is visible in the panel

We double checked the range of values and changed the color scale to -0.4 to 0.4.

* lines 285-286 - perhaps I'm overlooking something, but if the solar constant is set to a fixed value, then does the zenith angle have any effect? If so, it seems redundant to set the zenith angle to a fixed value too. (Maybe it factors in to the surface albedo calculation in some models, but if so, wouldn't that mean that the model's energy budget is somewhat dependent on the surface parameterization? If so, that's worth mentioning)

This set-up follows RCEMIP-I. With no diurnal cycle (in which zenith angle would vary) insolation is calculated based on the solar constant and zenith angle, so both need to be specified. As described in Wing et al. (2018):

*A reduced solar constant of 551.58 W m$^{-2}$ and a fixed zenith angle of 42.05$^{\circ}$ should be used (Table 2); these values yield an insolation of 409.6Wm$^{-2}$, equal to the tropical (0–20$^{\circ}$) annual mean. The zenith angle is equal to the average insolation-weighted zenith angle between the Equator and 20$^{\circ}$ (see Cronin, 2014). The surface albedo is to be fixed at a value of 0.07, corresponding to its insolation-weighted globally averaged value.*

We have revised the paper to state: Following \citet{wingetal2018}, a reduced solar constant of 551.58 W m$^{-2}$, a fixed zenith angle of 42.04$^\circ$, and a fixed surface albedo of 0.07 should be used. The values of zenith angle and surface albedo are equal to the Equator to 20$^\circ$ and global average insolation-weighted values, respectively \citep{cronin2014}.

* line 246 - "$\lambda \ne L_x$ introduces substantial noise at higher wavenumbers (contributing to the majority of the variance of dSST$^2$/dx$^2$)" <-- Is this speculation, or was this tested and not shown? Please clarify.

This was tested but is not shown. We have clarified this in the manuscript.

* lines 187-188 - "the RCE_large300 simulation from RCEMIP-I should be repeated with the new version of the model, to represent a reference point" <-- the way this is phrased seems to conflict with the statement--that is repeated several times--that participation in RCEMIP-I is not required. Consider rephrasing to "if possible, the RCE_large300 simulation from RCEMIP-I..."

We meant that if the model *did* participate in RCEMIP-I, they should use the same version as they did then for fair comparison. If they can't use the same version, but still want to compare, then they should re-run the RCEMIP-I simulation with the new version.

*We have clarified this in the paper.*

\* line 177 - the capitalized, bold NOT certainly draws the reader's attention: especially since modern, informal text communication styles tend to use capitalization to denote yelling. That said, this statement is repeated multiple times throughout and seems to come across clearly. Consider just using `\emph{not}` instead.

*We have changed this to be lowercase and bold.*